# Ubiquitination regulates ER-phagy and remodelling of endoplasmic reticulum

Alexis González[1,11], Adriana Covarrubias-Pinto[1,11], Ramachandra M. Bhaskara[1,2,3], Marius Glogger[4], Santosh K. Kuncha[1,2], Audrey Xavier[1,2], Eric Seemann[5], Mohit Misra[1,2], Marina E. Hoffmann[1], Bastian Bräuning[6], Ashwin Balakrishnan[4], Britta Qualmann[5], Volker Dötsch[7], Brenda A. Schulman[6], Michael M. Kessels[5], Christian A. Hübner[8], Mike Heilemann[4], Gerhard Hummer[3,9] & Ivan Dikić[1,2,10 ✉]

The endoplasmic reticulum (ER) undergoes continuous remodelling via a selective autophagy pathway, known as ER-phagy[1]. ER-phagy receptors have a central role in this process[2], but the regulatory mechanism remains largely unknown. Here we report that ubiquitination of the ER-phagy receptor FAM134B within its reticulon homology domain (RHD) promotes receptor clustering and binding to lipidated LC3B, thereby stimulating ER-phagy. Molecular dynamics (MD) simulations showed how ubiquitination perturbs the RHD structure in model bilayers and enhances membrane curvature induction. Ubiquitin molecules on RHDs mediate interactions between neighbouring RHDs to form dense receptor clusters that facilitate the large-scale remodelling of lipid bilayers. Membrane remodelling was reconstituted in vitro with liposomes and ubiquitinated FAM134B. Using super-resolution microscopy, we discovered FAM134B nanoclusters and microclusters in cells. Quantitative image analysis revealed a ubiquitin-mediated increase in FAM134B oligomerization and cluster size. We found that the E3 ligase AMFR, within multimeric ER-phagy receptor clusters, catalyses FAM134B ubiquitination and regulates the dynamic flux of ER-phagy. Our results show that ubiquitination enhances RHD functions via receptor clustering, facilitates ER-phagy and controls ER remodelling in response to cellular demands.

FAM134B is a mammalian reticulon-like protein that shapes the ER membrane[3,4]. It is also an ER-phagy receptor, mediating the fragmentation and selective degradation of ER sheets[3]. Structural modelling and molecular simulations have revealed that the RHD of FAM134B forms wedge-shaped membrane inclusions that induce positive membrane curvature to promote ER fragmentation, assisted by RHD clustering[4–6]. However, it is unclear how FAM134B-mediated ER-phagy is regulated in mammalian cells.

## FAM134B is ubiquitinated at the RHD

Ubiquitination regulates a large number of cellular processes, so we used mass spectrometry (MS) to investigate its potential role in ER-phagy by mapping the ubiquitination profile of FAM134B (Fig. 1a). Proteomic analysis of FAM134B-derived diGly peptides identified residues K90, K160, K264 and K247 as the primary ubiquitination sites (Fig. 1b), which are located within the cytosolic segments of the FAM134B RHD (Fig. 1c). The induction of ER-phagy with Torin 1 increased the representation

of all four diGly peptides (Fig. 1b) as well as the overall endogenous ubiquitination level of haemagglutinin (HA)-tagged FAM134B (Fig. 1d,e and Extended Data Fig. 1a, compare lanes 3 and 1, and Extended Data Fig. 1b). The accumulation of total and ubiquitinated endogenous FAM134B in cells treated with bafilomycin A1 (BafA1) indicated lysosomal degradation (Extended Data Fig. 1a, compare lane 4 to lanes 3 and 2, and Extended Data Fig. 1b). No accumulation was observed in cells treated with the proteasome inhibitor MG132 (Extended Data Fig. 1c). Accordingly, cycloheximide chase experiments showed that BafA1 rendered FAM134B more stable (Extended Data Fig. 1d,e). We also found that TAK243, a potent inhibitor of the ubiquitin (Ub)-activating enzyme, abolished endogenous FAM134B ubiquitination (Extended Data Fig. 1f) and delayed its basal turnover (Extended Data Fig. 1g,h). As an alternative method, we substituted lysine residues identified by MS (K90, K160, K247 and K264) and their neighbours (K252, K265, K278 and K291) with arginine, but overall ubiquitination levels, binding to LC3B-II and the number of FAM134B–LC3B-decorated ER fragments were not affected in this mutant (Extended Data Fig. 1i,j). Only the replacement of the entire

[1]Institute of Biochemistry II, Faculty of Medicine, Goethe University Frankfurt, Frankfurt am Main, Germany. [2]Buchmann Institute for Molecular Life Sciences, Goethe University Frankfurt, Frankfurt am Main, Germany. [3]Department of Theoretical Biophysics, Max Planck Institute of Biophysics, Frankfurt am Main, Germany. [4]Institute of Physical and Theoretical Chemistry, Goethe University Frankfurt, Frankfurt, Germany. [5]Institute of Biochemistry I, Jena University Hospital, Friedrich Schiller University Jena, Jena, Germany. [6]Department of Molecular Machines and Signaling, Max Planck Institute of Biochemistry, Martinsried, Germany. [7]Institute of Biophysical Chemistry, Center for Biomolecular Magnetic Resonance, Goethe University Frankfurt, Frankfurt, Germany. [8]Institute of Human Genetics, University Hospital Jena, Friedrich Schiller University, Jena, Germany. [9]Institute of Biophysics, Goethe University Frankfurt, Frankfurt am Main, Germany. [10]Fraunhofer Institute of Translational Medicine and Pharmacology, Frankfurt am Main, Germany. [11]These authors contributed equally: Alexis González, Adriana Covarrubias-Pinto. ✉e-mail: dikic@biochem2.uni-frankfurt.de

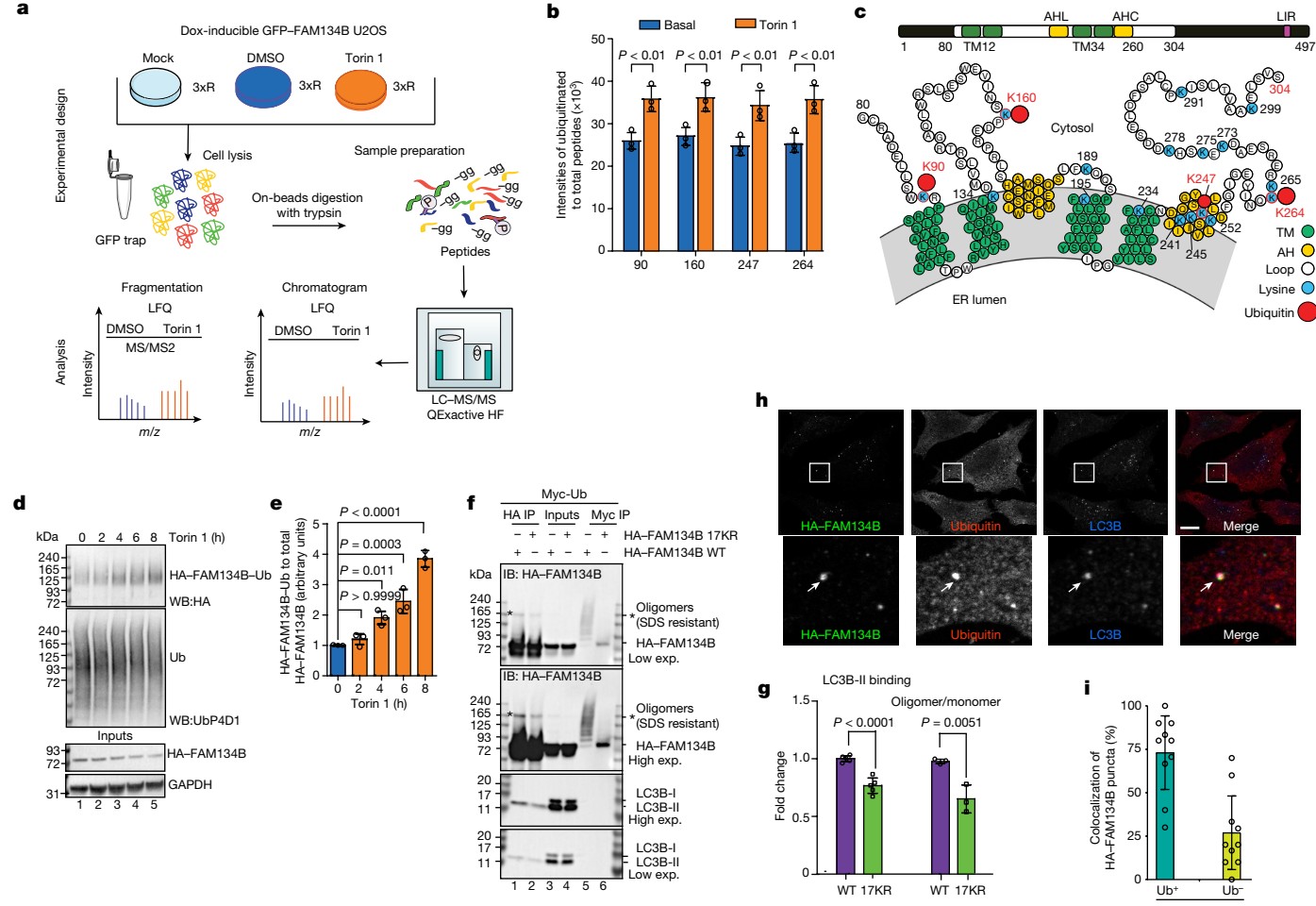

**Fig. 1 | Ubiquitination of FAM134B RHD. a**, MS workflow for the analysis of GFP–FAM134B co-immunoprecipitated from cell lysates following treatment with Torin 1 (6 h, orange) and DMSO (basal, blue) or from mock-treated cells (GFP empty, light blue). HF, Qexactive HF (Ultra-High-Field Orbitrap) mass spectrometer; LFQ, label free quantification; 3xR 3 biological replicates. Panel **a** was partly generated using Servier Medical Art (Servier), licensed under a Creative Commons Attribution 3.0 unported license. **b**, Ubiquitination of FAM134B under control conditions and in response to 250 nM Torin 1 for 6 h. The diGly peptide intensities of FAM134B are normalized to the total intensities of modified and non-modified FAM134B peptides (data are mean ± s.d.; $n = 3$ independent experiments, two-way ANOVA, Bonferroni post-hoc test). **c**, Schematic organization of FAM134B. The RHD consists of two transmembrane segments (TM, green) separated by a linker and two conserved amphipathic helices (AH, yellow). The conserved lysine residues (blue) and ubiquitinated lysines (red) are highlighted. **d**, TUBE-2 pulldown assay showing increased time-dependent ubiquitination of FAM134B following Torin 1 treatment. WB, western blot. **e**, Densitometric quantification of the immunoblot signals for ubiquitinated HA–FAM134B normalized to total HA–FAM134B levels (**d**). Data are mean ± s.d.; $n = 3$ independent experiments; one-way ANOVA, Bonferroni post-hoc test. **f**, FAM134B RHD ubiquitination assay in cells showing a reduction of ubiquitination when the 17 conserved lysine residues are replaced with arginine (Myc-Ub immunoprecipitation (IP)). Lack of RHD ubiquitination reduces binding to LC3B-II and the abundance of oligomeric species (HA–FAM134B IP). IB, immunoblot; exp., experiment. **g**, Densitometric quantification of the immunoblot signals from **f** (lanes 1 and 2): LC3B-II bound to HA–FAM134B WT or HA–FAM134B 17KR, and the oligomers. Data are mean ± s.d.; $n = 5$ and $n = 3$ independent experiments, respectively; one-tailed unpaired Student's $t$-test. **h**, Confocal fluorescence microscopy analysis of HA–FAM134B co-labelled with LC3B and Ub in cells treated with 250 nM Torin 1 for 2 h. Arrows indicate autophagosome (LC3B positive in blue) that colocalizes with HA FAM134B (green) and ubiquitin (red). Scale bar, 10 μm. **i**, Quantification of the fluorescence signal of HA–FAM134B–LC3B puncta that colocalizes or not with Ub from **h** (based on Pearson's correlation coefficients). Data are mean ± s.d., $n = 10$ cells.

set of 17 highly conserved lysine residues (mutant HA–FAM134B17KR) resulted in a strong decrease in FAM134B RHD ubiquitination (Fig. 1f, compare lanes 6 and 5). Of note, the lack of FAM134B RHD ubiquitination correlated with decreased binding to LC3B-II (Fig. 1f, compare lanes 2 and 1, and Fig. 1g; approximately 1.32-fold reduction). High-molecular-weight (oligomeric) species of FAM134B were less abundant in the HA–FAM134B17KR mutant (Fig. 1f, compare lanes 2 and 1, and Fig. 1g; approximately 1.5-fold reduction). Similar results were observed following the chemical crosslinking of intact membranes from cells expressing the wild-type (HA–FAM134BWT) or mutant (HA–FAM134B17KR) receptor are consistent with the high-molecular-weight species representing oligomers (Extended Data Fig. 1k).

Several results indicated that ubiquitination of the FAM134 RHD can promote the formation and/or stabilization of FAM134B oligomers implicated in ER-phagy. First, high-molecular-weight species of FAM134B accumulate when the flux of ER-phagy is inhibited and/or following the deletion of the LC3-interacting region (LIR), indicating that FAM134B is delivered to lysosomes in its oligomeric form (Extended Data Fig. 1l,m). Second, in the absence of the FAM134B LIR, the ubiquitinated forms of FAM134B accumulate, confirming its autophagy-mediated degradation (Extended Data Fig. 1n). Third, immunofluorescence analysis following exposure to Torin 1 showed that approximately 70% of FAM134B⁺ autophagosomes contained Ub, suggesting Ub has a wide-engaging role in FAM134B-driven ER-phagy

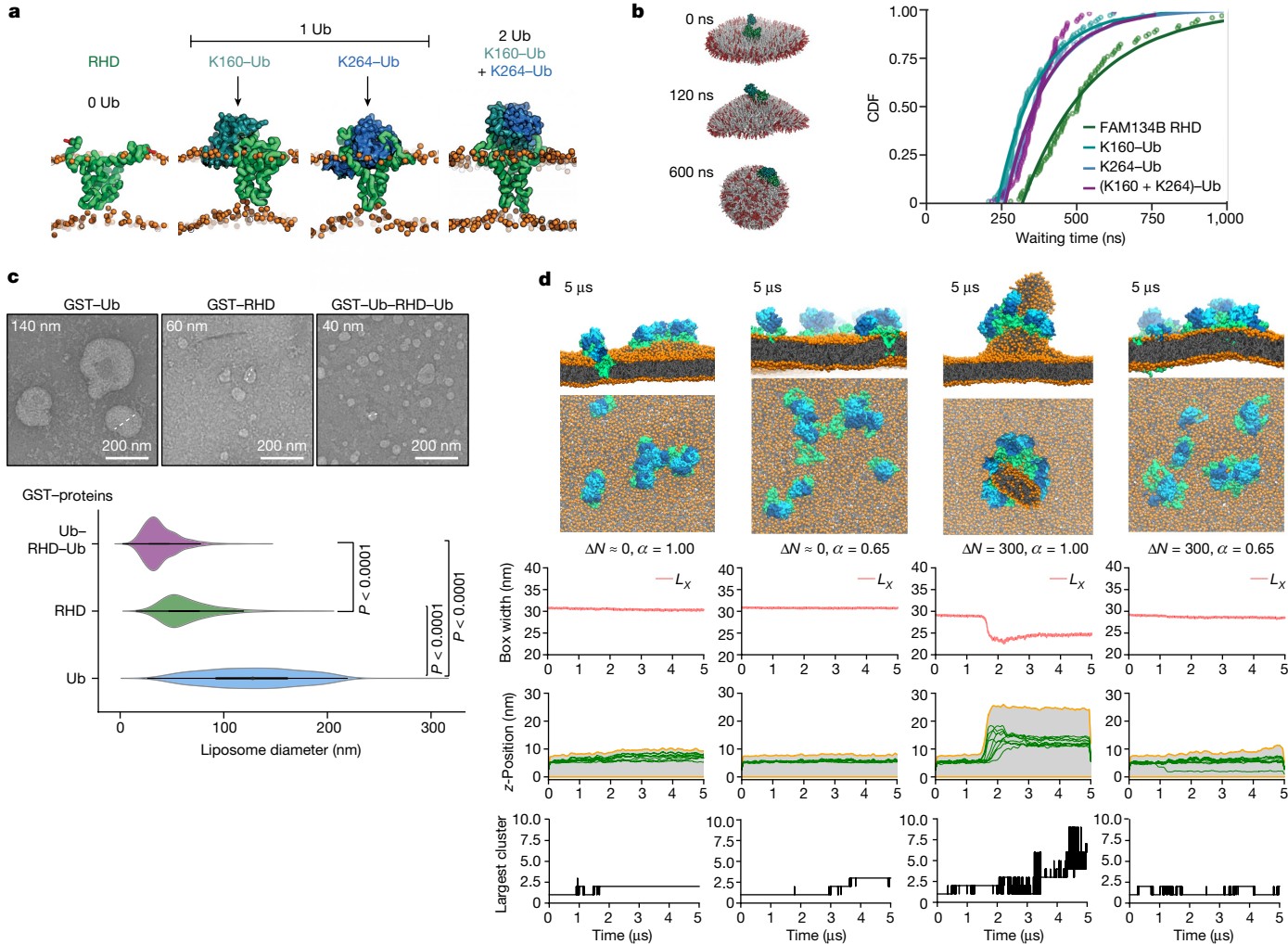

**Fig. 2 | Role of Ub in the structure and function of RHD. a**, Equilibrated structures of ubiquitinated and non-ubiquitinated FAM134B RHD variants in MD simulations highlight the arrangement of Ub moieties (cyan and blue) with respect to the RHD (green) and the POPC bilayer (orange beads). **b**, Ubiquitination accelerates membrane curvature induction. K160–Ub (left) induces faster transitions of bicelles to closed vesicles than non-ubiquitinated RHDs (cumulative distribution function (CDF) of waiting times shown on the right; compare cyan, blue and purple versus green). **c**, In vitro liposome remodelling experiments using purified protein samples (GST–Ub, GST–RHD$_{90-264}$, GST–Ub–RHD$_{90-264}$–Ub) incubated with liposomes for 8 h at 25 °C. The top panel shows representative negative-stain transmission electron micrographs. Remodelled proteoliposomes were quantified by measuring their diameters (dotted lines) using ImageJ (version 1.51w). The bottom panel shows violin plots of liposome size distributions. Violin plots show the boxplots with median value (black dot), the interquartile range (black shaded region), the minimum and maximum values (1.5× the interquartile region)

and mirrored probability density estimates on sides (coloured shaded region). GST–RHD mean = 64.40 nm; GST–Ub–RHD$_{90-264}$–Ub mean = 39.10 nm; GST–Ub mean = 127.47 nm. GST–RHD$_{90-264}$ $n$ = 625; GST–Ub–RHD$_{90-264}$–Ub $n$ = 961; GST–Ub $n$ = 1,573; Kruskal–Wallis or Dunn's post-hoc test. **d**, Ub promotes receptor clustering and membrane remodelling. Snapshots show the arrangement of nine (K160 + K264)–Ub–RHD molecules in model POPC bilayers (orange) at the end of the simulations. MD simulations were performed under four different conditions by altering bilayer asymmetry, $\Delta N$ = 0 and 300, and protein–protein interaction strength, $\alpha$ = 1 and 0.65, as quantified in the panels below. For the rows: time traces of the box width ($L_x$) during the four simulations (top); vertical displacement ($z$) of individual ubiquitinated proteins (centre-of-mass positions shown as green lines), with the highest and lowest points in the membrane shown as orange lines, and the intervening range in light grey (middle); the size of the largest protein cluster as a function of time for different simulation conditions (bottom).

(Fig. 1h,i). These results show that the ubiquitination of the FAM134 RHD can promote the formation and/or stabilization of FAM134B oligomers implicated in ER-phagy.

## Ubiquitination enhances and fine tunes RHD membrane remodelling functions

We studied the effect of ubiquitination on RHD structure and dynamics with coarse-grained MD simulations of mono-ubiquitinated and bi-mono-ubiquitinated FAM134B RHD molecules, namely, K160–Ub, K264–Ub and (K160 + K264)–Ub, embedded in POPC (16:0/18:1 PC) lipid bilayers. The most populated conformations of the ubiquitinated

FAM134B RHD molecules are shown in runs of up to 10 μs each (Fig. 2a). The Ub moieties perturb the RHD structure locally by mediating multiple interactions with the proximal cytosolic loops of the RHD and the POPC bilayer (Extended Data Fig. 2a–d). The main hydrophobic face of K160–Ub makes substantial contact with POPC lipids, widening the wedge shape of the RHD (Extended Data Fig. 2b). In the RHD of K264–Ub, Ub interacts primarily with the cytosolic loops and is located on top of the RHD (Extended Data Fig. 2c,d). In the bi-mono-ubiquitinated variant (K160 + K264)–Ub, the two Ub moieties mediate *cis*-interactions with each other and are bundled on top of the RHD (Extended Data Fig. 2e,f). Increased radii of gyration of ubiquitinated RHDs result in a larger footprint on the bilayer, thus possibly perturbing it more severely

(Extended Data Fig. 2g,h). These changes in intrinsic RHD structure due to ubiquitination may affect its membrane curvature induction and sensing functions.

To test whether ubiquitination affects the RHD-mediated induction of membrane curvature, we used in silico simulations to remodel a discontinuous bicelle (DMPC + DHPC lipids) into a closed vesicle[4] at 300 K (Fig. 2b and Supplementary Videos 1–3). The ubiquitinated and non-ubiquitinated RHDs induced a positive mean curvature ($+H$) leading to vesicle formation within the first 500 ns ($3 \times 100$ replicates; 1 μs each; Extended Data Fig. 3a–d). By measuring the kinetics and rates for vesicle formation (see Methods; Extended Data Fig. 3e,f), we estimated that single K160–Ub, K264–Ub and (K160 + K264)–Ub species accelerated vesicle formation moderately by factors of 1.49, 1.37 and 1.38, respectively (Fig. 2b and Extended Data Fig. 3g). To highlight the effect of Ub on curvature induction, we performed these assays at 280 K, which increases the energy barrier to induce curvature and vesiculation. We found that bicelles containing the FAM134B RHD and K160–Ub did not effectively transit to closed vesicles (Extended Data Fig. 3h,k), whereas bicelles containing K264–Ub or (K160 + K264)–Ub formed closed vesicles, indicating that these variants can overcome the barrier even at 280 K (Extended Data Fig. 3i,j). Lateral diffusion of ubiquitinated variants in buckled bilayers showed no differences in preferred curvature, indicating that the strong curvature-sensing function of the RHD is preserved upon ubiquitination (Extended Data Fig. 4a–c).

Next, we tested how ubiquitination affects the FAM134B RHD-mediated remodelling of liposomes in vitro. We created and purified N-terminal and C-terminal gene fusions of Ub to FAM134B RHD$_{90–264}$ (described as Ub–RHD–Ub). As a control, we created a construct encompassing the entire RHD of FAM134B without Ub (RHD$_{90–264}$). The incubation of liposomes with Ub–RHD–Ub led to significantly smaller structures with a narrow distribution than the liposomes incubated with RHD proteins alone (Fig. 2c), indicating a significant gain of membrane-remodelling activity for ubiquitinated RHD proteins. In cells, both constructs localized to the ER, based on the ER marker REEP5. We observed a significant increase in the number of RHD–REEP5-containing puncta in cells expressing Ub–RHD–Ub. These puncta may represent clusters of RHD-containing proteins, enhanced by the presence of ubiquitinated FAM134B RHDs (Extended Data Fig. 4d,e).

## Ub interactions facilitate large-scale membrane deformations

Next, we simulated the behaviour of ubiquitinated RHDs to induce curvature-mediated protein sorting and clustering (Extended Data Fig. 4f). In the absence of ubiquitination, the two RHD molecules on the buckled membrane diffused to the top of the buckle, where they formed a loose cluster. Ubiquitination slowed down the diffusion somewhat but resulted in a tighter cluster at the top of the buckle that persisted for the entire simulation (up to 25 μs). From 19 μs onwards, Ub-mediated contacts stabilized the dimeric complex (inset in Extended Data Fig. 4f and Supplementary Video 4). Next, we simulated ten ubiquitinated FAM134B RHD molecules (five K160–Ub and five K264–Ub) embedded in a closed tubule (up to approximately 8.5 μs). The Ub moieties initiated trans-interactions among the RHDs and enabled the formation of three protein clusters (dimers and trimers) on the MD timescale (Supplementary Video 5). These clusters deformed the tubule in both principal directions (squares in Extended Data Fig. 4g).

Motivated by these locally bud-shaped structures, we studied the effect of ubiquitination on spontaneous membrane budding by simulating nine bi-mono-ubiquitinated RHDs ((K160 + K264)-Ub) embedded in POPC bilayers under four different simulation conditions (Fig. 2d). We increased the lipid-number asymmetry of the bilayer leaflets from $\Delta N = 0$ to 300 to increase the energetic driving force for budding, and we reduced the protein–protein interaction (PPI) strength from 100%

($\alpha = 1$) to 65% ($\alpha = 0.65$)[6] to weaken Ub-mediated protein clustering. Spontaneous budding was observed in the asymmetric bilayer with 100% PPI strength (Fig. 2d and Supplementary Video 6), triggered by the formation of a cluster of ubiquitinated RHDs on top of the nascent bud. After budding, the remaining ubiquitinated RHDs sorted onto the membrane bud to form a Ub-rich protein coat in the form of a ring around its neck. Ub–Ub interactions are characteristic of these RHD clusters (blue and cyan moieties in Fig. 2d). By contrast, in all the other three simulations, RHD clusters only formed transiently (because of reduced PPI strength) and induced at most local membrane bulges (because of the high-energy penalty for budding at low asymmetry, $\Delta N = 0$). We conclude that the stabilization of RHD clusters by Ub-mediated interactions facilitates FAM134B-induced membrane budding.

Analysis of all possible Ub–Ub interactions (Extended Data Fig. 5a–d) indicated that intermolecular or trans-Ub–Ub interactions triggered protein clustering and oligomerization (Extended Data Fig. 5c). Although changing the membrane asymmetry and PPI strength during the simulations did not change the intramolecular or cis-Ub–Ub interactions, they affected the character of intermolecular trans-Ub–Ub interactions (Extended Data Fig. 5a–d, left versus right). We also found that the membrane asymmetry enhanced the lifetime of the Ub–Ub interactions, stabilizing them to organize as a cluster (Extended Data Fig. 5e). Furthermore, whereas the intramolecular cis-interactions were predominantly mediated by specific residues on the Ub surface, namely, those forming hairpins β12, β34 and β5 (Extended Data Fig. 6a), the intermolecular trans-Ub–Ub interactions required for cluster formation were spread all over the Ub surface, indicating that no specific residues were dominant (Extended Data Fig. 6b). When the PPI strength was reduced to 65%, the alternative interactions that emerged were also spread over the surface, confirming the absence of specific interactions. Nonspecific Ub–Ub interactions create a crowded membrane environment with multiple RHDs, causing proteins to sort and aggregate locally in the membrane. These crowded regions appear to be driven by volume-exclusion effects and curvature-mediated protein-sorting mechanisms. Thus, volume exclusion in combination with multiple low-affinity nonspecific Ub–Ub interactions result in the nucleation of RHD clusters, which then induce membrane bud formation. The high curvature of the membrane bud stabilizes the RHD clusters, increases their longevity and further favours the sorting of individual Ub–RHDs to the site of the bud.

## RHD ubiquitination increases FAM134B cluster size and the flux of ER-phagy

Given that FAM134B functions in membrane remodelling, we hypothesized that ubiquitination affects FAM134B-driven ER fragmentation in cells. Using high-content imaging, we observed that the basal number and size of FAM134B–LC3B-decorated ER fragments were significantly lower in cells expressing the FAM134B 17KR mutant than in the WT control (Extended Data Fig. 7a–c). In vitro liposome remodelling driven by recombinant full-length GST-tagged FAM134B was not affected by the 17KR mutant, indicating that the folding and intrinsic activity of FAM134B were not impaired by the mutations (Extended Data Fig. 7d,e). Compared with liposomes treated with GST control, GST–FAM134B WT and 17KR decreased the liposome diameter to a similar extent (Extended Data Fig. 7f). Next, we investigated whether ubiquitination of the FAM134B RHD also affects the flux of ER-phagy using two validated reporter assays[3,7,8]. First, we generated cells expressing inducible mCherry–eGFP-tagged FAM134B 17KR or its WT counterpart (Fig. 3a). As expected, the reporter localized to the ER and the mCherry signal was concentrated in microscale puncta corresponding to autophagosomes or lysosomes (Fig. 3b). The lack of FAM134B RHD ubiquitination significantly reduced the flux of ER-phagy following Torin 1 or Earle's Balanced Salt Solution (EBSS) treatment (Fig. 3c). Similarly, the flux of ER-phagy was reduced when FAM134B WT was

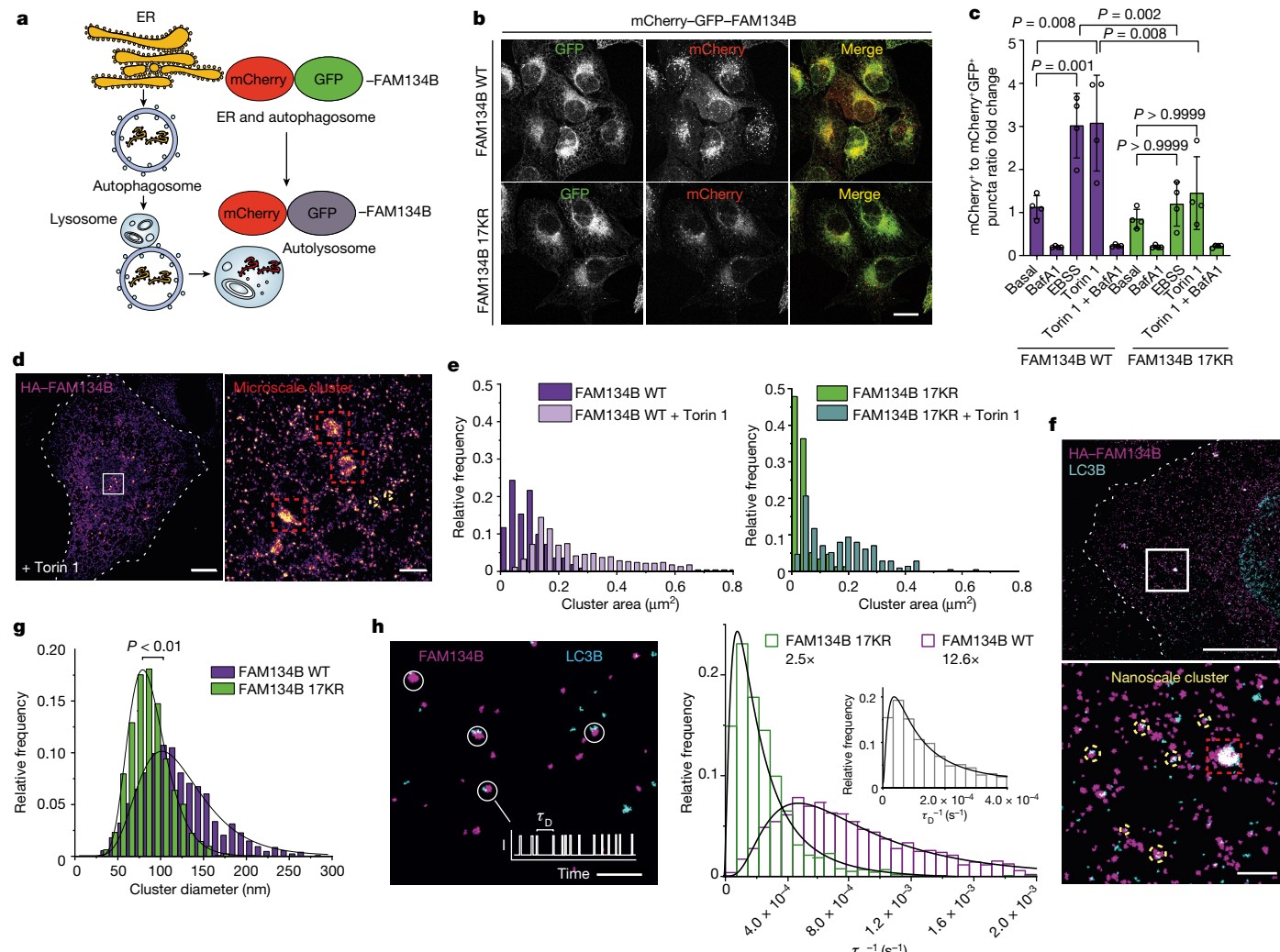

**Fig. 3 | Effect of RHD ubiquitination on the flux of ER-phagy and FAM134B cluster size. a**, The ER-phagy reporter system mCherry–GFP–FAM134B. **b**, U2OS TRex stable cell lines expressing mCherry–GFP–FAM134B WT or mCherry–GFP–FAM134B 17KR. Scale bar, 10 µm. **c**, ER-phagy flux was quantified as the ratio between mCherry⁺GFP⁻ and mCherry⁺GFP⁺ puncta. $n = 4$ independent experiments in which the total number of cells per condition for mCherry–GFP–FAM134B WT were: 482 (basal (DMSO)), 738 (BafA1), 673 (EBSS), 842 (Torin1) and 667 (Torin1 + BafA1). The number of cells per condition for mCherry–GFP–FAM134B 17KR: 440 (DMSO), 864 (BafA1), 723 (EBSS), 968 (Torin1), 535 (Torin1 + BafA1). Data are mean ± s.d.; one-way ANOVA, Bonferroni post-hoc test. **d**, DNA-PAINT super-resolution image of HA–FAM134B. Microscale (red) and nanoscale (yellow) clusters (red) are indicated. Scale bars, 10 µm (left panel) and 1 µm (right panel; magnified region from the left panel). **e**, Relative frequency distribution of HA–FAM134B WT and HA–FAM134B 17KR cluster areas ($110 > n_{cluster} < 251$) identified in U2OS cells under basal conditions ($n_{WT} = 10$ cells, $n_{17KR} = 8$ cells) or following Torin1 treatment ($n_{WT} = 11$ cells, $n_{17KR} = 14$ cells). **f**, DNA-PAINT super-resolution imaging of HA–FAM134B WT and

LC3B-II nanoscale clusters (yellow dashed circles). Dashed white line in top panel indicates cell outline. Scale bars, 10 µm (top panel) and 1 µm (bottom panel; magnified region from the top panel). **g**, Relative frequency distribution of the diameter of HA–FAM134B and HA–FAM134B 17KR nanoscale cluster ($n_{cells} = 4$, $n_{WT\ clusters} = 1,278$; $n_{17KR\ clusters} = 1,255$). Histograms were fitted with a log-normal distribution followed by a non-parametric one-tailed Mann–Whitney $U$-test. Cluster diameters were determined from the mode of the log-normal distribution using the mean and standard deviation (HA–FAM134B 17KR: $\mu = 88$ nm, $\sigma = 24$ nm; HA–FAM134B WT: $\mu = 123$ nm, $\sigma = 46$ nm). **h**, Quantitative analysis of FAM134B copy numbers in nanoscale clusters. Scale bar, 1 µm. Relative frequency distribution of the inverse dark times ($\tau_D$) of single-molecule binding time intervals recorded with DNA-PAINT (right). In the inset, the grey bars indicate calibration. Histograms were fitted with a log-normal distribution: FAM134B 17KR $\mu = 3.1 \times 10^{-4}$ s⁻¹, $\sigma = 3.5 \times 10^{-4}$ s⁻¹; FAM134B WT $\mu = 9.4 \times 10^{-4}$ s⁻¹, $\sigma = 7.1 \times 10^{-4}$ s⁻¹; calibration cluster $\mu = 1.9 \times 10^{-4}$ s⁻¹, $\sigma = 2.8 \times 10^{-4}$ s⁻¹.

exposed to the E1 inhibitor TAK243 (Extended Data Fig. 7k). Second, in cells expressing the ER-phagy flux reporter ssRFP–GFP–KDEL[8], the flux of ER-phagy (basal and induced by Torin1 or EBSS) was reduced in cells co-expressing HA–FAM134B 17KR compared with the WT version (Extended Data Fig. 7g–j). Accordingly, less EBSS-induced REEP5 degradation was observed in the ssRFP–GFP–KDEL/HA–FAM134B 17KR cells (Extended Data Fig. 7j).

We next investigated the ultrastructure of FAM134B clusters by single-molecule localization microscopy[9] in U2OS cells expressing HA–FAM134B WT or HA–FAM134B 17KR. Using DNA point accumulation

in nanoscale topography (DNA-PAINT)[10], we identified microscale and nanoscale clusters of HA–FAM134B (Fig. 3d). The microscale clusters corresponded to the puncta observed by confocal microscopy (Extended Data Fig. 7a). Furthermore, 2D and 3D high-resolution images showed that microscale HA–FAM134B clusters colocalized with the autophagosomal membrane marker LC3B-II (Extended Data Fig. 7l and Supplementary Video 7). We determined the size of HA–FAM134B WT and HA–FAM134B 17KR microscale clusters by Voronoi tessellation[11], indicating the area for the heterogeneous morphologies that we observed. Torin1 treatment resulted in larger clusters for HA–FAM134B

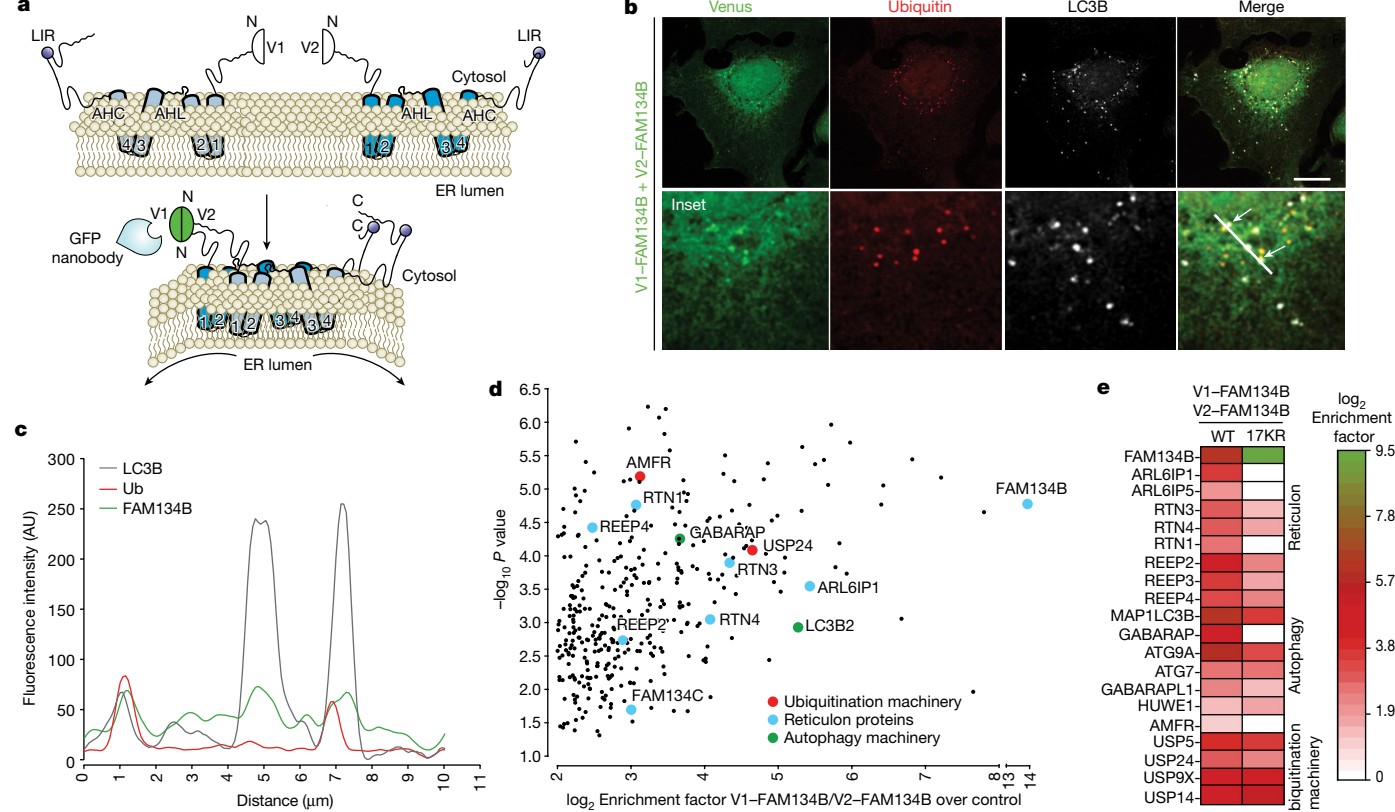

**Fig. 4 | Ubiquitination site profiling and protein interactors of FAM134B-containing oligomers. a**, Schematic representation of the bimolecular complementation affinity purification assay of FAM134B dimers. The full-length FAM134B was fused to the C-terminal of the two non-fluorescent complementary fragments of the Venus fluorescent protein (V1–FAM134B and V2–FAM134B). **b**, Confocal fluorescence microscopy analysis of the bimolecular fluorescence complementation signal resulting from the interaction between V1–FAM134B and V2–FAM134B. Fixed cells expressing V1–FAM134B and V2–FAM134B were stained with anti-Ub(FK2) (red) or anti-LC3B (grey). Arrows indicate triple colocalization of the Venus signal (green), ubiquitin (red) and LC3B (grey). Scale bar, 10 μm. **c**, Histogram analysis of the fluorescence intensity distribution reveals that dimeric FAM134B colocalized into Ub+ and LC3B+ vesicles. **d**, Single-sided volcano plot of the quantitative label-free interactome of FAM134B homodimers depicting RHD-containing ER proteins (blue), autophagy-related proteins (green) and ubiquitination machinery (red). Data represent three independent experiments, one-sided unpaired Student's *t*-test. **e**, Heatmap comparing the interaction of FAM134B WT homodimers versus FAM134 17KR homodimers with RHD-containing ER proteins, autophagy-related proteins and ubiquitination machinery. Interaction partners with $\log_2$ enrichment > 2.0 and $-\log_{10} P$ > 1.3 were plotted and compared (one-sided unpaired Student's *t*-test).

WT and the ubiquitination-deficient mutant (Fig. 3e). However, the cluster areas were significantly larger for HA–FAM134B WT than the mutant under control conditions and following exposure to Torin 1 (Fig. 3e and Extended Data Fig. 7m). Super-resolution images ($28 \pm 2$ nm, mean $\pm$ s.d.)[11] revealed the existence of HA–FAM134B nanoclusters much smaller than the resolution of light microscopy (Fig. 3d, dotted dashed circle). Co-labelling with REEP5 revealed that HA–FAM134B nanoscale clusters were distributed within the ER network (Extended Data Fig. 7n). Using DNA-PAINT super-resolution imaging, we found that a subset of HA–FAM134B nanoscale clusters colocalized with the autophagosomal membrane marker LC3B-II, and may represent ER-phagy initiation sites (Fig. 3f). Quantitative analysis of these sites using the DBSCAN clustering algorithm[12] revealed that the diameter of the clusters containing ubiquitination-deficient HA–FAM134B was significantly lower (diameter = 79 nm) than those containing FAM134B WT (diameter = 101 nm) (Fig. 3g). We inferred the number of molecules in the nanoscale clusters by applying a kinetic analysis of single-molecule DNA-PAINT data[13], revealing that ubiquitination increased the oligomeric state of FAM134B WT in nanoclusters on average by fivefold ($n_{17KR} = 2.5$, $n_{WT} = 12.6$), thus promoting the assembly of high-density clusters (Fig. 3h). These data suggest that FAM134B RHD ubiquitination increases the size of ER-phagy initiation clusters, leading to larger autophagosomal structures that tune the dynamic flux of ER-phagy.

## Protein interactors of ubiquitinated FAM134B clusters in cells

To investigate the composition of the ER-phagy receptor complexes in more detail, we used a bimolecular complementation affinity purification (BiCAP) assay to visualize the PPIs and to characterize the cluster-specific interactome[14] (Fig. 4a). The clustering of FAM134B was enriched by immunoprecipitation with anti-GFP antibodies following the co-expression of V1–FAM134B and V2–FAM134B (Extended Data Fig. 8a, compare lane 3 to lanes 2 and 1). Similar results were observed following the BiCAP of FAM134C, a paralogue that acts in concert with FAM134B[5] (Extended Data Fig. 8a, compare lane 7 to lanes 6 and 5). The analysis of diGly peptides from isolated FAM134B oligomers revealed seven ubiquitinated lysine residues within the FAM134B RHD, including the four previously detected sites (Fig. 1b and Extended Data Fig. 8b). This indicated that multiple lysine residues in the RHD can be ubiquitinated within receptor clusters. Accordingly, FAM134B clusters (observed using BiCAP) were found in ER fragments, colocalizing with LC3B and Ub (Fig. 4b,c), indicating that ubiquitinated FAM134B clusters are colocalized with LC3+ autophagosomes. We characterized the interactome of immunopurified FAM134B clusters and detected 363 significantly enriched proteins ($\log_2$ enrichment factor $\geq 2$, $P \leq 0.05$), including novel candidates that had not been detected in previous FAM134B interactome datasets[15]. MAP1LC3B

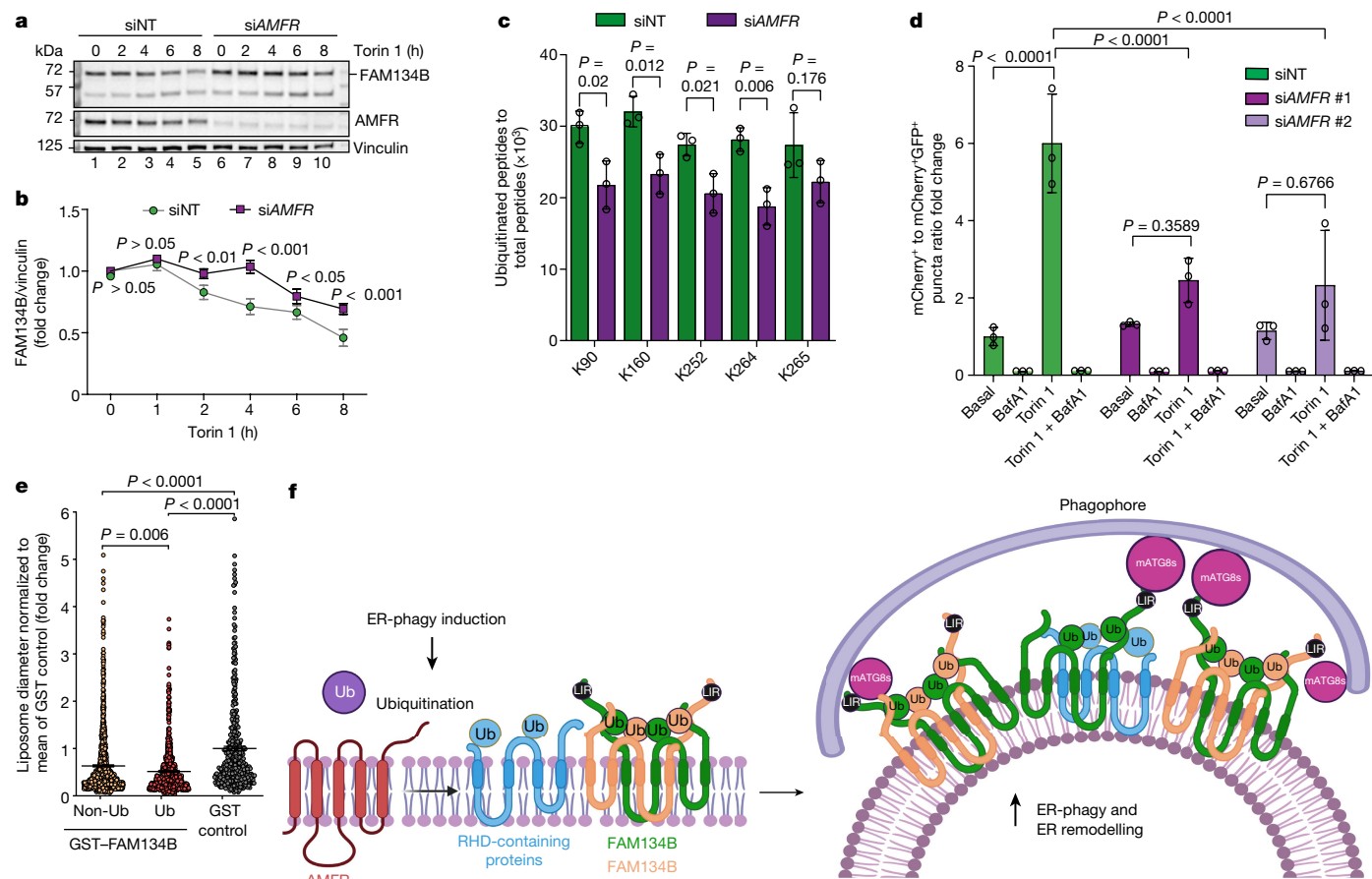

**Fig. 5 | Effect of AMFR on FAM134B RHD ubiquitination and ER-phagy.**
**a**, HeLa cells were transfected with siNT or si*AMFR*, and were treated with
250 nM Torin 1 for the indicated time. Protein extracts were analysed by western
blot for FAM134B, AMFR or vinculin. **b**, Densitometric quantification of the
western blot signal of FAM134B in **a** (data are mean ± s.d.; *n* = 3 independent
experiments; two-way ANOVA, Bonferroni post-hoc test). **c**, Ubiquitination of
SFB-tagged FAM134B in *AMFR*-knockdown cells. The diGly peptide intensities
were normalized to total intensities of modified and non-modified FAM134B
peptides (data are mean ± s.d.; *n* = 3 independent experiments; two-way
ANOVA, Bonferroni post-hoc test). **d**, U2OS TREx stable cell lines expressing
mCherry–GFP–FAM134B WT were transfected with siNT, *AMFR*-targeting
siRNA #1 or siRNA #2. Cells were treated as indicated. The flux of ER-phagy was
quantified as the ratio between mCherry⁺GFP⁻ and mCherry⁺GFP⁺ puncta. The
data are representative of three independent experiments in which the total
number of cells per condition were: 699 siNT/DMSO, 683 siNT/BafA1, 695 siNT/

Torin 1, 678 siNT/Torin 1 + BafA1, 627 si*AMFR* #1/DMSO, 668 si*AMFR* #1/BafA1,
713 si*AMFR* #1/Torin 1, 591 si*AMFR* #1/Torin 1 + BafA1, 624 si*AMFR* #2/DMSO,
593 si*AMFR* #2/BafA1, 575 si*AMFR* #2/Torin 1 and 615 si*AMFR* #2/Torin 1 + BafA1.
Data are mean ± s.d., one-way ANOVA, Bonferroni post-hoc test. **e**, Diameters
of freeze-fractured liposomes incubated with either non-ubiquitinated or
ubiquitinated GST–FAM134B. Data are mean ± s.e.m. normalized to the mean
liposome diameter of GST control, representing three independent liposome
preparations and experiments (*n* = 1,064 for the non-Ub sample, *n* = 793 for the
Ub sample and *n* = 434 for the GST control; Kruskal–Wallis/Dunn's post-test).
**f**, Model in which the E3 ligase AMFR is recruited to ER-phagy receptor clusters
to induce the ubiquitination of FAM134B. This event triggers changes in the
conformation and composition of ER-phagy receptor clusters, enabling the
clusters to grow in size, thus controlling ER remodelling and ER-phagy. mATG8s,
mammalian ATG8 proteins. The schematic in panel **f** was created using
BioRender (https://biorender.com).

and GABARAP were identified among the most enriched proteins (Fig. 4d,
green dots). In addition, we found that FAM134B homodimers strongly
interacted with several other RHDs and RHD-containing proteins (Fig. 4d,
light blue dots, and Extended Data Fig. 8g) and with proteins of the ubiq-
uitination machinery (Fig. 4d, red dots), including E3 ligases and deu-
biquitinases. Autophagy receptors containing Ub-binding domains (for
example, p62, OPTN and TAX1BP1)[16] were not detected, implying that
FAM134B ubiquitination is not a recruitment signal for these proteins.
Accordingly, there was no significant colocalization between FAM134B
clusters and p62 in cells (Extended Data Fig. 8h,i). The interactomes
of FAM134 isoforms, FAM134C heterodimers and FAM134B–FAM134C
dimers revealed overlapping sets of interaction partners (Extended
Data Fig. 8c–f). By contrast, the clustering of FAM134B 17KR reduced or
abolished interactions between FAM134B and several RHD-containing
proteins, mammalian ATG8 proteins and the ubiquitination machinery
(Fig. 4e), indicating that ubiquitination promotes the formation of mul-
timeric ER-phagy clusters.

## The E3 ubiquitin ligase AMFR catalyses FAM134B ubiquitination

The interaction partners within oligomeric FAM134B clusters included
the endogenous ER-anchored E3 ligase AMFR (also known as gp78),
which is implicated in ER-associated degradation[17] (Fig. 4d). This colo-
calized with FAM134B clusters in BiCAP experiments and with LC3B
(Extended Data Fig. 9a,b). The expression of WT AMFR, but not its
catalytically inactive counterpart (C356G H361A), increased the ubiqui-
tination of HA–FAM134B in cells (Extended Data Fig. 9c,d). Evaluating
their functional interaction in cells using BiCAP assays (Extended Data
Fig. 9e) revealed that V2–AMFR–V1–FAM134B complexes colocalized
with LC3B and Ub⁺ structures (Extended Data Fig. 9f). This effect was
reduced when AMFR was replaced with its catalytically inactive mutant
(Extended Data Fig. 9f,g). MS-based interactome analysis confirmed
that FAM134B–AMFR complexes also recruited RHD-containing pro-
teins, mammalian ATG8 proteins and components of the ubiquitination

machinery (Extended Data Fig. 9h,i). Of note, the E2-conjugating enzyme UBE2G2, which cooperates with AMFR[18], was also enriched (Extended Data Fig. 9h). The Torin 1-induced ubiquitination profile of FAM134B complexes with AMFR (WT V2–AMFR–V1–FAM134B) was significantly reduced in complexes with the catalytically inactive AMFR mutant (V2–AMFR(C356G,H361A)–V1–FAM134B) (Extended Data Fig. 9j). In addition, clustering of FAM134B with catalytically inactive AMFR reduced the interactions with other RHD-containing proteins, mammalian ATG8 proteins and the ubiquitination machinery (Extended Data Fig. 9k). Silencing the expression of AMFR caused a significant increase in total endogenous FAM134B (Extended Data Fig. 10a,b) and diminished its cellular turnover induced by Torin 1 (Fig. 5a,b). Furthermore, we observed decreased ubiquitination of S protein-FLAG-streptavidin-binding peptide (SFB)-tagged FAM134B in response to AMFR depletion (Fig. 5c), also reducing FAM134B-mediated ER fragmentation in response to Torin 1 (Extended Data Fig. 10c,d). Torin 1 treatment also reduced the levels of endogenous AMFR (Fig. 5a and Extended Data Fig. 10e). This decay was diminished in the presence of HA–FAM134B 17KR (Extended Data Fig. 10f–h) or ΔLIR (Extended Data Fig. 10i,j), indicating that active FAM134B and its ubiquitination promote efficient AMFR degradation via ER-phagy. Moreover, siRNA-mediated knockdown of *AMFR* significantly slowed the flux of FAM134B-mediated Torin 1-induced ER-phagy compared with control siRNA (siNT) (Fig. 5d; approximately 58% and 60% reduction with *AMFR*-targeting siRNA #1 (si*AMFR* #1) and si*AMFR* #2, respectively).

Next, we tested whether AMFR-mediated ubiquitination affected the membrane-shaping functions of FAM134B in vitro. First, we ubiquitinated GST-tagged FAM134B using purified recombinant AMFR (Extended Data Fig. 10k,l) and detected K160, K278 and K299 as direct targets of AMFR-dependent ubiquitination (Extended Data Fig. 10m,n). This prompted us to repeat the in vitro liposome remodelling assay in the presence of AMFR. Compared with liposomes treated with GST, the non-ubiquitinated GST–FAM134B (in the presence of AMFR, no ATP) decreased the liposome diameter, but ubiquitinated GST–FAM134B (in the presence of AMFR + ATP) reduced the diameter even further (Fig. 5e). This significant difference between non-ubiquitinated and ubiquitinated samples suggests that ubiquitination of multiple sites on full-size FAM134B promotes the formation of smaller liposomes. This agrees with the increased membrane remodelling activity for the chimaera Ub–RHD$_{90–264}$–Ub compared with non-ubiquitinated RHD$_{90–264}$ (Fig. 2c). Furthermore, the ubiquitination of FAM134B RHD$_{90–264}$ in vitro by AMFR promoted the formation of larger complexes detected by blue native polyacrylamide gel electrophoresis (Extended Data Fig. 10o–r).

## Discussion

Our data reveal an unprecedented role for Ub in the conformational changes of FAM134B that, in turn, drive ER membrane remodelling and the flux of ER-phagy. Although FAM134B RHDs inherently curve bilayers and populate regions of high local membrane curvature, the addition of Ub boosts this effect by promoting multiple *cis*-interactions and *trans*-interactions that stabilize multimeric clusters. This enables the clusters to grow in size and to nucleate large-scale membrane remodelling events in the ER (Fig. 5f). ER-phagy clusters also include multiple RHD-containing proteins such as ARL6IP1 (a recurrent finding in the interactome of FAM134B homodimers), which is also required for efficient ER-phagy[19]. In addition, the clusters contain Ub ligases and deubiquitinases, which can alter the dynamic ubiquitination of RHD-containing proteins and influence the formation and growth of these multivalent clusters. Indeed, the E3 ligase AMFR regulates ER-phagy by acting as a critical ER quality control mechanism. It is tempting to speculate that other E3 ligases can modify ER-phagy

receptor clusters in a cell-type-specific manner. The receptor clustering phenomenon depends on several interrelated factors, such as receptor abundance, distribution and other post-translational modifications (for example, phosphorylation[7] or UFMylation[20]). Studying these events will shed light on the dynamics of the entire ER-phagy pathway and will pave the way for a better understanding of defects in ER dynamics influencing the pathogenesis of many diseases[21].

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

## Methods

### Cell culture and inducible cell lines

Cell lines were cultured in DMEM supplemented with 10% heat-inactivated FBS and penicillin–streptomycin (Thermo Fisher Scientific) in a 5% $CO_2$ atmosphere at 37 °C. U2OS TRex cells (provided by S. Blacklow, Brigham and Women's Hospital and Harvard Medical School) and U2OS cell lines (American Type Culture Collection (ATCC)) were used to generate inducible cell lines based on lentiviral infection[22]. The mCherry–eGFP–FAM134B WT and mCherry–eGFP–FAM134B 17K constructs were introduced into the vector pcDNA5/FRT/TO using GATEWAY technology and transfected with the recombinase vector pOG44 into Flp-In U2OS TRex cells. Following selection with 300 µg ml$^{-1}$ hygromycin, the resistant cells were expanded. We generated U2OS cell lines expressing HA–FAM134B or HA–FAM134B 17KR under the control of a doxycycline-inducible promoter. Expression was induced with 0.5 µg ml$^{-1}$ doxycycline for the indicated time. Lentiviruses were produced in HEK 293T cells (ATCC). In brief, HEK 293T (ATCC) cells were co-transfected with the lentiviral plasmids (1.1 µg complementary DNA (cDNA)), containing the cDNA of FAM134B WT or 17KR along with the two packaging vectors pPAX2 (2.2 µg cDNA) and pMD2.G (1 µg cDNA) using Turbofect reagent (Thermo Fisher Scientific). The lentivirus-containing medium was collected after 48 h. The medium was centrifuged to remove dead HEK 293T cells and stored at −80 °C. After 24 h, cells were selected in fresh DMEM containing 3 µg ml$^{-1}$ puromycin. Cells were maintained in the presence of antibiotics for selection and were grown to subconfluence before each experiment. Cells were treated with 0.5 µg ml$^{-1}$ doxycycline (Sigma-Aldrich), 200 ng ml$^{-1}$ bafilomycin A1 (LC Laboratories), 50 µg ml$^{-1}$ cycloheximide (AppliChem PanReac), 250 nM Torin 1 (LC Laboratories) and/or 1 µM or 10 µM TAK243 (MedChemExpress) for the indicated time periods. Transient transfection was carried out using Turbofect reagent. All cell lines were regularly tested for mycoplasma contamination using the LookOut Mycoplasma PCR Detection Kit (Sigma).

Human *AMFR*-targeting siRNAs (siRNA #1 5′-GCA AGG AUC GAU UUG AAU A-3′; and siRNA #2, 5′-GUA AAU ACC GCU UGC UGU G-3′) were purchased from Dharmacon. A non-targeting siRNA was used as a control (Qiagen). Transfection with siRNA was carried out using Lipofectamine RNAiMAX (Thermo Fisher Scientific). Assays were carried out 72 h post-transfection.

### Antibodies

Primary and secondary antibodies are presented in Supplementary Table 1.

### Plasmids

We introduced the cDNAs into vector pDONR223 using the BP Clonase Reaction Kit (Invitrogen) followed by transfer to GATEWAY destination vectors using the LR Clonase Reaction Kit (Invitrogen), resulting in vectors pLTD-N-HA-PURO, pcDNA5-FRT/TO-N-mCherry-eGFP, pcDNA3.1-N-HA, pMH-SFB, pGEX6P1-DEST, pDEST 527 6×His, pDEST-V1ORF, pDEST-V2-ORF, pDEST-ORF-V1 and pDEST-ORF-V2. GST–FAM134B was generated by subcloning into the vector pGEX-6P1 using the EcoRI site. GST–FAM134B 17KR was similarly generated by targeting the EcoRV and SmaI sites of pGEX-6P2.

Plasmids are presented in Supplementary Table 2.

### Ubiquitination assays in cells, co-immunoprecipitation and TUBE-2 pulldown

To assess the ubiquitination status of FAM134, HEK 293T cells transfected with Myc–Ub, HA–FAM134B constructs and WT AMFR–Flag or its catalytically inactive mutant as indicated were lysed (50 mM Tris-HCl pH 7.5, 150 mM NaCl, 0.5 mM ethylenediaminetetraacetic acid (EDTA), 1% Triton X-100, 10 mM *N*-ethylmaleimide (NEM) and Roche EDTA-free protease inhibitors, freshly added). The lysates were incubated for 15 min on ice and centrifuged (12,000*g* at 4 °C for 30 min) before 20 µl of the supernatant was supplemented with Laemmli sample buffer, boiled for 5 min at 95 °C and stored at −20 °C as the input control. Ubiquitinated proteins (Myc–Ub proteins) were immunoprecipitated from soluble extracts with Myc-Trap agarose (Chromotek). Beads were washed three times in lysis buffer and proteins were denatured by heating at 95 °C for 5 min before SDS–PAGE and western blot analysis with anti-HA antibodies for the detection of HA–FAM134B constructs. For other immunoprecipitation assays, cleared lysates were incubated with GFP-Trap (Chromotek), HA-agarose beads (Sigma-Aldrich) or TUBE-2 agarose beads (Life Sensors) at 4 °C overnight. The next day, tubes were centrifuged (800 rcf at 4 °C for 5 min) to sediment the beads, and the supernatant was removed and washed with ice-cold lysis buffer. The inputs and co-precipitation fractions were analysed by SDS–PAGE and western blot.

### Blue-native PAGE, SDS–PAGE and western blot

Blue-native PAGE was performed following the manufacturer's instructions (#BN1001BOX, protocol pub. no. MAN0007893 Rev. A.0, Life Technology). For SDS–PAGE, proteins were denatured at 90 °C for 5 min in Laemmli buffer. Native or denatured proteins were transferred to methanol-activated polyvinylidene fluoride membranes (Amersham Hybond P, 0.45 µm). Membranes were blocked for 1 h in 5% skimmed milk in PBS containing 0.1% Tween-20 and incubated overnight at 4 °C with the specified primary antibody, and for 2 h with the corresponding secondary antibody at room temperature. We used horseradish peroxidase-conjugated anti-rabbit (1:10,000), anti-mouse (1:10,000) and anti-rat (1:10,000) secondary antibodies as appropriate, followed by signal development using Western ECL substrate (sc-2048, SantaCruz) and the Chemidoc automated detection system (Bio-Rad). Densitometric quantification of western blot bands was carried out using ImageJ (version 1.51w).

### Densitometric quantification and statistical analysis

Quantifications of western blot signals were performed using ImageJ software (version 1.51w). For each assay, protein bands were quantified from at least three independent experiments. Data analysis was performed using Microsoft Excel 2016 (Microsoft Corporation) or Prism 9.4.1 (GraphPad Software). Results were graphed as the mean ± standard deviation. Statistical significance was determined by one-tailed, paired *t*-test. $P > 0.05$ or $P \leq 0.05$ were regarded as not statistically significant or statistically significant, respectively. One-way analysis of variance (ANOVA) was used in assays with two independent variables. Bonferroni's multiple comparison test was performed.

### Fluorescence microscopy

Cells were grown on 12-mm glass coverslips or in 96-well plates. Cells were washed with PBS before fixation in 4% paraformaldehyde (PFA) for 10 min at room temperature. After several further washes with PBS, cells were permeabilized with 0.25% saponin in PBS and blocked with 5% FBS for 1 h at room temperature. Primary antibodies were diluted in 5% FBS/0.25% saponin in PBS and incubated overnight at 4 °C. After three PBS washes, secondary antibodies were added and incubated for 1 h at room temperature. The cells were washed another three times before staining with the nuclear dye DAPI for 10 min, and washed again before mounting with Fluoromount-G. The following secondary antibodies were used in a 1:1,000 dilution: anti-rabbit Alexa 488, anti-mouse Alexa 546 and anti-rat Cy5. Slides were imaged using a Leica SP8 confocal microscope fitted with a ×63 oil-immersion lens and analysed with ImageJ software (version 1.51w). For ER fragmentation and ER-phagy flux assays, doxycycline-inducible mCherry–GFP–FAM134B U2OS cells were analysed using a Yokogawa CQ1 (vR1.08.01) confocal imaging cytometer.

### Protein expression and purification

*Escherichia coli* (BL21(DE3)) cells were transformed with constructs encoding GST-tagged FAM134B (WT, 17KR mutant, RHD$_{90–264}$ or

Ub–RHD$_{90-264}$–Ub), GST–Ub and GST, followed by expression and purification as previously described[4,23]. In brief, transformed cells were grown in the presence of appropriate antibiotics overnight and the primary culture was used to inoculate the main culture. When the OD$_{600 nm}$ reached 0.6, 0.25 mM isopropyl β-d-1-thiogalactopyranoside (IPTG) was used to induce protein expression for 16 h at 18 °C, shaking at 180 rpm. After induction, the cells were harvested and the pellets were suspended in ice-cold PBS, followed by sonication and centrifugation (10,000$g$ for 30 min at 4 °C). The supernatant was then fractionated by ultracentrifugation (80,000$g$ for 90 min at 4 °C). RHD fusion proteins were recovered by dissolving the pellets in PBS containing 0.05% dodecyl β-d-maltoside (DDM) and loading onto glutathione-Sepharose TM4 Fast Flow columns (GE Healthcare). The columns were washed with PBS containing 0.05% DDM and the proteins were eluted in PBS containing 15 mM reduced glutathione and 0.025% DDM. GST–Ub and GST were directly loaded from the centrifugation step after sonication and were purified on the GST column. The eluted fractions were concentrated and exchanged with storage buffer (50 mM HEPES pH 7.5, 150 mM NaCl and 0.0075% DDM).

Alternatively, cells expressing GST–FAM134B WT and 17KR were lysed in a French press (G. Heinemann Ultraschall & Labortechnik) to avoid protein denaturation by sonication. Bacterial lysates were mixed with 2% (v/v) Triton X-100, 20 mM MgCl$_2$, 40 µg ml$^{-1}$ DNAse I and 4 mg ml$^{-1}$ lysozyme for 30 min at 1 °C, and cell debris was removed by centrifugation (1,000$g$ for 20 min at 4 °C). The supernatant was incubated with ice-cold PBS, and GST fusion proteins were affinity purified on glutathione-Sepharose resin (Genscript). After washing, bound proteins were eluted by incubation with 20 mM glutathione in 50 mM Tris-HCl pH 8.0 and 120 mM NaCl. The purified proteins were concentrated using Amicon Ultra-4-10k centrifugal filters (Millipore) and dialysed at 4 °C against HN-buffer (20 mM HEPES/KOH pH 7.4, 150 mM NaCl and 2.5 mM DTT) for liposome and freeze-fracture assays.

For the purification of His–RHD$_{90-264}$–Strept-II and His–Ub–RHD$_{90-264}$–Ub–Strept-II, bacterial pellets were resuspended in ice-cold binding buffer (100 mM Tris-HCl pH 8.0, 150 mM NaCl, 1 mM EDTA and 1 mM TCEP) plus a cocktail of protease inhibitors (1 mM PMSF, 1 µM GM6001, 0.25 µM bestatin, 0.5 µM pepstatin and 1 µM E-64), DNAse I (50 µg ml$^{-1}$) and 0.1% DDM. The cells were disrupted by two passes through a microfluidizer at 1,500 bar, and the debris was removed by centrifugation (12,000$g$ for 1 h at 4 °C). The supernatant, including the membranes, was centrifuged (43,000$g$ for 2.5 h at 4 °C) on an Ultracentrifuge Optima L-90K with a 45 Ti rotor (Beckman Coulter). The pelleted membrane was solubilized in membrane extraction buffer (100 mM Tris-HCl pH 8.0, 300 mM NaCl, 1 mM EDTA, 25% glycerol and 2% DDM) at 4 °C with gentle stirring for approximately 2 h. The insoluble fractions were removed by ultracentrifugation (55,000$g$ for 1 h at 4 °C) and the supernatant containing the solubilized membrane proteins was passed through a 0.22-µM filter and supplemented with 40 µg ml$^{-1}$ avidin before purification. The first purification step targeted the strep II tag. A Hiprep strep II 5-ml column (Cytiva) on an Äkta FPLC system was pre-equilibrated with binding buffer supplemented with 0.1% DDM. The sample was allowed to bind to the column and then washed with binding buffer containing 0.01% DDM and lacking EDTA. Fractions in elution buffer (100 mM Tris-HCl pH 8.0, 150 mM NaCl, 0.03% DDM and 3 mM d-desthiobiotin) were analysed by SDS–PAGE and western blot with antibodies specific for the strep II tag. The desired fractions were pooled and supplemented with 25 mM imidazole followed by second-step purification targeting the His$_6$ tag. Talon beads were used and a linear gradient between buffer A (50 mM Tris-HCl pH 7.5, 500 mM NaCl, 25 mM imidazole, 5% glycerol, 0.03% DDM and 1 mM TCEP) and buffer B (buffer A with 250 mM imidazole) was used to purify the protein. The resulting protein was buffer exchanged (25 mM Tris-HCl pH 7.5, 100 mM NaCl, 5% glycerol, 1 mM TCEP and 0.03% DDM) using a centricon filter with a 10-kDa

cut-off. The protein was aliquoted and flash frozen in liquid nitrogen for further experiments.

For AMFR expression in mammalian cells, 3 l cultures of suspension-adapted HEK 293T cells were grown in customized DMEM[24] to a density of $1 \times 10^6$ cells per millilitre. We added 15% (v/v) P3 baculovirus to each flask and incubated on a shaking platform for 24 h, after which the temperature was reduced to 30 °C and 10 mM sodium butyrate was added. Cells were harvested after shaking for 48 h and the pellets were frozen at −80 °C. For AMFR protein purification, approximately 15 g of cells was thawed on ice and resuspended in 60 ml lysis buffer (20 mM HEPES pH 7.5, 300 mM NaCl, 5 mM DTT, Roche protease inhibitor cocktail, 1 mM MgCl$_2$ and 0.001 mg ml$^{-1}$ Benzonase) and sonicated on ice. To solubilize membranes, DDM and CHS was added to the lysate at a final concentration of 1% and 0.1%, respectively, and stirred for 1 h at 4 °C. Insoluble material was removed by centrifugation (40,000$g$ for 30 min at 4 °C). The supernatant was incubated with 2 ml StrepTactin resin (Cytiva) for 2 h at 4 °C to bind to gp78-TEV-TwinStrepII. The resin was poured into a column and washed with 24 column volumes (CV) of wash buffer (20 mM HEPES pH 7.5, 300 mM NaCl, 5 mM DTT and 0.05/0.005% DDM/CHS). Bound protein was eluted with wash buffer containing 5 mM desthiobiotin. To remove the C-terminal affinity tag, eluted protein was incubated for 2 h with 0.1 mg ml$^{-1}$ TEV protease at room temperature. Finally, pooled elution fractions were concentrated to 0.5 ml using 100-kDa cut-off centrifugal concentrators (Amicon) and purified by gel filtration on a Superose 6 10/300 column (Cytiva) running with SEC buffer (20 mM HEPES pH 7.5, 100 mM NaCl, 5 mM DTT and 0.05/0.005% DDM/CHS).

### Liposome preparation, liposome shaping assay and electron microscopy

Liposomes made from synthetic lipids were prepared as previously described[4]. In brief, 2-dioleoyl-*sn*-glycero-3-phosphocholine (DOPC) and 1,2-dioleoyl-*sn*-glycero-3 phosphoethanolamine (DOPE), both from Avanti Polar Lipids, were dissolved in a mixture of chloroform and methanol (4:1) in a round-bottom flask at a molar ratio of 0.8:0.2 (DOPC:DOPE). The organic solvent was removed by rotary evaporation to obtain a uniform dry lipid film, which was then hydrated for 2 h at room temperature with liposome buffer A (50 mM HEPES pH 7.4 and 150 mM NaCl) to obtain a final 15 mg ml$^{-1}$ solution. Liposomes were dissolved by vortexing followed by sonication in an ultrasound bath. Liposomes were equilibrated to 25 °C and extruded using a lipid extruder with 200-nm polycarbonate membranes (Avanti Polar Lipids).

For negative staining assays, 2.5 µM FAM134B WT and FAM134B 17KR mutant or GST were incubated with 1 mg ml$^{-1}$ liposomes in liposome buffer B (50 mM HEPES pH 7.4, 150 mM NaCl and 0.001% DDM) for 18 h at 25 °C on a table-top shaker at 600 rpm. GST–FAM134B RHD$_{90-264}$ or GST–Ub–RHD$_{90-264}$–Ub chimaera and Ub–GST were incubated at 0.5 µM with 1 mg ml$^{-1}$ liposomes in liposome buffer B for 8 h. We then added 5 µl of each sample to the carbon-coated copper grids (SPI Supplies) without glow discharge. After 1 min, the grids were washed twice with water and stained with 1% uranyl formate for 1 min at room temperature. Excess solution was removed by blotting with filter paper. Approximately 20 micrographs were recorded for each sample using a 120 kV Tecnai Spirit Biotwin electron microscope (FEI) equipped with a 4k × 4k CCD detector (US4000-1, Gatan).

Alternatively, membrane shaping by full-length GST–FAM134B ubiquitinated with AMFR was investigated by transmission electron microscopy using freeze-fractured liposomes prepared from Folch-fraction type I lipids (Sigma-Aldrich) as previously described[25]. We incubated 1 mg of liposomes with 2.5 µM protein in HN-buffer (20 mM HEPES/KOH pH 7.4, 150 mM NaCl and 2.5 mM DTT) containing 0.3 M sucrose for 15 min at 37 °C. Subsequently, 15 µg proteinase K was added and incubated for 40 min at 45 °C to avoid liposomal aggregates[26]. Small aliquots (1–2 µl) of the liposome suspension were

then freeze-fractured[26,27]. The samples were examined by systematic grid exploration using an EM 900 electron microscope (Zeiss) at 80 kV. Images were acquired using a wide-angle dual-speed 2K CCD camera (Tröndle). Diameters of liposomes were determined using ImageJ software (version 1.51w).

## In vitro ubiquitination of FAM134B using recombinant AMFR

AMFR-mediated ubiquitination assays were based on a modified in vitro ubiquitination assay (Abcam). In brief, purified substrate (5 μM full-length GST–FAM134B, 1 μM His–RHD$_{90-264}$–Strept-II or 1 μM His–Ub–RHD$_{90-264}$–Ub–Strept-II) was mixed with 10 μM Ub, 10 mM ATP and 10 mM MgCl$_2$ in 50 mM Tris-HCl pH 7.5, 150 mM NaCl, 0.8 μM E3-ligase AMFR, 100 nM E1 UBA1 and 0.8 μM E2 of AMFR UBE2G2 (Biotechne) for 2 h at 37 °C. The reaction mixture was analysed by SDS–PAGE or western blotting with antibodies against GST, His$_6$ or Ub(P4D1). Alternatively, the samples were prepared for MS. Samples were incubated with SDC buffer (1% sodium deoxycholate, 0.5 mM TCEP, 2 mM chloroacetamide and 50 mM Tris-HCl pH 8.5) and heated to 60 °C for 30 min. We then added 500 ng trypsin to each sample and incubated overnight at 37 °C. The reaction was stopped with 1% TFA in isopropanol. Peptides were cleaned up using SDB-RPS stage tips (Sigma-Aldrich). After one wash with 1% TFA in isopropanol and one wash with 0.2% TFA in water, peptides were eluted in 80% acetonitrile plus 1.25% ammonia. Eluted peptides were dried and processed for LC–MS.

## BiCAP interactome analysis and sample preparation for MS

HEK 293T cells were transiently co-transfected with the constructs V1–FAM134B WT and V2–FAM134B WT, V1–FAM134C WT and V2–FAM134C WT, V1–FAM134B WT and V2–FAM134C or V1–FAM134B 17KR and V2–FAM134B 17KR. After 16 h, cells were lysed with 1% Triton X-100 in lysis buffer (50 mM Tris-HCl pH 7.5, 150 mM NaCl, 0.5 mM EDTA, Roche EDTA-free protease inhibitor cocktail and NEM), followed by incubation with GFP-trap beads (Chromotek) overnight at 4 °C on a rotating platform. Protein-bound beads were washed three times with lysis buffer and three times in the same buffer without detergents before on-bead trypsin digestion. Samples were incubated with 25 μl SDC buffer (2% sodium deoxycholate, 1 mM TCEP, 4 mM chloroacetamide and 50 mM Tris-HCl pH 8.5) and heated to 60 °C for 30 min. We then added 500 ng trypsin in 25 μl 50 mM Tris-HCl (pH 8.5) to each sample and incubated overnight at 37 °C. The reaction was stopped with 150 μl of 1% TFA in isopropanol. Peptides were cleaned up using SDB-RPS stage tips (Sigma-Aldrich). After one wash with 1% TFA in isopropanol and one wash with 0.2% TFA in water, peptides were eluted in 80% acetonitrile plus 1.25% ammonia. Eluted peptides were dried and processed for LC–MS.

## LC–MS analysis

Dried peptides were reconstituted in 2% acetonitrile containing 0.1% TFA and analysed on a QExactive HF mass spectrometer coupled to an easy nLC 1200 (Thermo Fisher Scientific) fitted with a 35-cm, 75-μm ID fused-silica column packed in house with 1.9-μm C18 particles (Reprosil pur, Dr. Maisch). The column was maintained at 50 °C using an integrated column oven (Sonation). Peptides were eluted in a non-linear gradient of 4–28% acetonitrile over 45 min and directly sprayed into the mass spectrometer equipped with a nanoFlex ion source (Thermo Fisher Scientific). Full-scan MS spectra (300–1,650 $m/z$) were acquired in profile mode at a resolution of 60,000 at $m/z$ 200, a maximum injection time of 20 ms and an AGC (automatic gain control) target value of $3 \times 10^6$. Up to 15 of the most intense peptides per full scan were isolated using a 1.4-Th window for fragmentation by higher energy collisional dissociation (normalized collision energy of 28). MS/MS spectra were acquired in centroid mode with a resolution of 30,000, a maximum injection time of 45 ms and an AGC target value of $1 \times 10^5$. Single charged ions, ions with a charge state of more than four and ions with unassigned charge states were not considered for fragmentation, and dynamic exclusion was set to 20 s to minimize the acquisition of fragment spectra representing already acquired precursors.

## MS data processing

MS raw data were processed using MaxQuant v1.6.17.0 with default parameters. Acquired spectra were searched against the human 'one sequence per gene' database (Taxonomy ID 9606) downloaded from UniProt (12 March 2020; 20,531 sequences), and a collection of 244 common contaminants ('contaminants.fasta' provided with MaxQuant v1.6.17.0) using the Andromeda search engine integrated into MaxQuant v1.6.17.0 (ref. 28,29). Identifications were filtered to obtain false discovery rates below 1% for both peptide spectrum matches (minimum length of seven amino acids) and proteins using a target–decoy strategy[30]. Protein quantification and data normalization relied on the MaxLFQ algorithm implemented in MaxQuant v1.6.17.0 (ref. 31). The MS proteomics data have been deposited to the ProteomeXchange Consortium[32] via the PRIDE partner repository[33] with the dataset identifiers PXD032721 (Fig. 1a,b), PXD032740 (Extended Data Fig. 8b), PXD032741 (Fig. 4d and Extended Data Fig. 8c–e), PXD032743 (Fig. 4e), PXD032750 (Extended Data Fig. 9h–k), PXD039186 (Extended Data Fig. 10m,n), PXD039187 (Fig. 5c) and PXD039188 (Extended Data Fig. 10p,q). For protein assignment, spectra were correlated with the UniProt human database v2019 including a list of common contaminants. Searches were performed with tryptic specifications and default settings for mass tolerances in MS and MS/MS spectra. Carbamidomethyl cysteine, methionine oxidation and N-terminal acetylation were defined as fixed modifications. The match-between-run feature was used with a time window of 1 min. For further analysis, Perseus v2.0.7.0 was used and first filtered for contaminants and reverse entries as well as proteins that were only identified by a modified peptide.

## Structural modelling of ubiquitinated FAM134 proteins

The previously built molecular model of FAM134B RHD[4] was extended to include an additional ten residues at the C terminus of the RHD (residues 261–270). Isopeptide bonds between the lysine (K160 and K264) and the terminal glycine of Ub (G76) were modelled by modifying the side-chain lysine bead (SC2/+1) into the neutral backbone bead (BB/0) and restraining its distance to the terminal bead of Ub to 0.35 nm with a force constant $k = 1,250$ kJ mol$^{-1}$. Two mono-ubiquitinated and one bi-mono-ubiquitinated RHD structures were modelled (K160–Ub, K264–Ub and (K160 + K264)-Ub, respectively).

## MD simulations and analysis

Coarse-grained (CG) MD simulations were prepared using the MARTINI model (v2.2)[34,35]. Initial CG structures were built using martinize.py[36]. Assignments from DSSP (Dictionary of Secondary Structures in Proteins) program were used to generate backbone restraints to preserve local secondary structure[37,38]. CG models were embedded into POPC (16:0–18:1 phosphatidylcholine (PC)) bilayers spanning the periodic simulation box in the $x-y$ plane. Initial configurations for each system were assembled and then solvated with CG water containing 150 mM NaCl using insane.py[36]. Each system was energy minimized and equilibrated using the Berendsen thermostat[39] and barostat[40] along with position restraints on protein backbone beads, followed by production runs with a 20-fs time step. System temperature and pressure during the production phase were maintained at 310 K (unless otherwise stated) and 1 atm with the velocity rescaling thermostat[41] and the semi-isotropic Parrinello–Rahman[42] barostat, respectively. All simulations were performed using gromacs (v2019.3)[43,44]. Long-lived, highly populated RHD conformations were obtained after clustering evenly sampled conformations ($n = 10,000$) from each trajectory. Clusters were obtained using backbone root-mean-square deviation (cut-off = 0.8 nm) by using the gromos method[45], as implemented in the gmx_cluster tool.

## Curvature induction by ubiquitinated RHDs: bicelle-to-vesicle transitions and kinetics

Discontinuous bicelle systems containing saturated DMPC (14:0 PC) and DHPC (7:0 PC) lipids were assembled as previously described[4,5]. The equilibrated ubiquitinated and non-ubiquitinated RHD molecules obtained from simulations in the POPC bilayers (after 5 µs) were then embedded in the bicelle and solvated. One hundred replicates for each system were simulated with different initial velocities at 300 K to obtain statistics on the transition times to vesicles. Shape transformations from flat bicelles ($H = 0$ nm$^{-1}$) to curved vesicles ($H = 0.15$ nm$^{-1}$) were monitored by measuring the signed membrane curvature ($H(t)$). Lipid coordinates were fitted to spherical surfaces using least squares optimization to compute membrane curvature along simulations using MemCurv[4]. Curvature away from and towards the upper/cytoplasmic leaflet are reported as positive and negative values, respectively. The statistics of waiting times ($t$) for the formation of vesicles (bilayer curvature, $|H| > 0.15$ nm$^{-1}$) for the three systems were determined from individual replicates. We modelled the kinetics of the bilayer-to-vesicle transition using a Poisson process with a lag time ($t = t' + \tau$). The time $t' = 1/k'$ describes the Poisson process with rate $k'$. The constant lag time $\tau$ captures the time required for vesicle closure from the curved bilayer disc. The distributions of waiting times are thus $p(t) = k'\,e^{-k'(t-\tau)}$ for $t > \tau$. We determined the rate of vesicle formation ($k = 1/(t' + \tau)$) for different systems, from fitting the cumulative distribution function for the probability density, $P(t - \tau) = ke^{-k(t-\tau)}$ corresponding to $p(t)$, to the observed waiting time distributions estimated from replicates. We used the previously computed maximum likelihood estimates of the vesiculation rate for non-ubiquitinated RHD bicelles[4] ($k_{RHD} = 0.0018$ ns$^{-1}$) to compute the acceleration factors for each system (acc $= k_{sys}/k_{RHD}$). Furthermore, to show the effect of temperature on the energy barrier to form closed vesicles from flat bicelles, we also simulated 20 replicates of each system at 280 K and estimated the number of successful vesicle closure events within the simulation timescale.

## Curvature sensing by ubiquitinated RHDs on buckled membranes

A CG POPC bilayer was used to tile a tessellated buckled surface using LipidWrapper[46]. The buckled membrane was solvated with CG water and ions and equilibrated in a periodic box with a fixed $x$–$y$ plane ($57 \times 28$ nm$^2$) and excess membrane area (approximately 17 nm$^2$). This preserved the buckled shape of the bilayer, offering a range of curvature values to be sampled by embedded proteins ($H(x,y) = -0.05 \le 0 \le 0.05$ nm$^{-1}$). Ubiquitinated and non-ubiquitinated RHDs (in separate simulations) were initially embedded in regions with small local curvature ($H(x,y) \simeq 0$). Following another equilibration phase with proteins, the systems were simulated for more than 20 µs at 310 K. The membrane profiles of the buckled surface with and without embedded proteins were analysed by using the Monge representation to compute local principal curvatures $k_1(x,y)$ and $k_2(x,y)$, the mean curvature $H(x,y)$, and the Gaussian curvature $K_G(x,y)$ as implemented in MemCurv[4].

## RHD cluster formation: dimerization, tubule deformation and membrane budding

The clustering of ubiquitinated RHD molecules was simulated under periodic boundary conditions using buckled, tubular and flat membrane structures. MD simulations were initiated after embedding one K160–Ub RHD and one K264–Ub RHD into a single membrane buckle. Initially, the two proteins were placed more than 20 nm apart. The two proteins were tracked by measuring their minimum distance to identify initial contact and cluster formation. Once the initial contacts were made and dimers were formed, residue pairwise interactions were mapped across the tethered Ub moieties (K160-Ub and K264-Ub) to identify specific Ub–Ub *trans*-contacts. We then obtained the configuration of a closed membrane tubule (length of approximately 97–100 nm; diameter of approximately 12–15 nm) used in our previous work[4]. Ten ubiquitinated

RHD molecules were embedded along the length of the tubule such that the individual proteins were spaced maximally away from each other. The RHD-containing tubule was equilibrated in explicit solvent (approximately $3.6 \times 10^6$ beads) under NVT (constant volume, constant temperature ensemble) and NPT (constant pressure, constant temperature ensemble) conditions along with position restraints on protein backbone beads. A production run of 10 µs was carried out, after releasing position restraints to observe curvature-mediated protein sorting and the formation of RHD clusters. We also generated initial configurations of nine (K160 + K264)–Ub-RHDs on a $3 \times 3$ square grid embedded in POPC bilayers such that each protein was separated by its nearest neighbour by 10 nm. We changed the lipid-number bilayer asymmetry from $\Delta N = 0$ to $\Delta N = 300$ using the insane.py script. Following a previously implemented method[47], we scaled the protein–protein LJ pair interaction well depth, $\epsilon_\alpha = \epsilon_0 + \alpha(\epsilon_{original} - \epsilon_0)$. A value of $\alpha = 0.65$ corresponds to a reduction in PPI strength in the MARTINI model and a value of $\alpha = 1.0$ recovers the full interaction in the MARTINI forcefield, $\epsilon_1 = \epsilon_{original}$. The resulting $2 \times 2 = 4$ bilayer configurations were then solvated with CG water containing 150 mM NaCl. We embedded the ubiquitinated and non-ubiquitinated RHD proteins in the asymmetric membranes ($30 \times 30 \times 20$ nm$^3$) in a square grid and energy minimized the systems using a soft-core potential and steepest-decent algorithm to remove steric clashes with lipids. Production runs for 5 µs at NPT conditions were carried out for each simulation condition, in which position restraints were released to observe curvature-mediated protein sorting effects and Ub–Ub interactions, leading to the formation of RHD clusters. We monitored the box dimensions over time, which acted as proxies to indicate spontaneous budding. In addition, we tracked the $z$-coordinates of the centre-of-mass (COM) of all nine proteins along with the lowest and the highest points of the PO4 beads comprising the POPC bilayer. We also computed the distance matrix specifying all inter-RHD distances computed from COM positions and used hierarchical clustering with single linkage and a cut-off of 10 nm to obtain the distinct protein clusters in each frame of the simulation. We quantified the number of clusters and their sizes to identify the largest cluster for each simulation frame.

## Antibody–oligonucleotide conjugation for DNA-PAINT

For exchange DNA-PAINT experiments, donkey anti-rabbit antibodies (#711-005-152, AffiniPure) and goat anti-mouse antibodies (#115-005-003, AffiniPure) were covalently labelled with the short DNA-docking strands anti-R2 (5′-ACCACCACCACCACCACCA-3′) and anti-R1 (5′-TCCTCCTCCTCCTCCTCCT-3′), respectively, using DBCO-sulfo-NHS ester chemistry[48]. In brief, concentrated secondary antibodies were incubated with a 20-fold molar excess of DBCO-sulfo-NHS ester (Jena Bioscience). After removing excess reagent, azide-functionalized DNA-docking strands were added at a tenfold molar excess and incubated overnight at 4 °C. Unbound DNA was removed from the samples concomitant with storage buffer exchange (PBS) using Amicon centrifugal filters (100 kDa cut-off). Antibody–DNA complexes were concentrated to 5 mg ml$^{-1}$ and stored at 4 °C.

## Exchange DNA-PAINT sample preparation

For the exchange DNA-PAINT experiments, U2OS cells were seeded in Ibidi µ-slide VI chambers at 70% confluency. The cells were fixed for 30 min with pre-warmed (37 °C) 4% methanol-free formaldehyde (Sigma-Aldrich) in PBS followed by three washes with PBS. Fixed cells were then incubated in permeabilization/blocking buffer (10% FBS and 0.1% saponin) containing primary antibodies for 60 min at room temperature: rabbit anti-FAM134B (Genscript), mouse anti-LC3B or mouse anti-REEP5, each diluted 1:200. Excess primary antibodies were removed from the chambers by three washes with PBS. Cells were then incubated with custom DNA-labelled secondary antibodies in the permeabilization/blocking buffer for 60 min, followed by the removal of free antibodies by three washes with PBS. Finally, samples were

post-fixed with 4% methanol-free formaldehyde in PBS for 10 min at room temperature, followed by three washes with PBS. For the exchange DNA-PAINT experiments, 125-nm gold beads (Nanopartz) were used as fiducial markers. The gold beads were sonicated for 10 min, diluted 1:30 in PBS and sonicated again for 10 min. We added 100 µl of the gold bead solution to each Ibidi µ-slide VI chamber, settled for 10 min and washed three times with PBS. The cell chambers were finally connected to a microfluidic device (Bruker). Before imaging, ATTO-655-labelled imager strands R1 or R2 (50 pM, 0.5 M NaCl and PBS, pH 8.3) were injected into the flow chamber at a flow rate of 200 µl per minute. In the exchange DNA-PAINT experiments, imager strands were exchanged between sequential imaging cycles by washing the samples with PBS and injecting new imager strands under equal conditions.

## DNA-PAINT microscopy setup and data acquisition

Exchange DNA-PAINT data were captured as previously described[49] using the N-STORM super-resolution microscopy system (Nikon) equipped with a ×100 oil immersion objective (Apo, NA 1.49) and an EMCCD camera (Andor Technology). ATTO-655-conjugated oligonucleotides were excited with a collimated 647-nm laser beam (at an intensity of 1.1 kW cm$^{-2}$ measured at the objective) in highly inclined and laminated optical sheet mode. We acquired 20,000 consecutive frames at 10 Hz in active frame transfer mode with an EMCCD gain of 200, a pre-amp gain of 1 and at an effective pixel size of 158 nm. For astigmatism-based 3D exchange DNA-PAINT experiments, a customized cylindrical lens (RCX-39.0.38.0-5000.0-C-425-675, 10-m focal length, CVI Laser Optics) was inserted into the emission light path. NIS Elements (Nikon), LCControl (Agilent) and Micro-Manager (v1.4.22)[50] were used for optical setup control and data acquisition. FAM134B, LC3B-II and REEP5 were imaged sequentially following the microfluidic-assisted exchange of R2 and R1 imager strands.

## DNA-PAINT image processing

Single-molecule localization and image reconstruction were carried out using Picasso (v0.2.8)[10]. Single-molecule localization was achieved by integrated Gaussian maximum likelihood estimation with the following parameters: minimum net gradient = 40,000, baseline = 205, sensitivity = 4.78 and quantum efficiency = 0.95. Fiducial gold bead markers were used for post-imaging drift correction and alignment of FAM134B, LC3B-II and REEP5 channels. Single-molecule point-spread functions were filtered based on the ATTO-655 single-molecule footprint (point-spread function (PSF) symmetry 0.7 < full width at half maximum (FWHM)$(x)$/FWHM$(y)$ < 1.4, intensity threshold and localization precision < 50 nm). Signals from the same origin were linked within a radius of five times the nearest neighbour based analysis (NeNa)[51] localization precision, and with a maximum dark time of eight consecutive frames. Signals arising from the fiducial marker that passed the filtering process were removed by excluding traces from the same origin with a length exceeding 20 consecutive frames. For exchange 3D DNA-PAINT experiments, a 3D calibration curve was recorded using z-stacks (step size = 50 nm) of surface-immobilized 0.1-µm TetraSpeck Microspheres (Thermo Fisher Scientific). The axial section was limited to 800–1,000 nm, depending on the robustness of the fit. Multi-channel single-molecule 3D localizations from exchange DNA-PAINT experiments were aligned in Picasso and visualized in ViSP (v1.0)[52]. The 3D movie (Supplementary Video 7) was generated using ViSP v1.0 open-source software[53].

## Clusters and statistical analysis

FAM134B and LC3B-II nanoclusters were identified in DNA-PAINT images using the density-based spatial clustering and application with noise (DBSCAN) algorithm with a radius of 31 nm and a minimum density of ten localizations. Cluster diameters were calculated for each condition from cluster areas. Microscale FAM134B clusters were segmented from nanoscale ER-phagy initiation sites using SR-Tesseler (v1.0.0.1)[48]. Voronoi tessellation was computed by calculating the first rank order local density map from single-molecule localizations. Thresholds for cluster segmentation were determined previously for each cell using Picasso (density factor 2 > δ < 6, localizations $n = 800$ and minimum diameter = 100 nm). HA–FAM134B nanocluster diameters and microcluster areas were tested for normal distribution using a Shapiro–Wilk normality test, and statistical significance was determined using a non-parametric Mann–Whitney $U$-test. The relative frequency distribution of nanocluster diameters was fitted with a log-normal distribution. The mode of the log-normal distribution is a measure for the cluster diameter ($d$) and was calculated as $d = e^{(\mu - \sigma^2)}$. OriginPro 2020 v9.7 (Origin Lab) was used for statistical analysis.

## Quantitative analysis of FAM134B copy numbers in nanoscale clusters

For the quantification of FAM134B copy numbers in nanoscale clusters, the mean dark time of binding events was analysed from intensity–time traces as previously described[13]. In brief, FAM134B clusters that colocalized with LC3B-II were selected manually from DNA-PAINT images and the mean dark time of binding events was determined from the plot of cumulative histograms using Picasso built-in functions. As a calibration, we used extracellular primary–secondary antibody complexes adhering to the sample coating. The inverse of dark times, also known as the qPAINT index[54], is proportional to the number of docking strands in clusters and was used for protein copy number determination. We fitted the relative frequency distribution of inverse dark times with a log-normal distribution and determined the mode of distribution.

## Statistical analysis

All experiments were performed in at least three independent experiments if not indicated otherwise. Data are presented as mean ± standard error of mean if not indicated otherwise. For statistical analysis, raw data were analysed for normal distribution with the Kolmogorov–Smirnov test or with graphical analysis using the Q–Q plot. If appropriate, we either used one-way ANOVA (with Bonferroni post-hoc test if not indicated otherwise), repeated-measures two-way ANOVA, Kruskal–Wallis $H$-test, Student's $t$-test (one-tailed or two-tailed) or the Mann–Whitney $U$-test. $P < 0.05$ were considered significant.

## Reporting summary

Further information on research design is available in the Nature Portfolio Reporting Summary linked to this article.

## Data availability

The proteomics data are deposited in the ProteomeXchange Consortium via the PRIDE partner repository with the dataset identifiers: ubiquitination promotes FAM134B-mediated ER-phagy (PXD032721), FAM134B homodimer ubiquitination (PXD032740), FAM134B oligomer ubiquitination and ER-phagy (PXD032741), binding partners of FAM134B WT and 17KR oligomers (PXD032743), E3 ligase regulates FAM134B ubiquitination (PXD032750), in vitro FAM134B ubiquitination (PXD039186), in vivo FAM134B ubiquitination in *AMFR*-knockdown cells (PXD039187) and in vitro FAM134B RHD and Ub–RHD–Ub ubiquitination (PXD039188); MD simulation trajectory files and corresponding parameter files are large and span long microsecond timescales and multiple replicates can only be shared upon specific requests. All the data analysis of this study is in the Supplementary information. Source data for gels and blots are provided as Supplementary information. Source data are provided with this paper.

## Code availability

The analysis codes for bicelle-to-vesicle simulations and curvature computations on simulations with buckled membranes are provided in the GitHub repository (https://github.com/bio-phys/MemCurv).

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

**Acknowledgements** We thank A.-C. Jacomin, A. Gubas and D. Höller for critical reading of the manuscript; all the members of the I.D. laboratory for their support and constructive discussion; the q-Proteomics platform at the Institute of Biochemistry II (Frankfurt), G. Tascher, T. Colby and I. Matic for the proteomics support; and A. Kreusch, M. Öhler, S. Berr and K. Gluth in the B.Q. and M.M.K. laboratories for their practical help with cloning, protein purification, liposome generation and liposome shaping assays. We thank the Central Electron Microscopy facility of the Max Planck Institute of Biophysics for technical support. G.H. thanks the Max Planck Society for support. R.M.B. and G.H. thank the Center for Supercomputing, GUF, and the MPCDF, Garching for computing resources. This research is supported by the Deutsche Forschungsgemeinschaft (DFG; German Research Foundation), project numbers 259130777-SFB 1177 to I.D., V.D., M.H. and G.H., and QU116/9-1 to B.Q., KE685/7-1 to M.M.K. and HU800/15-1 to C.A.H., the European Union (ERC, ER-REMODEL, 101055213), the HMWK-funded cluster grant ENABLE, the Leistungszentrum Innovative Therapeutics (TheraNova) funded by the Fraunhofer Society and the Hessian Ministry of Science and Art, and grants from Else Kroener Fresenius Stiftung and Dr. Rolf M. SchwieteStiftung to I.D. S.K.K. is supported by a postdoctoral fellowship from EMBO (ALTF 199-2021).

**Author contributions** The present study was conceived by I.D., A.G. and A.C.-P. The experiments and analysis were performed by A.G., A.C.-P., R.M.B., M.G., S.K.K., A.X. and E.S. A.G., A.C.-P. and E.S. prepared all the constructs. A.G. and A.C.-P. carried out all the cell biology and biochemical experiments, generated all the inducible cell lines and samples for MS experiments and analysed the results, including MS data. M.E.H. performed the initial analysis of MS data. R.M.B. performed structural modelling, MD simulations and analysis with the support of G.H. M.G. and A.B. carried out the super-resolution experiments and analysis (under the supervision of M.H.). Protein purification was carried out by M.M., B.B. (under the supervision of B.A.S.), S.K.K. and E.S. Membrane-shaping assays and data analysis were carried out by S.K.K., A.X., A.C.-P., A.G., E.S., M.M.K., B.Q. and C.A.H. A.G., A.C.-P., R.M.B. and I.D. wrote the manuscript with contributions from all authors.

**Competing interests** The authors declare no competing interests.

**Additional information**
**Correspondence and requests for materials** should be addressed to Ivan Dikić.

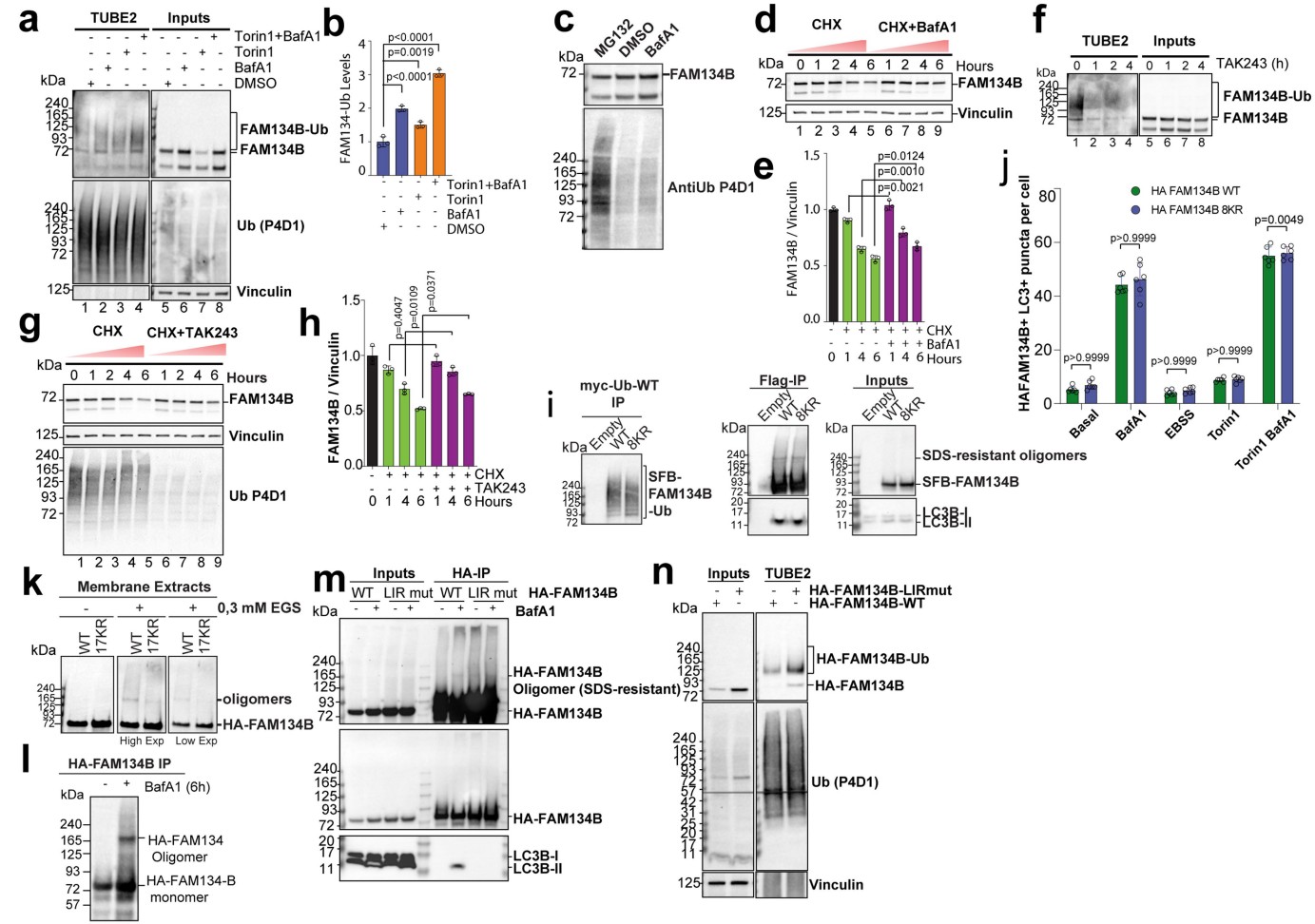

**Extended Data Fig. 1 | Constitutive and inducible FAM134B-RHD ubiquitination regulates ER-phagy. a**, TUBE-2 pulldown assay showing increased ubiquitination of endogenous FAM134B following Torin 1 treatment. Cells were treated with DMSO (control), 200 nM BafA1 for 6 h, 250 nM Torin 1 for 6 h, or a combination of 250 nM Torin 1 and 200 nM BafA1 for 6 h. Protein samples were analysed by SDS-PAGE and western blotting, as indicated. **b**, Densitometric quantification of ubiquitinated FAM134B (FAM134B-Ub) in panel **a** (Data are mean ± s.d.; $n = 3$ independent experiments, one-way ANOVA, Bonferroni post-hoc test). **c**, Cells were treated with DMSO (control), 200 nM BafA1 for 6 h or 10 µM MG132 for 6 h. Detergent-soluble extracts were analysed by western blot with antibodies against FAM134B and UbP4D1. The panels show representative immunoblots. **d**, Cycloheximide (50 µg/ml) chase for 0–6 h in HeLa cells without or with 200 nM BafA1. Detergent-soluble extracts were analysed by western blot with antibodies against FAM134B and vinculin. **e**, Densitometric quantification of FAM134B (normalised to vinculin) in panel **d** (Data are mean ± s.d.; $n = 3$ independent experiments, one-way ANOVA, Bonferroni post-hoc test). **f**, Cells were treated with DMSO (control) or 10 µM TAK243 for 1, 2 and 4 h before TUBE-2 pulldown assays. Endogenous ubiquitination of FAM134B was detected by western blot ($n = 1$ experiment). **g**, Cycloheximide (50 µg/ml) chase for 0–6 h in HeLa cells with or without 10 µM TAK243. Detergent-soluble extracts were analysed by western blot with antibodies against FAM134B, UbP4D1 and vinculin. **h**, Densitometric

quantification of FAM134B (normalised to vinculin) in panel **g** (Data are mean ± s.d.; $n = 3$ independent experiments; One-way ANOVA, Tuckey's post-hoc test). **i**, Ubiquitination assay in cells showing FAM134B-RHD ubiquitination despite the replacement of eight conserved lysine residues with arginine (myc-Ub-IP). LC3B-II binding to FAM134B and the formation of high-molecular-weight species (oligomers, SDS-resistant) of FAM134B were not affected by 8KR (Flag FAM134B-IP) ($n = 1$ experiment). **j**, Co-localisation of HA-FAM134B/LC3B⁺ puncta per cell in cells expressing FAM134B-WT or 8KR. Number of cells per condition for HA-FAM134B WT: 2039 (DMSO), 2366 (BafA1), 1515 (EBSS), 2280 (Torin1), 1987 (Torin1+BafA1). Number of cells per condition for HA-FAM134B-8KR: 2416 (DMSO), 2494 (BafA1), 1753 (EBSS), 2652 (Torin1), 2580 (Torin1+BafA1). Bars represent means ± s.d. ($n = 6$ independent experiments; One-way ANOVA, Bonferroni post-hoc test). **k**, Isolated membranes of cells expressing HA-FAM134B WT or 17KR were treated with 0.3 mM EGS and analysed by SDS-PAGE. Oligomers were visualised by western blot. ($n = 1$ experiment). **l**, U2OS cells stably expressing HA-FAM134B-WT under the control of a doxycycline-inducible promoter were treated with DMSO (control) or 200 nM BafA1 for 6 h. **m**, HeLa cells stably expressing HA-FAM134B-WT or the LIR mutant under the control of a doxycycline inducible promoter were treated with DMSO (control) or 200 nM BafA1 for 6 h. ($n = 1$ experiment). **n**, TUBE-2 pulldown assay showing the accumulation of endogenous ubiquitinated HA-FAM134B LIR-mutant. The panels show representative immunoblots.

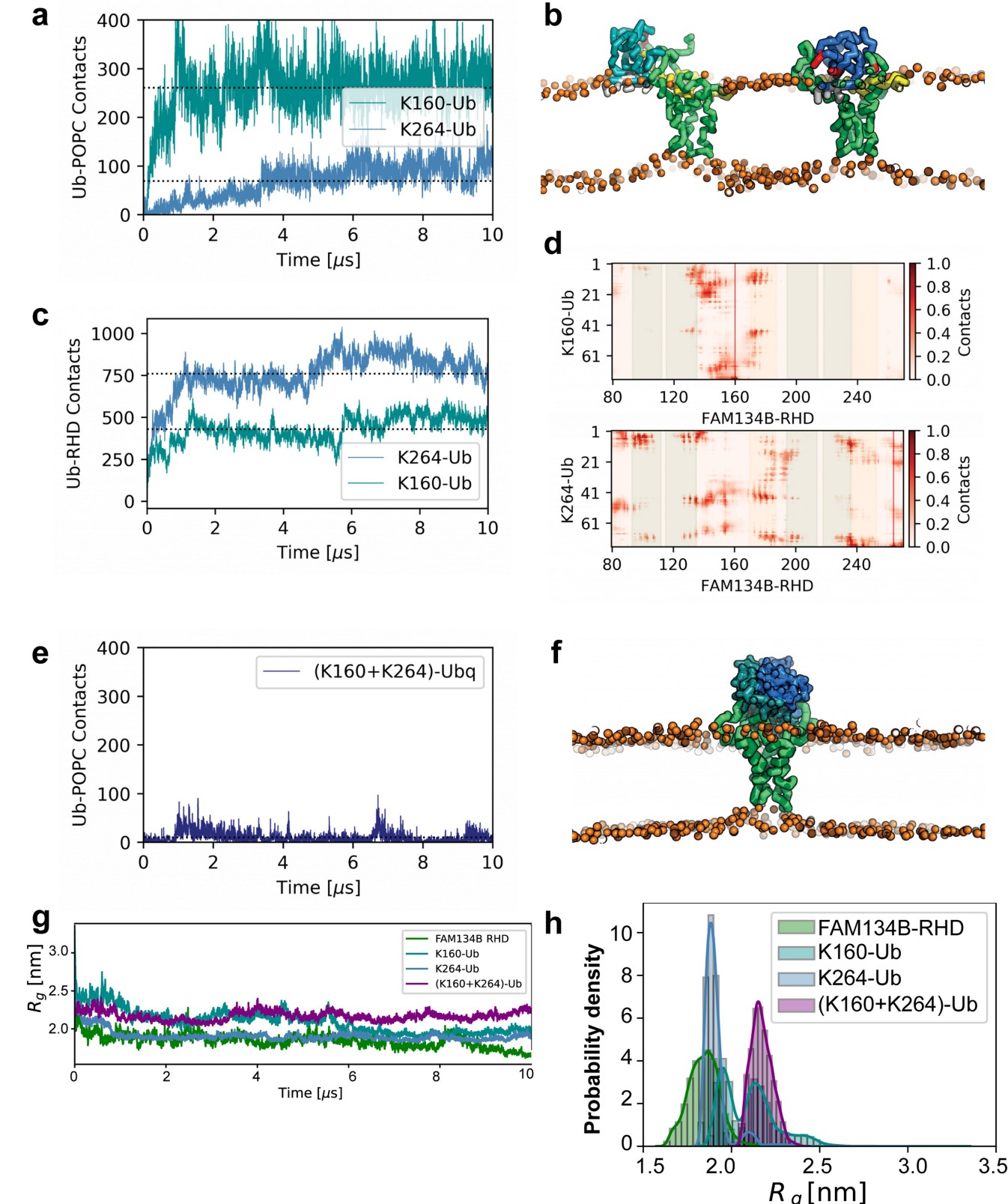

**Extended Data Fig. 2** | See next page for caption.

**Extended Data Fig. 2 | Ubiquitin-membrane and ubiquitin-RHD interactions. a**, Time series of the number of contacts between CG-beads of Ubiquitin and POPC bilayers in MD simulations of mono-ubiquitinated RHDs. **b**, Snapshots from MD simulations of K160-Ub (left) and K264-Ub (right) showing the structures of the Ub moieties (in cyan and blue), and their interactions with the membrane (orange beads). Modified lysines (red) show the relative orientations of the Ub-moieties with respect to the amphipathic helices (yellow) and the cytosolic portion of the RHD. **c**, Time series of the number of contacts between CG beads of RHD and covalently linked Ub moieties. **d**, Residue-wise contact maps between Ub and RHDs. The solid red lines indicate the position of the lysine residue on the RHD linked to Ub. **e**, Time series of the number of contacts between CG beads of Ubiquitin and POPC bilayers in MD simulations of bi-mono-ubiquitinated RHD, (K160+K264)-Ub. **f**, Snapshot from MD simulation of (K160 + K264)-Ub variant showing how bi-mono-ubiquitination of the RHD reduces the interactions between ubiquitin and POPC bilayer. The two Ub-moieties are involved in intra-molecular/cis interactions on top of the cytosolic face of the RHD. **g**,**h** Time series (**g**) and probability distribution (**h**) of the radii of gyration, $R_g$, sampled by ubiquitinated and non-ubiquitinated FAM134B-RHDs during the 10-μs MD simulations.

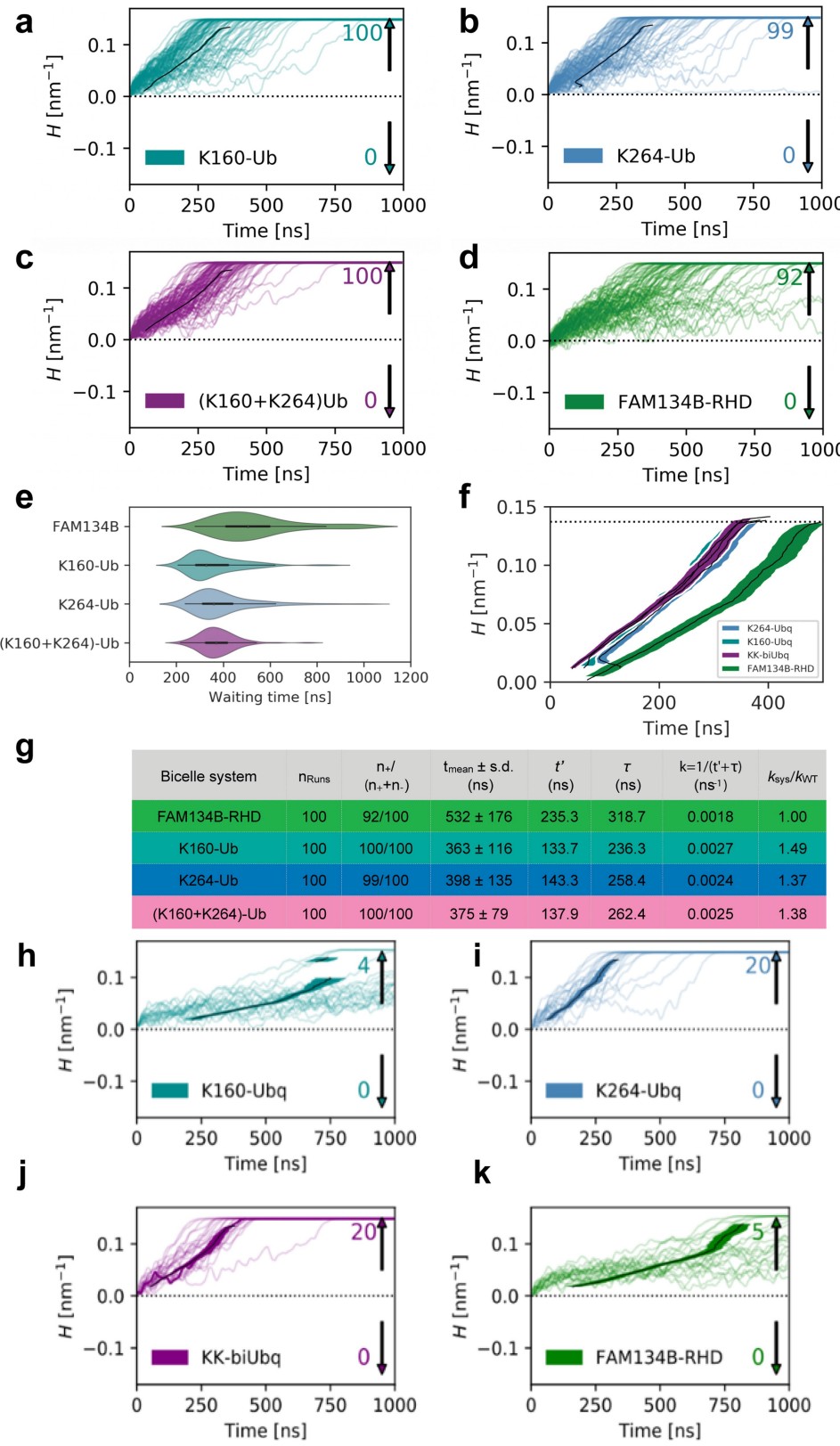

**Extended Data Fig. 3** | See next page for caption.

**Extended Data Fig. 3 | Ubiquitinated RHDs induce membrane curvature.**
**a**–**d**, Time series of mean curvature ($H$) of bicelle containaing single FAM134B RHD. One hundred MD simulations for each system were initiated with different initial velocities to study curvature induction (positive/negative) and transition of the bicelles into closed vesicles ($H$ = +0.14 nm$^{-1}$) at 300 K. **e**, Violin plots of estimated waiting times required for bicelle-to-vesicle transitions induced by WT and Ub-variants of FAM134B-RHD. n = 92 (FAM134B-RHD), 100 (K160-Ub), 99 (K264-Ub), 100 (K160+K264-Ub) runs **f**, Comparison of mean ± SEM curvature time traces (black line ± shaded region) for each system shows that Ub-RHDs induce bicelles to curve swiftly, resulting in faster kinetics. **g**, Table showing the kinetics of the *in silico* curvature induction process. The numbers $n_+$ and $n_-$ indicate bicelle transitions resulting in bilayer curvature away from and towards the upper/cytoplasmic leaflet. Observed waiting times ($t$) for vesicle formation were recorded as the time taken for the bilayer or bicelle discs to reach a curvature of $|H|$ = 0.14 nm$^{-1}$. The vesicle formation rate $k_{sys}$ = 1/($t'$ + $\tau$) was estimated from exponential fits of the cumulative distribution functions of Poisson-distributed waiting times ($t'$) and a constant lag time ($\tau$). The acceleration in vesicle formation due to a protein inclusion was estimated as the ratio $k_{sys}/k_{WT}$. **h**–**k**, Time series of bicelle mean curvature ($H$) for 20 simulations initiated at 280 K for each system. Fewer bicelles with embedded K160-Ub and WT-RHD proteins (4/20 and 5/20) transitioned into closed vesicles at 280 K.

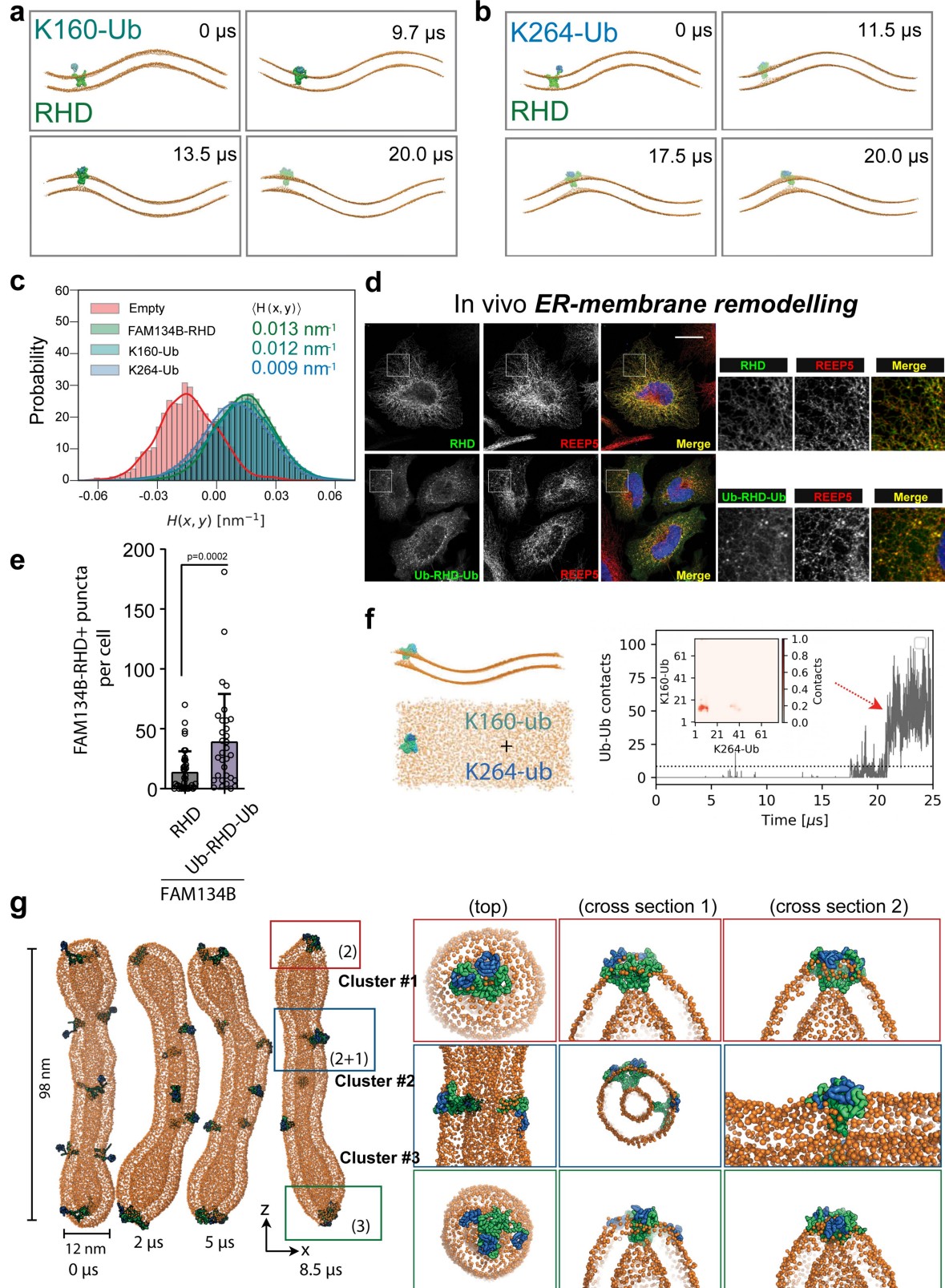

**Extended Data Fig. 4** | See next page for caption.

**Extended Data Fig. 4 | Curvature preference of ubiquitinated FAM134B-RHDs. a,b**, Selected snapshots from 20-μs MD simulations of single K160-Ub and K264-Ub variants on buckled POPC bilayers. The Ub-variants sample regions of high mean curvature and preferentially occupy the top of the buckle. **c**, Histograms of the mean curvature $H(x, y)$ sampled by FAM134B RHD (green), K160-Ub (cyan) and K264-Ub (steel blue) in coarse-grained simulations (1-ns intervals for 20 μs) indicate a preference for highly curved regions of the buckle. For reference, the distribution of local mean curvature values on the empty buckled membrane (red) was estimated by random sampling of points in the $xy$ plane, ignoring small curvature corrections. The time-averaged values of $H(x, y)$ for each system are also provided. **d**, Confocal fluorescence microscopy analysis of cells transiently expressing HA-tagged FAM134B $RHD_{90-264}$ and the chimaera Ub-$RHD_{90-264}$-Ub. Fixed cells were stained with anti-HA (green) and anti-REEP5 (red). Scale bar = 10μm. **e**, Co-localisation analysis of RHD/REEP5+ puncta per cell in cells expressing FAM134B $RHD_{90-264}$ or Ub-$RHD_{90-264}$-Ub (bar plots of data are presented, statistical significance was determined using a non-parametric two-tailed Mann-Whitney U-test, $n$ = 10 cells per condition) Bars represent means ± s.d. **f**, Ubiquitin moieties tethered to different RHDs trigger inter-molecular Ub-Ub interactions, which induce RHD clustering and dimerisation on top of the buckle (left) after ~18 μs. (right) Time series of inter-molecular Ub-Ub contacts across the two molecules. The inset shows the average Ub-Ub contact map stabilising the dimer structure (21–25 μs). **g**, Snapshots of membrane tubule with 10 Ub-RHDs (left) Snapshots of Ub-RHD cluster formation that locally deform membrane tubules from ideal cylindrical geometry. Coloured squares (red, blue and green) highlight the number of molecules (dimers/trimers) that form the clusters locally on different parts of the tubule. Ubiquitinated RHD clusters shape the tubule (right). Zoomed RHD clusters show Ub-Ub interactions (blue moieties).

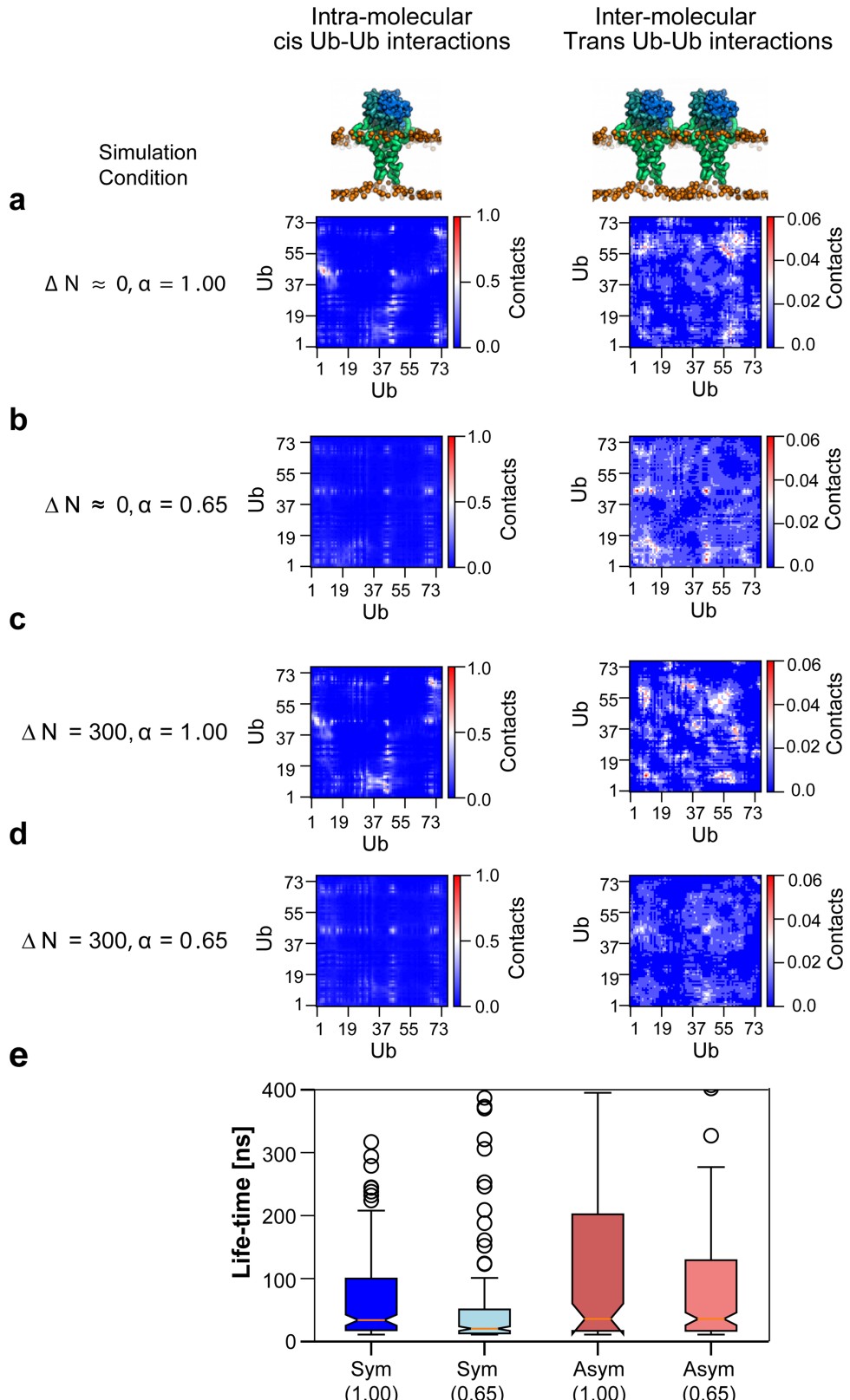

**Extended Data Fig. 5** | See next page for caption.

**Extended Data Fig. 5 | Ub-Ub interactions from MD simulations of ubiquitinated RHDs.** 9 (K160+K264)-Ub-RHD molecules were simulated in a 3x3 square grid embedded in POPC bilayer under 4 different simulation conditions ($\Delta N = 0$ and 300, and $\alpha = 1$ and 0.65) to quantify the Ub-Ub interactions (schematic at the top). **a-d**, The two Ub moieties of (K160+K264)-Ub-RHD molecules are engaged in both cis/intra-molecular (left) and trans/inter-molecular interactions (right). Interactions under each simulation condition were quantified by computing an average Ub-Ub contact map, $UU_{cnts} = \left[ \sum_{i \in Ub_1} \sum_{j \in Ub_2} \sigma(|r_{ij}|) \right]$, where the sums extend over residue COM positions of interacting Ub moieties and $\sigma(|r_{ij}|)$ is a smooth sigmoidal counting function to limit interactions below the cut-off ($r_{ij} \leq 10$ Å). Interaction maps were averaged over only the bound states ($\sum UU_{cnts} \geq 5$) for all nine intra-molecular Ub-Ub pairs (left) and 144 inter-molecular Ub-Ub pairs (right), respectively. **e**, Box plots showing the distribution of lifetimes of trans/inter-molecular Ub-Ub interactions from each of the four simulation conditions. The lifetime for each trans-Ub-Ub bound state was estimated as a contiguous stretch (in ns) where (($\sum UU_{cnts} \geq 5$) for 10 ns). Lifetimes from a total of 216, 276, 143 and 304 trans-Ub-Ub bound states were sampled over the course of 5-μs runs for each of the four simulation conditions, respectively.

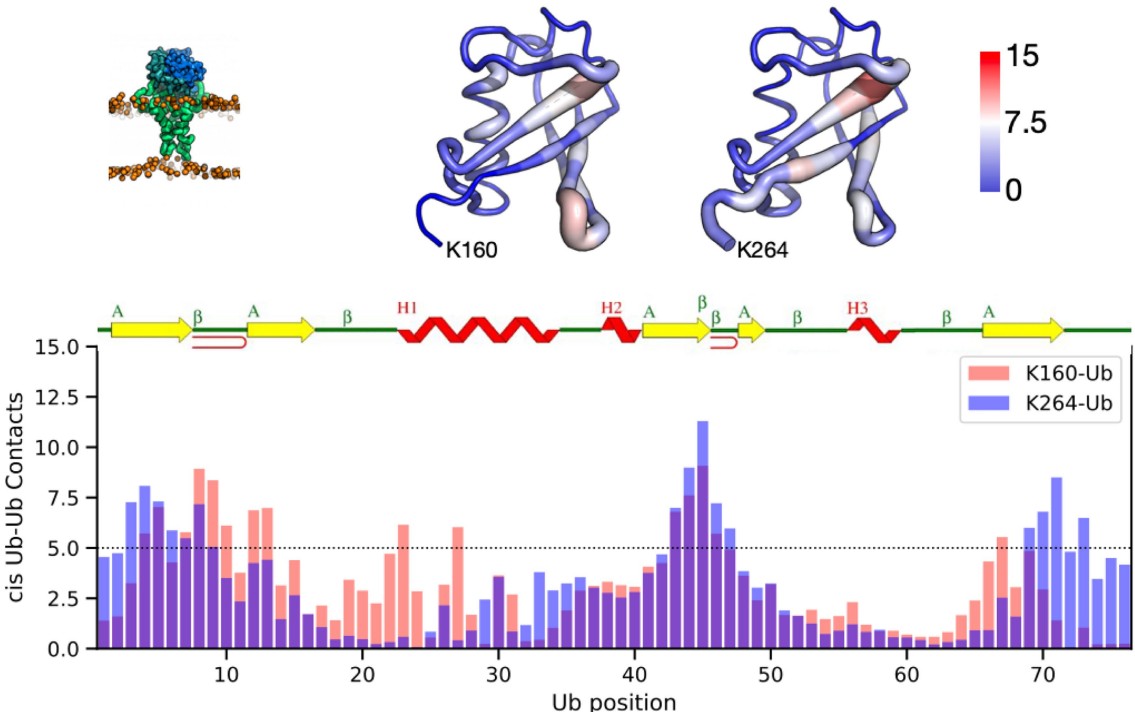

## a Intra-molecular (cis Ub-Ub interactions)

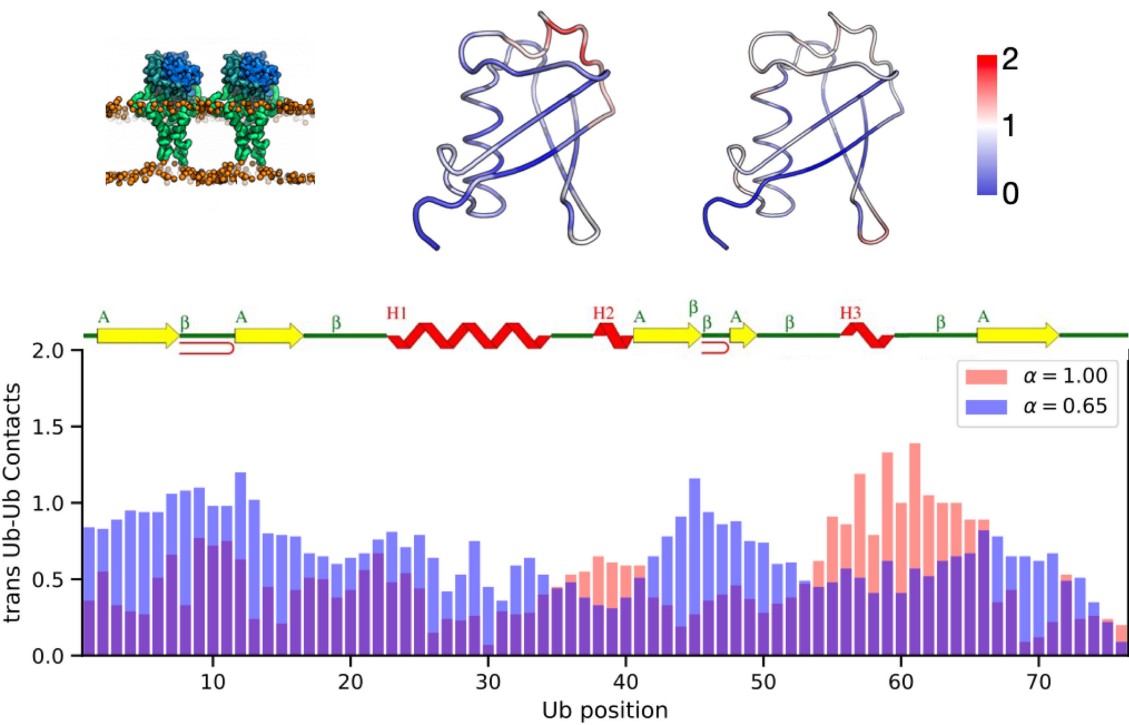

## b Inter-molecular (trans Ub-Ub interactions)

**Extended Data Fig. 6 | Map of Ub-Ub interactions. a**, (Top) Schematic showing the cis/intra-molecular Ub-Ub interactions and **b**, the trans/inter-molecular Ub-Ub interactions mapped onto the 3D structure of Ub. Strongly interacting (red), moderately interacting (white), and weakly interacting (blue) sites are coloured, and the cartoon backbone size is scaled accordingly. (Bottom) The same interactions are mapped along the sequence of Ub to highlight the various secondary structural elements and residues involved, revealing that hairpins β12 and β34, along with the C-terminal region of Ub, dominate the intra-molecular interactions of K160-Ub and K264-Ub. However, trans-Ub-Ub interactions are spread throughout the Ub sequence, indicating their non-specific nature.

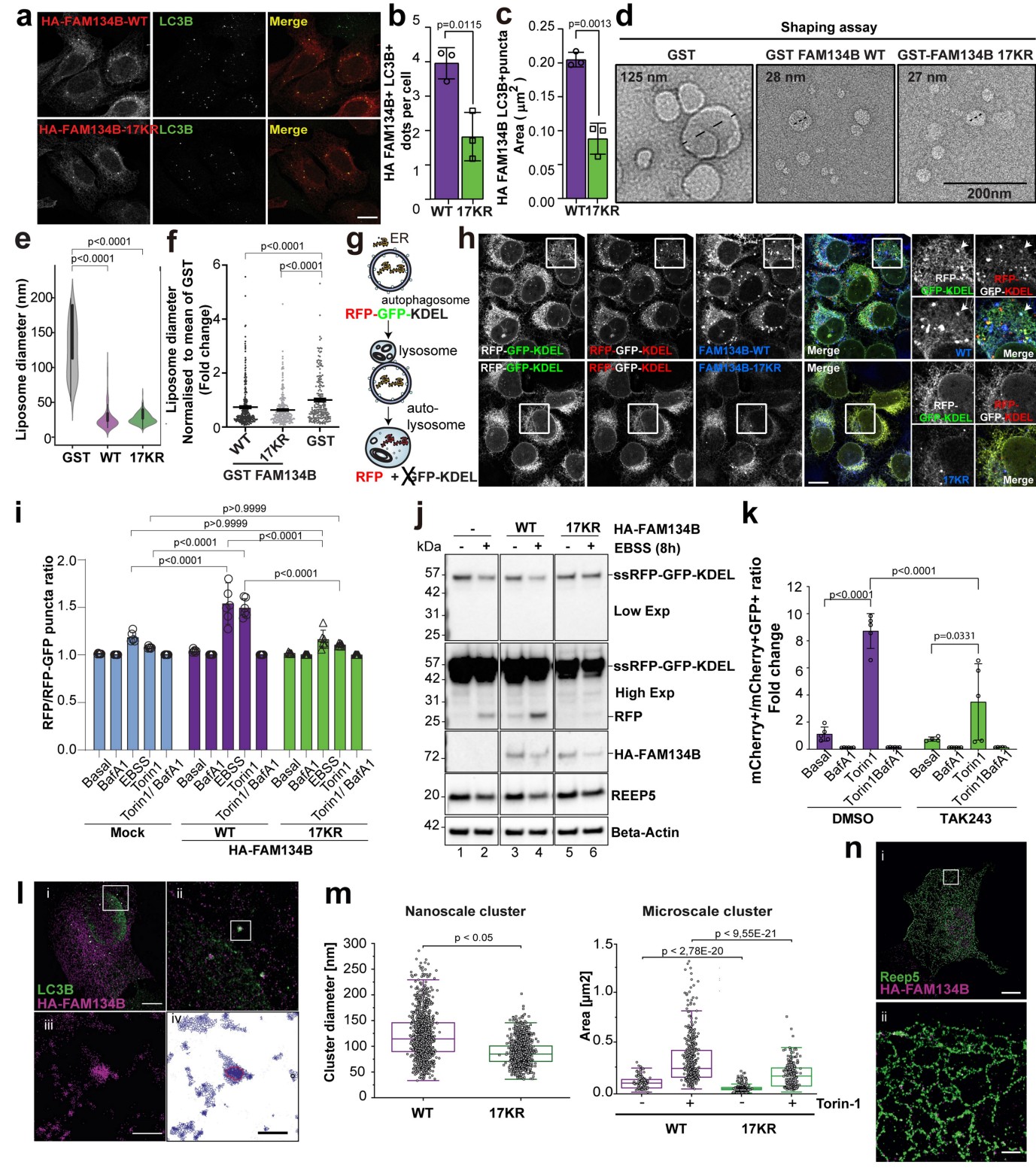

**Extended Data Fig. 7** | See next page for caption.

**Extended Data Fig. 7 | Effect of RHD ubiquitination on the flux of ER-phagy and FAM134B cluster size. a**, Representative confocal images of HA-FAM134B WT and 17KR (red) co-stained with LC3B (green). **b**, Co-localisation of HA-FAM134B/LC3B+ puncta and the **c**, corresponding area ($\mu m^2$) were lower for 17KR compared to WT. The data are representative of three independent experiments in which the total number of cells per condition were $n_{cells}$ = 488 WT, 392 17KR. (Data are mean ± s.d.; two-tailed unpaired Student's t-test). **d**, Representative negative-stain transmission electron microscopy (TEM) images of remodelled proteoliposomes (scale bars = 200 nm). Empty liposomes were incubated with purified GST empty, GST-FAM134B-WT or GST-FAM134B-17KR for 18 h at 25 °C. Images show examples of representative proteoliposomes. **e**, Violins shows the box-plots with median value, white dot, interquartile range (black shaded region), min and max values (1.5 x interquartile region) and mirrored probability density estimates on sides. (WTmean = 28.25; 17KR mean = 27.73; GSTmean = 128.78; GST empty ($n$ = 167), GST-FAM134B-WT ($n$ = 277) or GST-FAM134B-17KR ($n$ = 297); Kruskal-Wallis/Dunn's post-hoc test. **f**, Quantitative TEM analyses of diameters of freeze-fractured liposomes that were incubated with either GST control, GST FAM134B WT or GST FAM134B 17KR. Data, mean ± SEM presented as normalized to the mean liposome diameter of GST control. Two independent liposome preparations and experiments. Kruskal-Wallis/Dunn's post-test. ($n$ = 310 for WT, $n$ = 250 for 17KR and $n$ = 208 for GST control). **g**, Schematic representation of the ER-phagy reporter system RFP-GFP-KDEL. **h**, Representative confocal images of U2OS TRex stable cell lines co-expressing RFP-GFP-KDEL with either HA-FAM134B WT or HA FAM134B 17KR. Cells were treated for 16 h with 1 μg/ml doxycycline to induce the expression of both proteins. Cells were fixed, permeabilised, and stained for HA and LC3B. Bar = 10 μm. **i**, ER-phagy flux was quantified as the ratio between RFP+/GFP− and RFP+/GFP+ puncta, quantified using CQ1 software. Cells were treated with DMSO (control), 200 nM BafA1 for 6 h, EBSS for 6 h, 250 nM Torin1 for 6 h or 250 nM Torin1 plus 200 nM BafA1 for 6 h. Data are means ± s.d. of $n$ = 6 independent experiments, in which the number of RFP-GFP-KDEL cells per condition are: 2366 (DMSO), 2202 (BafA1), 1460 (EBSS), 2228 (Torin1), 2378 (Torin1+BafA1). Number of RFP-GFP-KDEL/HA-FAM134B WT cells: 670 (DMSO), 773 (BafA1), 578 (EBSS), 986 (Torin1), 747 (Torin1+BAfA1), Number of HA-FAM134B 17KR cells: 1160 (DMSO), 1151 (BafA1), 1313 (EBSS), 1353 (Torin1), 1295 (Torin1+BafA1). One-way ANOVA, Bonferroni post-hoc test. **j**, RFP-GFP-KDEL, RFP-GFP-KDEL/HA-FAM134B WT and RFP-GFP-KDEL/HA-FAM134B 17KR cells were left untreated or treated with EBSS for 8 h.

Detergent-soluble extracts were analysed by western blot using antibodies against RFP, HA, REEP5 and β-actin. ($n$ = 1 experiment). **k**, The E1 inhibitor decreases the flux of ER-phagy in mCherry-GFP-FAM134B-WT cells induced with Torin 1 (Data are mean ± s.d.; one-way ANOVA, Bonferroni post-hoc test). ER-phagy flux was quantified as the ratio between mCherry+/GFP− and mCherry+/GFP+ puncta, quantified using CQ1 software. Data are means ± s.d. of $n$ = 5 independent experiments in which the number of cells per condition were: (DMSO) 837 basal, 1072 BafA1, 1038 Torin1, 966 BafA1+Torin1. Number of cells (10 μM TAK243): 729 basal, 1174 BafA1, 1060 Torin1, 1121 Torin1+BafA1. **l**, Two-colour DNA-PAINT super-resolution image of HA-FAM134B (magenta, R2-ATTO655) and the autophagosomal membrane marker LC3B-II (green, R1-ATTO655) (i). White box indicates the magnified region shown in (ii). (iii) Point localisations of HA-FAM134B from the magnified region shown in (ii) and corresponding Voronoi diagrams (blue polygons) with red line representing FAM134B cluster contour (iv). Clusters are identified based on previously determined thresholds (density factor, minimum number of localisations and minimal distance parameter). Scale bars = 10 μm (i) and 1 μm (ii–iv). **m**, Box plot of HA-FAM134B-WT and HA-FAM134B-17KR nanoscale and microscale cluster sizes. For nanoscale clusters, ubiquitination-deficient FAM134B significantly reduces the cluster diameter (35–202 nm, median = 85 nm) compared to its WT counterpart (33–286 nm, median = 114 nm). Nanoscale cluster ($n_{cells}$ = 4, $n_{WTclusters}$ = 1278; $n_{17KRclusters}$ = 1255). For larger microscale clusters, significantly larger areas were detected for HA-FAM134B-WT (0.017–0.20 $\mu m^2$, median = 0.08 $\mu m^2$) compared to the 17KR mutant (0.01–0.21 $\mu m^2$, median = 0.03 $\mu m^2$). Torin 1 treatment further increased the HA-FAM134B cluster area with the effect being stronger for ubiquitinated HA-FAM134B-RHD (median$_{WT}$ = 0.23 $\mu m^2$, median$_{17KR}$ = 0.16 $\mu m^2$). Quantitative analysis of nanoscale clusters was carried out using the DBSCAN algorithm and microscale clusters were identified using SR-tessellation. Box-plots of FAM134B wildtype (magenta, grey dots) and FAM134B 17 KR (green, grey squares) nanoscale cluster diameters (left panel) and microscale cluster areas (right panel, grey dots) showing median values (horizontal lines in boxes), the interquartile ranges (width of the boxes) and whiskers defining minimum and maximum values (excluding outliers). A non-parametric one-tailed Mann-Whitney U-test was applied to the data. **n**, HA-FAM134B forms nanoscale clusters within the ER network. Two-colour super-resolution image of HA-FAM134B-WT (magenta, R2-ATTO655) and ER-membrane marker REEP5 (green, R1-ATTO655), with (ii) showing the magnified region from box (i). Scale bars = 10 μm (i) and 1 μm (ii).

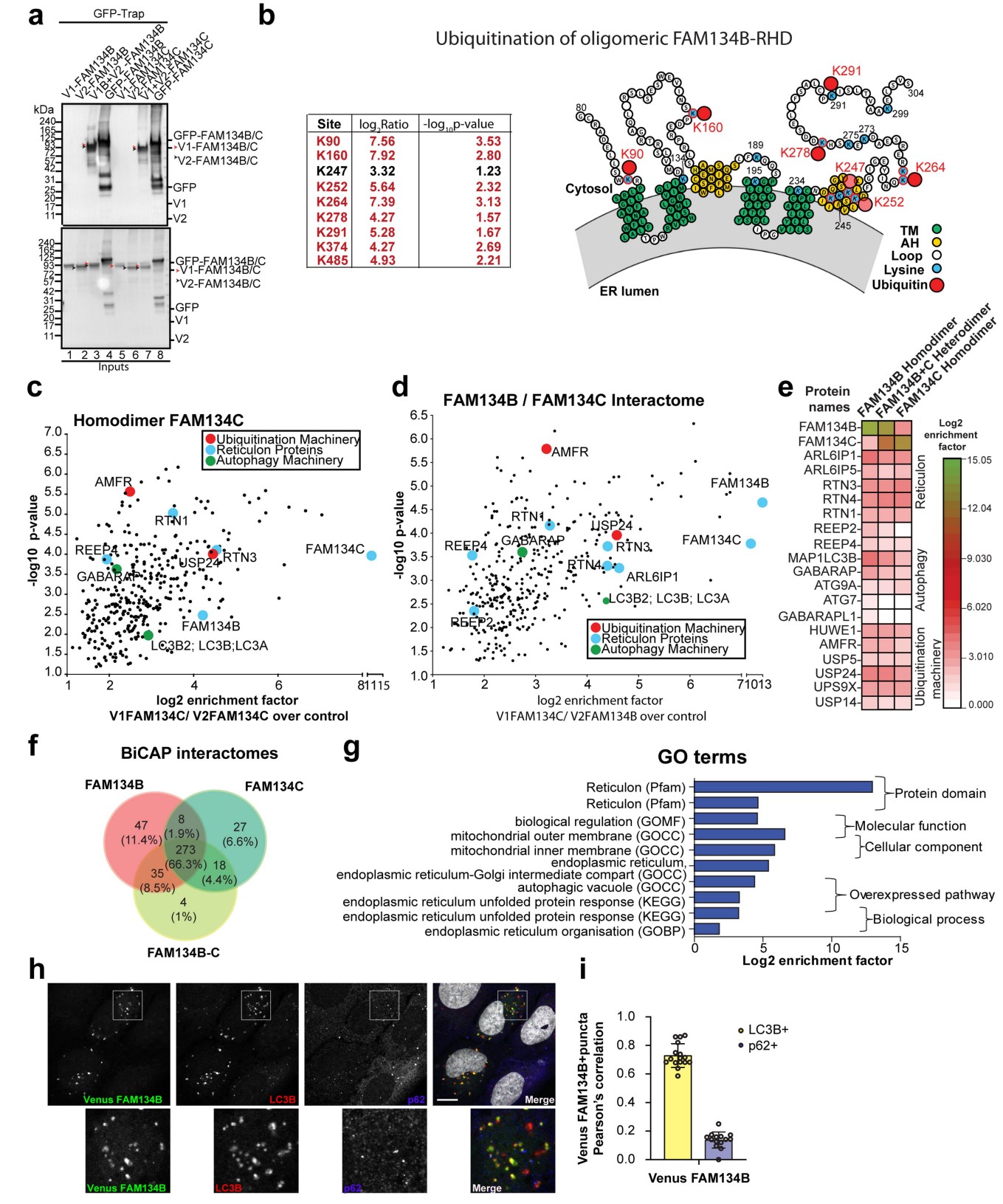

**Extended Data Fig. 8** | See next page for caption.

**Extended Data Fig. 8 | Protein interactors of FAM134B/FAM134C-containing oligomers. a**, HEK 293T cells were transfected with a control plasmid (GFP), V1-FAM134B-WT, V2-FAM134B-WT, V1-FAM134C-WT or V2-FAM134C-WT. Cells were also co-transfected with V1-FAM134B-WT and V2-FAM134B-WT or V1-FAM134C-WT and V2-FAM134C-WT. The GFP trap was analysed by western blot. (*n* = 1 experiment). **b**, Ubiquitinated lysine residues identified by proteomics analysis in the RHD of immuno-isolated dimeric FAM134B: diGly peptides were significantly enriched ($\log_2$ enrichment > 2.0 and $-\log_{10}$ p value > 1.3, one-tailed unpaired Student's test). *n* = 3 independent experiments. Schematic of the FAM134B-RHD showing the localisation of ubiquitinated lysine residues. **c**, Single-sided volcano plot of the quantitative label-free interactome of FAM134C homodimers and **d**, FAM134B/FAM134C heterodimers depicting identified RHD-containing ER proteins (blue), autophagy-related proteins (green), and components of ubiquitination machinery (red) ($\log_2$ enrichment > 2.0 and $-\log_{10}$ p value > 1.3). Data are means ± s.d. of *n* = 3 independent experiments. **e**, Heat map comparing the interaction of RHD-containing ER proteins, autophagy-related proteins and the ubiquitination machinery with WT FAM134B homodimers, FAM134C homodimers and FAM134B/FAM134C heterodimers (immuno-isolated using BiCAP). Interaction partners with $\log_2$ enrichment > 2.0 and $-\log_{10}$ p value > 1.3 were plotted. *n* = 3 independent experiments, one-tailed unpaired Student's test. **f**, Venn diagram of interactors of FAM134B homodimers, FAM134C homodimers and FAM134B /FAM134C heterodimers. Numbers represent significantly enriched interaction partners ($\log_2$ enrichment > 2.0 and $-\log_{10}$ p > 1.3, one-tailed unpaired Student's test). *n* = 3 independent experiments. **g**, Annotation enrichment analysis of the interactome of FAM134B and FAM134C heterodimers. Bars represent significantly enriched gene ontology biological process (GOBP), gene ontology cellular component (GOCC), gene ontology molecular function (GOMF), and domain enrichment (Pfam). **h**, Confocal fluorescence microscopy analysis of the BiFC signal produced by interactions between V1-FAM134B and V2-FAM134B. Fixed cells expressing V1-FAM134B and V2-FAM134B were stained for LC3B (red) and p62 (blue). Scale bar = 10 μm. **i**, Pearson's correlation coefficients obtained from the co-localisation analysis of fluorescent signals representing FAM134B clusters and p62 or LC3B. Data are means ± s.d. of *n* = 10 cells per analysis.

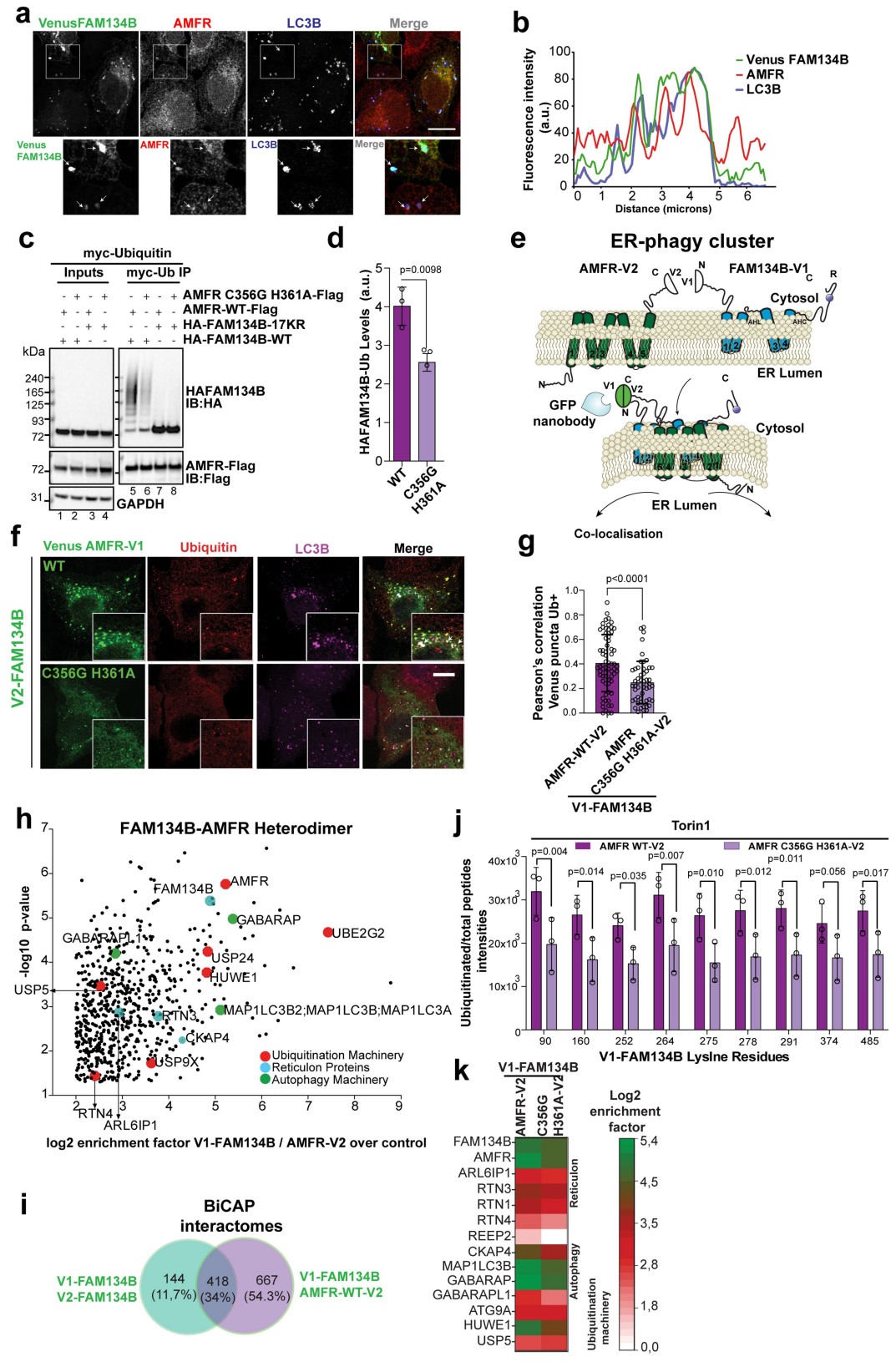

**Extended Data Fig. 9** | See next page for caption.

**Extended Data Fig. 9 | Analysis of the functional interaction between AMFR and FAM134B in cells. a**, Confocal fluorescence microscopy analysis of the BiFC signals following the interaction between V1-FAM134B and V2-FAM134B. Fixed cells expressing V1-FAM134B and V2-FAM134B were stained for AMFR (red) and LC3B (blue). Scale bar = 10 μm. **b**, The fluorescence intensity distribution reveals that FAM134B clusters (V1FAM134B+V2FAM134B) co-localise with endogenous AMFR and LC3B in punctate structures. **c**, Ubiquitination assay of HA-FAM134B in cells co-expressing WT-AMFR-Flag or the catalytically inactive AMFR-Flag (C356G H361A) variant. **d**, Densitometric quantification of western blot signals for ubiquitinated HA-FAM134B following co-expression with WT-AMFR-Flag or the catalytically inactive AMFR-Flag (C356G H361A) variant as presented in Fig. 9c. Data are means ± s.d. of $n$ = 3 independent experiments, two-tailed unpaired Student's t test. **e**, Schematic of the BiCAP method to study the functional interaction between FAM134B and AMFR. Full-length FAM134B was fused to the C-terminal of the non-fluorescent N-terminal (V1) fragment of Venus protein (V1-FAM134B), whereas full-length AMFR was fused to the N-terminal of the non-fluorescent C-terminal (V2) fragment (AMFR-V2). **f**, Confocal microscopy analysis of fixed cells co-expressing V1-FAM134BWT and AMFR-V2 WT or V1-FAM134BWT and AMFR-V2 C356G H361A stained for Ub and LC3B (magenta). Scale bar = 10 μm. **g**, Pearson's correlation coefficients obtained from co-localisation of the fluorescent signals representing refolded Venus and Ub(FK2) in cells co-expressing V1-FAM134BWT and AMFR-V2 WT or V1-FAM134BWT and AMFR-V2 C356G H361A (Extended Data Fig. 9e), (Two-tailed non-parametric Mann-Whitney U-test was applied to the data) Data are means ± s.d. $n_{\text{V1FAM134B/V2AMFRWT}}$ = 68 puncta and $n_{\text{V1FAM134B/V2AMFR C356G H361A}}$ = 52 puncta. **h**, Single-sided volcano plot of the quantitative label-free interactome of the affinity-purified (using BiCAP) WT-AMFR-V2/V1-FAM134BWT complex. RHD-containing ER proteins (blue), autophagy-related proteins (green), and ubiquitination machinery (red) ($\log_2$ enrichment > 2.0 and $-\log_{10}$ value > 1.3). Data are means of $n$ = 3 independent experiments; one-tailed unpaired Student's test. **i**, Venn diagram of interactors of FAM134B homodimers and AMFR/FAM134B heterodimers. Numbers represent significantly enriched peptides ($\log_2$ enrichment > 2.0 and $-\log_{10}$ p value > 1.3). Data are means of $n$ = 3 independent experiments, one-tailed unpaired Student's test. **j**, Comparison of Torin 1-induced ubiquitination of FAM134B within complexes with either AMFR-WT (WT-AMFR-V2/V1-FAM134B) or the catalytically inactive AMFR mutant (C356G H361A AMFR-V2/V1-FAM134B). Graphs show the abundance of FAM134B peptides carrying a diGly modification expressed as intensities. Data are means ± s.d. of $n$ = 3 independent experiments and the identified diGly peptides intensities were normalised to the total intensities of modified and non-modified peptides (two-tailed unpaired Student's test). **k**, Heat map comparing the interaction of RHD-containing ER proteins, autophagy-related proteins and the ubiquitination machinery of WT-AMFR-V2/V1-FAM134B and AMFR C356G H361A-V2/V1-FAM134B complexes. Interaction partners with $\log_2$ enrichment > 2.0 and $-\log_{10}$ p value > 1.3 were plotted. Data are means of $n$ = 3 independent experiments, one-tailed unpaired Student's test.

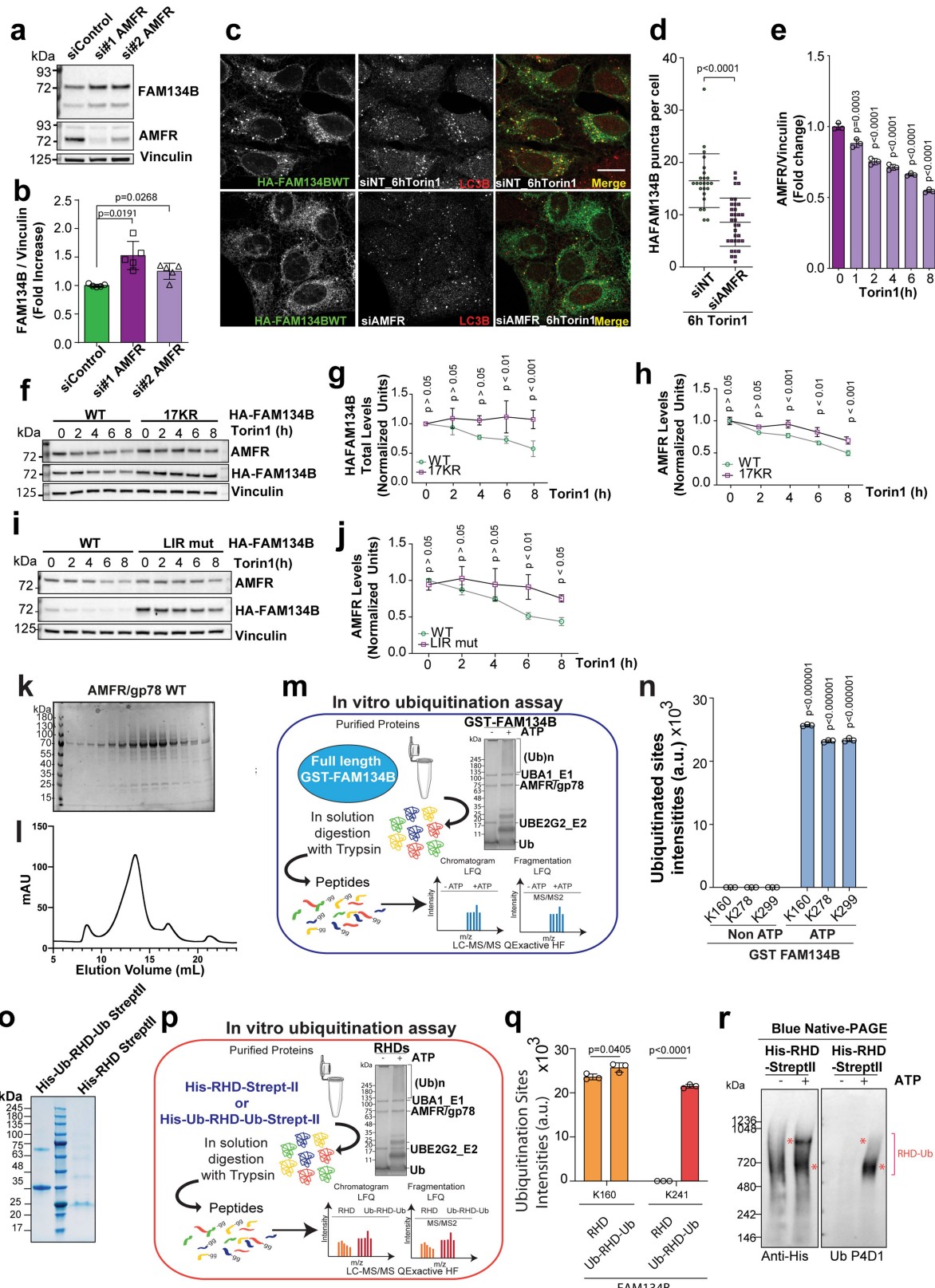

**Extended Data Fig. 10 | Analysis of the functional interaction between AMFR and FAM134B in cells and *in vitro*. a**, HeLa cells were transfected with control siRNA (siNT), siRNA#1 or siRNA#2 targeting AMFR (siAMFR) for 72 h. Detergent-soluble protein extracts were analysed by western blot using antibodies against FAM134B, AMFR or vinculin (loading control). **b**, Densitometric quantification of endogenous FAM134B (normalised to vinculin) in panel a (Data are means ± s.d. of $n = 5$ independent experiments, One-way ANOVA, Bonferroni post-hoc test). **c**, Confocal fluorescence microscopy of U2OS cells stably expressing HA-FAM134BWT transfected with either control siRNA (siNT) or siRNA#1 targeting AMFR (siAMFR) for 72 h, followed by incubation with 250 nM Torin 1 for 6 h. Cells were fixed and stained for HA-FAM134B and endogenous LC3B, respectively. **d**, Quantification of HA-FAM134B-WT/LC3B-II-containing puncta per cell of images in panel c. Scatter plot graphs represent means ± s.d. ($n_{siNT} = 24$ cells, $n_{siAMFR} = 33$ cells; two-tailed Mann-Whitney-U-test). **e**, HeLa cells were treated with 250 nM Torin 1 for the indicated time (h). Densitometric quantification of endogenous AMFR (normalised to vinculin), as presented in Fig. 5a. Data are means ± s.d. of $n = 3$ independent experiments; two-way ANOVA, Bonferroni post-hoc test. **f**, U2OS cells stably expressing HA-FAM134BWT or HA-FAM134B-17KR were incubated with 250 nM Torin 1 for 0, 2, 4, 6 and 8 h. Detergent-soluble extracts were analysed by western blot using antibodies against HA, AMFR and vinculin. **g**,**h**, Densitometric quantification of HA-FAM134B and AMFR (normalised to vinculin) from panel f. Data are means ± s.d. of $n = 3$ independent experiments, two-way ANOVA, Bonferroni post-hoc test. **i**, U2OS cells stably expressing HA-FAM134BWT or HA-FAM134B-LIR incubated with 250 nM Torin 1 for 0, 2, 4, 6 and 8 h. Detergent-soluble extracts were analysed by western blot using antibodies against HA, AMFR and vinculin. **j**, Densitometric quantification of AMFR (normalised to vinculin) in panel i. Data are means ± s.d. of $n = 3$ independent experiments, two-way ANOVA, Bonferroni post-hoc test. **k**, and **l**, Purification of AMFR from HEK 293T cells. **m**,**n**, Mass spectrometry (MS) analysis of the *in vitro* ubiquitination of full length GST-tagged FAM134B using recombinant AMFR. (Data are means ± s.d. of $n = 3$ independent experiments; two-tailed unpaired Student's t-test). **o**–**q**, MS analysis of the *in vitro* ubiquitination of His-RHD$_{90-264}$-Strept-II and His-Ub-RHD$_{90-264}$-Ub-Strept-II using recombinant AMFR. Data are means ± s.d. of $n = 3$ independent experiments; two-tailed unpaired Student's t-test. **r**, Immunodetection of native His-RHD$_{90-264}$-Strept-II following ubiquitination by AMFR. The ubiquitination reaction was analysed by western blot after blue native polyacrylamide gel electrophoresis (BN-PAGE) using antibodies against His$_6$ or UbP4D1. Control reaction (in the presence of AMFR, no ATP).

# Reporting Summary

## Statistics

For all statistical analyses, confirm that the following items are present in the figure legend, table legend, main text, or Methods section.

| n/a | Confirmed | |
|---|---|---|
| ☐ | ☒ | The exact sample size (*n*) for each experimental group/condition, given as a discrete number and unit of measurement |
| ☐ | ☒ | A statement on whether measurements were taken from distinct samples or whether the same sample was measured repeatedly |
| ☐ | ☒ | The statistical test(s) used AND whether they are one- or two-sided<br>*Only common tests should be described solely by name; describe more complex techniques in the Methods section.* |
| ☒ | ☐ | A description of all covariates tested |
| ☐ | ☒ | A description of any assumptions or corrections, such as tests of normality and adjustment for multiple comparisons |
| ☐ | ☒ | A full description of the statistical parameters including central tendency (e.g. means) or other basic estimates (e.g. regression coefficient) AND variation (e.g. standard deviation) or associated estimates of uncertainty (e.g. confidence intervals) |
| ☐ | ☒ | For null hypothesis testing, the test statistic (e.g. *F*, *t*, *r*) with confidence intervals, effect sizes, degrees of freedom and *P* value noted<br>*Give P values as exact values whenever suitable.* |
| ☒ | ☐ | For Bayesian analysis, information on the choice of priors and Markov chain Monte Carlo settings |
| ☒ | ☐ | For hierarchical and complex designs, identification of the appropriate level for tests and full reporting of outcomes |
| ☐ | ☒ | Estimates of effect sizes (e.g. Cohen's *d*, Pearson's *r*), indicating how they were calculated |

*Our web collection on statistics for biologists contains articles on many of the points above.*

## Software and code

Policy information about availability of computer code

| Data collection | 1. Western Blots signal detection was carried out with the Chemidoc automated detection system (Biorad). Using Image Lab 6.1 Software For Windows.<br>2. ER fragmentation and ER-phagy flux assays were acquired with high content microscope-Yokogawa CQ1 confocal imaging cytometer (CQ1 software, v1.04.07.01).<br>3. MS raw data was processed with MaxQuant (v1.6.17.0).<br>4. We performed coarse-grained MD simulations using the MARTINI model (version 2.2).<br>5. Leica LAS X v2.0.2.15022 |
|---|---|
| Data analysis | 1. Western blot analysis was perfomed using Using Image Lab 6.1 Software For Windows.<br>2. Densitometric quantification of western blots bands was carried out using ImageJ (version 1.51w).<br>3. ER fragmentation and ER-phagy flux assays were analyzed with high content microscope-Yokogawa CQ1 confocal imaging cytometer (CQ1 software).<br>4. MS raw data was analysed with MaxQuant (v1.6.17.0). Protein quantification and data normalisation relied on the MaxLFQ algorithm implemented in MaxQuant<br>5. The Perseus software (v2.0.7.0 ) was used and first filtered for contaminants and reverse entries as well as proteins that were only identified by a modified peptide.<br>6. The data analysis and graphs were generated with GraphPad Prism 9.4.1<br>7. Single-molecule localisation and image reconstruction was conducted with the modular software package Picasso v0.2.8<br>8. Multi-channel 3D-localised single-molecule localisations from exchange DNA-PAINT experiments were aligned in Picasso v0.2.8 software and visualised in ViSP 1.0<br>9. FAM134B and LC3B-II nanocluster were identified in DNA-PAINT images using the density-based spatial clustering and application with noise (DBSCAN) algorithm. DBSCAN algorithm is included in Picasso v0.2.8 software |

10. Microscale FAM134B clusters were segmented from nanoscale ER-phagy initiation sites using SR-Tesseler software 1.0.0.1
11. Diameters of liposomes were determined using ImageJ software (version 1.51w).

For manuscripts utilizing custom algorithms or software that are central to the research but not yet described in published literature, software must be made available to editors and reviewers. We strongly encourage code deposition in a community repository (e.g. GitHub). See the Nature Portfolio guidelines for submitting code & software for further information.

# Data

Policy information about availability of data

All manuscripts must include a data availability statement. This statement should provide the following information, where applicable:

- Accession codes, unique identifiers, or web links for publicly available datasets
- A description of any restrictions on data availability
- For clinical datasets or third party data, please ensure that the statement adheres to our policy

1. Acquired spectra were searched against the human "one sequence per gene" database (Taxonomy ID 9606) downloaded from UniProt (12-03-2020; 20531 sequences), and a collection of 244 common contaminants ("contaminants.fasta" provided with MaxQuant v1.6.17.0) using the Andromeda search engine integrated into MaxQuant v1.6.17.0

2. For protein assignment, spectra were correlated with the Uniprot human database (v. 2019) including a list of common contaminants.

3. The proteomics data are deposited in the ProteomeXchange Consortium via the PRIDE partner repository with the dataset identifiers: Ubiquitination promotes FAM134B-mediated ER-phagy: PXD032721, www.ebi.ac.uk/pride/archive/simpleSearch?q=PXD032721; FAM134B homodimer ubiquitination: PXD032740, www.ebi.ac.uk/pride/archive/simpleSearch?q=PXD032740; FAM134B oligomer ubiquitination and ER-phagy: PXD032741, www.ebi.ac.uk/pride/archive/simpleSearch?q=PXD032741; Binding partners of FAM134B WT and 17KR oligomers: PXD032743, www.ebi.ac.uk/pride/archive/simpleSearch?q=PXD032743; E3 ligase regulates FAM134B ubiquitination: PXD032750, www.ebi.ac.uk/pride/archive/simpleSearch?q=PXD032750; In vitro FAM134B ubiquitination: PXD039186, www.ebi.ac.uk/pride/archive/simpleSearch?q=PXD039186; In vivo FAM134B ubiquitination in AMFR KD cells: PXD039187, www.ebi.ac.uk/pride/archive/simpleSearch?q=PXD039187; In vitro FAM134B RHD and UB RHD UB ubiquitination: PXD039188, www.ebi.ac.uk/pride/archive/simpleSearch?q= PXD039188. MD simulation trajectory files and corresponding parameter files are large and span long microsecond time-scales and multiple replicates can only be shared upon specific requests. All the data analysis of this study is in the Supplementary information. Source data for gels and blots are provided as supplementary information.

# Field-specific reporting

Please select the one below that is the best fit for your research. If you are not sure, read the appropriate sections before making your selection.

☒ Life sciences          ☐ Behavioural & social sciences          ☐ Ecological, evolutionary & environmental sciences

For a reference copy of the document with all sections, see nature.com/documents/nr-reporting-summary-flat.pdf

# Life sciences study design

All studies must disclose on these points even when the disclosure is negative.

| | |
|---|---|
| Sample size | No sample size calculation was done. Assays were repeated at least three times and sample size was chosen based on the significance of measured difference between groups.<br>Sample size was determined based on similar studies in this field. E.g.<br>Grumati P, et. al. (2017) Full length RTN3 regulates turnover of tubular endoplasmic reticulum via selective autophagy. Elife 6: e25555 |
| Data exclusions | No data were excluded from analysis. |
| Replication | All the data with statistical analysis presented in this manuscript was repeated at least three times. Ubiquitination of FAM134B and its role in ER-phagy was validated in different cell lines with different approaches (e.g. mass spectrometry and biochemistry). Single cell analysis included at least three replicates and representative images are presented (confocal and super-resolution images). Results from all technical- and biological replicates were consistent among them. |
| Randomization | No randomization was necessary. Mass spectrometry and biochemistry samples were measured sequentially. Images were automatically acquired for the data analysis by high throughput imaging or super resolution microscopy. |
| Blinding | No blinding was applied in this study. Blinding was not possible as all samples were analyzed pairwise or multiple compared. In all assays in this study the treatment (or different conditions tested) cannot be disguised from the scientist. |

# Reporting for specific materials, systems and methods

We require information from authors about some types of materials, experimental systems and methods used in many studies. Here, indicate whether each material, system or method listed is relevant to your study. If you are not sure if a list item applies to your research, read the appropriate section before selecting a response.

## Materials & experimental systems

| n/a | Involved in the study |
|-----|----------------------|
| ☐ | ☒ Antibodies |
| ☐ | ☒ Eukaryotic cell lines |
| ☒ | ☐ Palaeontology and archaeology |
| ☒ | ☐ Animals and other organisms |
| ☒ | ☐ Human research participants |
| ☒ | ☐ Clinical data |
| ☒ | ☐ Dual use research of concern |

## Methods

| n/a | Involved in the study |
|-----|----------------------|
| ☒ | ☐ ChIP-seq |
| ☒ | ☐ Flow cytometry |
| ☒ | ☐ MRI-based neuroimaging |

## Antibodies

**Antibodies used**

Most of the antibodies are commecially availaible and catalog numbers are provided in supplementary information.
1. GAPDH (14C10) Cell signalling # 2118 WB (1/5000) lot14
2. HA Roche (11867423001) Clone 3F10, WB (1/10,000), IF (1/2000)
3. LC3B Rabbit mAb (clone (D11) XP®) #3868 CST WB 1/1000; IF (1/500)
4. REEP5 Proteintech (14643-1-AP) WB, IF (1/1000) Lot: 00042892
5. FLAG (M2) Sigma (F3165-5MG) WB (1/10000), IF (1/1000) Lot#SLBQ7119V
6. FAM134B Proteintech (21537-1-AP) WB (1/2000) Lot:00094171
7. GFP Clontech (Cat. 632460) WB (1/1000) Lot #K1616
8.AMFR Proteintech (16675-AP) IF (1/300), Lot: 00046373
9. FAM134B antibody (U7432CL010) working for immunofluorescence (dilution 1/100)
 was produced by Genescript, Lot: A318020492. Please, request to Dikic laboratory.
10.Mono-polyubiquitin FK2 Biomol # BML-PW8810, Lot: 05021240
11.Mono- and polyubiquitinylated conjugates monoclonal antibody (UBCJ2), enzolifesciences,  ENZ-ABS840-0500, Lot: 08072015
12. Ubiquitin-P4D1 Cell Signalling # 3936 Lot19
13.Vinculin Sigma (V4505) Lot #000013524
14.Anti myc tag Cell signalling #2276 Lot24
15.BSA-free RGS-Hist Antibody, Qiagen, (Cat.No./ID:34650)
16. LC3B MBL (PM036), Lot: 035

Secondary Antibodies for Immunoblot and immunofluorescence:

17. HRP-conjugated anti-rat Cell Signaling (#7077S)
18. Goat anti-mouse HRP, Bio-Rad (Cat Number: 170-6516) Lot: 64510108
19. Goat anti-rabbit HRP, Dako P0448, Lot: 41424306.
20. Anti-rabbit Alexa 488 Life Technology (A21206) Lot 2256732
21. Anti-rabbit Alexa 647 Life Technology (A21244) Lot 1696456
22. Anti-mouse Alexa 488 Life Technology (A21202) Lot 2428531
23. Anti-mouse Alexa 647 Invitrogen (A31571) Lot 2136787
24. Anti-mouse Cy3 MerckMillipore (#AP124C)
25. Anti-rat Alexa 488 Life Technology (A21208)
26. Anti-rat Cy3 MerckMillipore (#AP189C)

**Validation**

1.https://www.cellsignal.de/products/primary-antibodies/gapdh-14c10-rabbit-mab/2118
GAPDH (14C10) Rabbit mAb detects endogenous levels of total GAPDH protein
2. https://www.fishersci.com/shop/products/anti-ha-high-affinity-50-ug/501003325
3. https://www.cellsignal.de/products/primary-antibodies/lc3b-d11-xp-rabbit-mab/3868
LC3B (D11) XP® Rabbit mAb detects endogenous levels of total LC3B protein. Cross-reactivity may occur with other LC3 isoforms. Stronger reactivity is observed with the type II form of LC3B.
4. https://www.ptglab.com/products/REEP5-Antibody-14643-1-AP.htm
14643-1-AP targets REEP5 in WB, IP, IHC, IF, FC, ELISA applications and shows reactivity with human, mouse, rat samples.
5. https://www.sigmaaldrich.com/DE/de/product/sigma/f3165
Anti Flag M2 antibody is used for the detection of Flag fusion proteins
6. https://www.ptglab.com/products/FAM134B-Antibody-21537-1-AP.htm
The immunogen of 21537-1-AP is FAM134B Fusion Protein expressed in E. coli.
7. https://www.labome.com/product/Takara-Bio-Clontech/632460.html
Li W, Yao A, Zhi H, Kaur K, Zhu Y, Jia M, et al. Angelman Syndrome Protein Ube3a Regulates Synaptic Growth and Endocytosis by Inhibiting BMP Signaling in Drosophila. PLoS Genet. 2016;12:e1006062
8. https://www.ptglab.com/products/AMFR-Antibody-16675-1-AP.htm
16675-1-AP targets AMFR/GP78 in WB, IP, IHC, IF, CoIP, ELISA applications and shows reactivity with human, mouse, rat samples.
9.FAM134B antibody (U7432CL010) working for immunofluorescence (dilution 1/100). Validation of this antibody by IF (localisation of FAM134B in the endoplasmic reticulum) is showed in this manuscript in Figure 3d, 3f and 3h, Extended Data 7l and extended Data 7n. Please, request to Dikic laboratory.
10.https://www.ncbi.nlm.nih.gov/pmc/articles/PMC3714537/
Figure 2A: Cells were simultaneously stained with antibodies against ubiquitin (FK2) and the Salmonella marker common structural antigen-1 (CSA-1)

11. https://www.enzolifesciences.com/ENZ-ABS840/mono-and-polyubiquitinylated-conjugates-recombinant-monoclonal-antibody-ubcj2/
Recognizes mono- and polyubiquitinylated protein conjugates in a wide range of species.
12. https://www.cellsignal.com/products/primary-antibodies/ubiquitin-p4d1-mouse-mab/3936
Ubiquitin (P4D1) Mouse mAb detects ubiquitin, polyubiquitin and ubiquitinated proteins. This antibody may cross-react with recombinant NEDD8.

13. https://www.sigmaaldrich.com/DE/en/search/vinculin-v4505?
focus=products&page=1&perpage=30&sort=relevance&term=vinculin%20v4505&type=product_name
The antibody reacts best with cultured chicken fibroblasts. Labeling also may be obtained with bovine, human, or mouse cells.
14. https://www.cellsignal.com/products/primary-antibodies/myc-tag-9b11-mouse-mab/2276
Myc-Tag (9B11) Mouse mAb detects exogenously expressed proteins containing the Myc epitope tag.
15. https://www.qiagen.com/us/products/discovery-and-translational-research/protein-purification/tagged-protein-expression-purification-detection/anti-his-antibodies-bsa-free
Highly sensitive and specific detection of RGS·His epitopes
16. https://www.mblintl.com/products/pm036/
This antibody reacts with LC3 (MAP1LC3A, B, C) on Western blotting, Immunoprecipitation, Immunohistochemistry, Immunocytochemistry and Flow cytometry. It does not react with GABARAP and GATE-16.

# Eukaryotic cell lines

Policy information about cell lines

| | |
|---|---|
| Cell line source(s) | HEK293T (ATCC® CRL-3216™), U2OS (ATCC®HTB-96™) and HeLa (ATCC® CCL-2™) cells were obtained from ATCC. U2OS TRex cells were provided by Prof. Stephen Blacklow (Brigham and Women's Hospital and Harvard Medical School). |
| Authentication | Cell line authentication was initially performed by ATCC. Further authentication was performed by microscopy, as all three cell lines used in this study (HEK293T, HeLa or U2OS Trex) have quite distinct morphology |
| Mycoplasma contamination | Cell lines were tested periodically for mycoplasma contamination. No contamination was found. |
| Commonly misidentified lines (See ICLAC register) | No commonly misidentified cell lines were used in this study |

