## [Peer Review File · Nature]

Manuscript Title: Ubiquitination regulates ER-phagy and remodelling of endoplasmic reticulum

Reviewer Comments & Author Rebuttals

Reviewer Reports on the Initial Version:

Referees' comments:

Referee #1 (Remarks to the Author):

Following previous work showing the contribution of the transmembrane protein FAM134B in shaping the ER and in triggering ER phagy, the Dikic and Hummer labs propose here that ubiquitination of FAM134B contributes to its role as an ER phagy receptor by boosting FAM134B clustering. The authors use two main complementary approaches: high resolution light microscopy assays coupled to protein engineering to evaluate FAM134B ubiquitination and clustering; coarsed-grain simulations to predict the impact of FAM134B ubiquitination on membrane dynamics. Overall, this is a high quality and innovative study. However, the reported effects are rather small and so whether ubiquitination is a key off/on switch to trigger ER phagy or just a tuning mechanism is not clear.

I have three main points: (1) although supported by convincing statistical analysis and numerous replicates (both in experiments and simulations), the reported effects are small: (in the range of 1.5-fold change in figures 1g, 2b, 3c-e-g) and are observed after extensive mutagenesis of FAM134B where all exposed lysines (17 residues) were mutated into arginines. (2) the molecular basis is unclear: direct interaction of Ub with the bilayer? Ub-Ub cis and trans interactions; if so why all possible ubiquitination sites would behave similarly? (3) There is no direct experimental approach to test the effect of ubiquitination on FAM134B clustering and curvature induction (in marked contrast to a previous paper by the same group, where they could reconstitute the protein into liposomes and analyzed membrane fragmentation by EM).

Other points:

(4) The western blots shown in panel f of figure 1 is key to understand the initial reasoning of the authors. I would suggest this panel to be scale up, perhaps at the expense of panels b and c, which could be smaller. I have some problems with this experiment. The two left lanes (1,2) show HA-IP, which allows the authors to capture HA-FAM134B (with or without the 17 KR mutations), whereas the two right lanes (5,6) shows mycIP, which allows the authors to capture ubiquitinated proteins. The comparison between 5 and 6 shows a dramatic decrease in the fraction of ubiquitinated protein upon the 17KR mutation. However, this decrease is much more moderated when looking at the oligomer SDS resistant fraction of HA-FAM134B (lanes 2 vs 1). Does it mean that the SDS resistant fraction is poorly correlated with ubiquitination? The statistics of the experiment seems OK (as quantified in panel f), both for oligomerization and LC3BII binding. However, the overall effect is very modest (1.32 and 1.5-fold decrease, respectively), despite the fact that 17KR mutations were needed. Note that the LC3B low exposure and high exposure immunoblots might have been inverted.

(5) The coarse-grained MD simulations of FAP134B with one Ub at position K160 suggest that « the ubiquitin-moiety pulled on the conserved linker amphipathic helix (AHL) to maximise interactions with the bilayer (Extended Data Fig. 2a-b). ». This sentence is obscure: I don't understand on which lever the Ub chain can act to exert a mechanical effect on the amphipathic helix.

(6) the number of contacts that the ubiquitinated chain can undergo with the POPC bilayer (in the 100-300 range) is intriguing. Is it known whether Ub can partition to liposomes (even when not connected to a transmembrane protein)? More generally, the MD data shown in the extended Figure 2 suggest many different interaction types of Ub (cis and trans interactions, lipid membrane binding...). It would be important to experimentally validate these interactions. For instance, the fact that the hydrophobic face displayed significant contacts with POPC lipids, should be tested with liposome-based assays (e.g. using NBD labelling or H/D exchange). Perhaps more important are the trans Ub interactions suggested by the simulations of figure 2 e and f. If the model is correct, such interactions should occur whatever the protein holding the Ub moieties as long as the Ub chains have some mobility. Would it be possible to conduct experiments with an artificial construct to mimic this general scenario?

(7) In Figure 2, the effect of FAM134B ubiquitination on the rate of curvature induction (using the bicelle to vesicle model) appears quite modest (factor in the range of $\times 1.4$). Can the authors comment on this factor. In a previous paper, (Bakhara et al Nat Comm 2019 ; figure 3), the authors convincingly shown a huge difference between a model transmembrane peptide and FAM134B. Could it be that any transmembrane protein asymmetrically decorated with Ub promotes membrane deformation? Do the authors favor a model similar to the one proposed by Derganc/Copic/Miller/Stachowiak where membrane curvature is induced by protein crowding and increases with the asymmetric mass distribution of the protein (see e.g. PMID: 23999615, PMID: 26969088, PMID: 22300850)?

(8) In Figure 2c, the control « empty » (red) to calibrate the curvature preference of FAM134B, whether ubiquitinated or not, is obscure. The reader has to refer to a previous figure in Bakhara et al (figure 3) to understand the panel.

(9) In figure 2d, the authors noticed: « By contrast, upon ubiquitination, the two RHD molecules diffused slowly towards each other, but formed a tight cluster at the top of the buckle and remained bound for the entirety of the simulation (up to 25 μ s). » Could the authors discuss the basis of this surprising speed reduction? Does Ub create some frictional effect?

(7) Extended Figure 6. I don't understand the sentence in the legend: « The two Ub moieties (blue and purple) provide volume exclusion and break the symmetry of the bilayer predominantly driving the spontaneous budding and coating the RHD clusters. ». Please clarify what you mean by breaking the symmetry of the bilayer and coating the RHD clusters.

(8) In figure 2f, the authors mention that « Ubiquitination of K160 + 144 K264 promoted sorting and clustering of all the RHDs onto the nascent bud ». From what I understood, this simulation also includes non-ubiquitinated RHD. Does this mean that even these RHD become included in the clusters and in the membrane buds and if so, what is the proposed molecular basis for this effect?

(9) Please show the inset for Figure 4c.

(10) One strong point of this manuscript is the identification of the E3 ligase AMFR for FAM134B ubiquitination (figure 5). However, the corresponding final scheme (figure 5i) suggests that the two proteins are present in stoichiometric amounts, which is probably not the case, isn't? In addition, I don't understand what the arc membrane decorated with orange proteins represents in the panel.

Referee #2 (Remarks to the Author):

Recent studies have revealed that multiple autophagy receptors mediate selective degradation of

the ER via autophagy (ER-phagy) in different physiological or pathological situations. However, whereas protein ubiquitination plays central roles in many other selective autophagy pathways, including mitophagy, pexophagy, lysophagy, and xenophagy, its involvement in ER-phagy remains poorly understood. In this study, the group of Dr. Ivan Dikic, who discovered FAM134B as the first ER-phagy receptor, showed that the ubiquitination of this protein promotes the formation of protein clusters containing other FAM134 paralogs and membrane-shaping proteins, leading to an increase in FAM134B binding to the autophagosomal protein LC3. Molecular dynamics simulations suggests that ubiquitination causes a conformational change in FAM134B that enhances its membrane remodeling activity and that inter-ubiquitin interactions facilitate the clustering of ubiquitinated FAM134B. In addition, the authors identified AMFR as an E3 ligase involved in FAM134B ubiquitination and required for ER-phagy. Thus, this study showed that protein ubiquitylation regulates ER-phagy by increasing the membrane remodeling function and clustering of the ER-phagy receptor FAM134B in a way quite different from that in other selective autophagy pathways. However, the authors should address the following issues to more convincingly draw their conclusions.

Specific comments:

1. The authors showed that arginine replacement of all the lysine residues found to be ubiquitinated was not sufficient to abolish overall FAM134B ubiquitination. However, it is not described whether this mutant was defective in ER-phagy. The authors should clarify this point to conclude that the position of ubiquitination is important or not for FAM134B regulation.
2. In this study, the authors evaluated ER-phagy flux based on the lysosomal transport of FAM134B itself but should also examine that of other ER proteins such as CLIMP63 to more convincingly draw their conclusions.
3. The 17KR mutation might affect the basic functions of FAM134B, such as its LC3 binding and membrane remodeling activity, rather than their enhancement via ubiquitination. The authors should exclude this possibility by confirming that these functions of FAM134B are not affected by the 17KR mutation using recombinant proteins. It will also be great if the authors could reconstitute the enhancement of liposome fragmentation activity of FAM134B by its AMFR-mediated ubiquitination.
4. Although how ubiquitination regulates FAM134B functions and behaviors was suggested by MD simulations, experimental validation would make them more reliable. Could the authors specify residues involved in interactions between ubiquitin and FAM134B as well as those between ubiquitin moieties on FAM134B? If possible, mutations into these residues may allow the authors to validate their models suggested by MD simulations.
5. MD simulations suggested interactions between ubiquitin moieties on FAM134B. Has this type of ubiquitin-ubiquitin interaction been described in previous studies? Is this interaction possible because ubiquitins are located at specific positions in FAM134B?
6. Fig. 3d: It would be interesting if the authors could estimate how many FAM134B molecules reside in microscale clusters.
7. Fig. 3e: Does the enlargement of FAM134B clusters upon Torin1 treatment depend on autophagosome formation? This will provide another insight into the mechanism of FAM134B clustering.
8. Extended Data Fig. 7g: The results seem to show that FAM134B resides in the tubular ER inconsistent with their original report that it localizes to ER sheets.
9. Examining the size and shape of ER fragments within autophagosomes in FAM134B 17KR cells by electron microscopy may add important information how FAM134B ubiquitination affects the

process of ER fission.

10. Fig. 4d: The LC3B peak at distance of ~5 microns is FAM134B-positive and ubiquitin-negative. Does this suggest that ER-phagy can proceed without FAM134B ubiquitination?

11. The authors identified many proteins including RHDs and RHD-like proteins by mass spectrometry of immuno-purified FAM134B clusters. However, it is not convincing to me that all of these proteins are associated via direct interactions. Instead, some of these proteins might be concentrated in the clusters due to other reasons such as their preferring high membrane curvature but co-isolated with FAM134B due to incomplete membrane solubilization. This point should be discussed in the text.

12. Fig. 5: The authors should examine FAM134B ubiquitination in cells simply knocked out or knocked down for AMFR. This is important to evaluate how significantly endogenous AMFR contributes to FAM134B ubiquitination.

13. Extended Data Fig. 1c and d: A decrease in the protein level of FAM134B is not completely blocked by BafA1. This might suggest that ubiquitinated FAM134B is also degraded by the proteasome. Is this true? Given this possibility, it would be necessary to confirm that TAK243 indeed blocks autophagic degradation of FAM134B rather than its proteasomal degradation.

14. Extended Data Fig. 9g and 9h and Fig. 5f and 5g: The authors should confirm that these changes in FAM134B levels were due to degradation via ER-phagy rather than proteasomal degradation.

15. Fig. 5f and Extended Data Fig. 9i: The authors should show that a decrease in AMFR levels caused by Torin 1 treatment depends on ER-phagy.

16. The authors seem to assume that FAM134B is mono-ubiquitinated. Is there any evidence for it?

17. Lines 209-211 "dimeric FAM134B (observed using BiCAP) was found in ER fragments" and lines 250-251 "FAM134B-mediated ER-fragmentation in response to Torin 1 was also reduced": How could the authors determine that FAM134B puncta represent ER fragments? These puncta might represent FAM134B clusters in the ER network.

18. Fig. 4c: Arrows seem not to be correctly positioned.

19. Fig. 5i should be cited in the text. It would also be better if the authors could depict the interaction between FAM and LC3B in this model.

Author Rebuttals to Initial Comments:

Referees' comments:

Referee #1 (Remarks to the Author):

Following previous work showing the contribution of the transmembrane protein FAM134B in shaping the ER and in triggering ER phagy, the Dikic and Hummer labs propose here that ubiquitination of FAM134B contributes to its role as an ER phagy receptor by boosting FAM134B clustering. The authors use two main complementary approaches: high resolution light microscopy assays coupled to protein engineering to evaluate FAM134B ubiquitination and clustering; coarsed-grain simulations to predict the impact of FAM134B ubiquitination on membrane dynamics. Overall, this is a high quality and innovative study. However, the reported effects are rather small and so whether **ubiquitination is a key off/on switch to trigger ER phagy or just a tuning mechanism is not clear.**

We thank the reviewer for her/his overall evaluation. In the revised version, we have highlighted the role of FAM134B ubiquitination as a tuning mechanism required for efficient ER-phagy. We have provided multiple lines of evidence to support this hypothesis, including *in vitro*, computational and *in vivo* data showing the functional role of RHD ubiquitination in ER-remodelling proteins that underlie the mechanistic explanation of this process. Detailed experimental approaches are described below.

(1) although supported by convincing statistical analysis and numerous replicates (both in experiments and simulations), the reported effects are small: (in the range of 1.5-fold change in figures 1g, 2b, 3c-e-g) and are observed after extensive mutagenesis of FAM134B where all exposed lysines (17 residues) were mutated into arginines.

We thank the reviewer for this critical comment. As now stated throughout the manuscript, Ub is not an on/off signal but rather an important facilitator of ER-phagy flux. We consider this question central to our revisions and multiple additional experiments were carried out to test the statistical and quantitative differences triggered by multiple Ub signals in this pathway. We also investigated the structural components of FAM134B membrane anchoring as well as using the FAM134B 17K mutant. The additional experiments are summarized below:

Referring to previous Figure 1g. We analysed the effect of different FAM134B-RHD structural elements on LC3B-II binding and the formation of SDS-resistant oligomers (Rebuttal letter Fig. R1). The deletion of single transmembrane hairpin segments reduced the formation of oligomers (Fig. R1a, lanes 12, 13) and LC3B-II binding (Fig. R1a, lane 12) similarly in extent to 17KR (Fig. 1f and g). Only the removal of both segments ($\Delta TM12 + \Delta TM34$) abolished the formation of oligomers (Fig. R1a, lane 14). This construct does not localize in the ER (Bhaskara, et. al., 2019) but still binds LC3B-II (Fig. R1a, lane 14). Both hairpins are essential for maximal membrane shaping and ER-phagy activity (Bhaskara, et. al., 2019). Interestingly, deletion of TM1-2 (Fig. R1b lane 12) or both segments (Fig. R1b lane 14) reduced the ubiquitination levels of FAM134B.

Figure R1: Abundance of oligomeric species (Left Panel) and ubiquitination levels (right panel) of FAM134B upon the deletion of different FAM134B-RHD structural element.

Based on these comparisons, we suggest that the reported differences between WT and 17KR in Fig. 1g are sufficient to have significant and strong effects on ER-phagy and membrane remodelling. Moreover, we tested the formation of SDS-resistant oligomers by different FAM134B lysine mutants (17KR, 13KR and 8KR). We observed a gradual decrease in the abundance of oligomers depending on ubiquitination levels (Rebuttal Letter Fig. R2).

Figure R2: Left Panel: representative western blot analysis of a TUBE2 pull-down assay showing ubiquitination levels of FAM134B-WT, 8KR, 13KR and 17KR. Right Panel: Abundance of oligomeric FAM134B species of FAM134B-WT, 8KR, 13KR and 17KR.

We used chemical cross-linking to determine the oligomeric state of FAM134B-WT and the 17KR mutant in cell membranes. A cross-linked band (~165 kDa) was detected and is consistent with the SDS-resistant oligomers showed in Fig. 1f (main text). Again, high-molecular-weight species of FAM134B were less abundant in the HA-FAM134B-17KR mutant compared to WT (Extended Data Fig. 1k).

Referring to previous Figure 2b. We performed a new set of bicelle-to-vesicle transition simulations at 280 K (originally performed at 300 K) to show the effect of ubiquitination. We found that the bicelles containing FAM134B-RHD alone and with K160-Ub did not effectively transit to closed vesicles. Only 4/20 and 5/20 bicelles, respectively, could undergo complete vesiculation within 1 (μ s), indicating that the barrier to induce curvature in the bicelle and subsequent vesiculation of the bicelle at 280 K is much higher (Extended Data Fig. 3h and 3k). By contrast, for bicelles containing K264-Ub and (K160+K264)-Ub, all 20 runs initiated at 280 K formed closed vesicles, showing that the K264-Ub and bi-Ub variants easily overcome the barrier even at 280 K (Extended Data Fig. 3i and 3j). Together with previous simulations at 300 K, these new simulations at 280 K shed light on the dynamic nature of ubiquitinated RHDs and their ability to break bicelle asymmetry and induce curvature.

Referring to previous Figures 3b and 3c. We performed new assays to show that ER-phagy is reduced when cells express the 17KR mutant. We generated monoclonal U2OS cell lines co-expressing ssRFP-GFP-KDEL (Chino *et al.*, 2019) with either HA-FAM134WT or HA-FAM134B-17KR under the control of doxycycline-inducible promoters. We observed a general reduction in ER-phagy flux in the presence of FAM134B-17KR (Extended Data Fig. 8g-j). REEP5 degradation triggered by EBSS was also reduced in cells expressing this mutant (Extended Data Fig. 8j). These results confirm that FAM134B-RHD ubiquitination is important for FAM134B-mediated ER-phagy.

Referring to previous Figures 3e and 3g. Additionally, we inferred molecule numbers in nanoscale clusters by the kinetic analysis of single-molecule DNA-PAINT data and found that ubiquitination favours the oligomeric state of FAM134B in nanoclusters on **average five-fold** ($n_{17KR} = 2.5$, $n_{WT} = 12.6$), suggesting that ubiquitination promoted the assembly of high-density clusters (Fig. 3h).

(2) the molecular basis is unclear: direct interaction of Ub with the bilayer? Ub-Ub cis and trans interactions; if so why all possible ubiquitination sites would behave similarly?

This is a central aspect of our new discovery. Our evidence indicates that the ability of ubiquitin to interact with multiple low-affinity binding sites may determine the avidity factor and stabilise clustered complexes. This means that in Ub attachments at multiple sites in RHD proteins may concentrate such interactions in the 2D plane and make them physiologically relevant.

The number of ubiquitination sites is relevant not only on one ER-phagy receptor but across the entire receptor cluster incorporating multiple ubiquitinated RHD proteins. The molecular effects of Ub on the RHD can be explained at various levels, from the perturbation of RHD structure to volume-exclusion and non-specific crowding and clustering effects of the RHDs.

1. At the level of RHD structure, adding Ub at K160 and K264 increases the RHD membrane footprint on the cytosolic side. We measured the radius of gyration of ubiquitinated and non-ubiquitinated FAM134B-RHD to highlight this difference, revealing that the K264-Ub variant occupies a single peak radius (1.8 nm) close to the non-ubiquitinated RHD (1.7 nm). However, K160-Ub occupies a slightly larger radius of gyration (1.9 nm) with a substantial shoulder at 2.1 nm, indicating that at least two significant conformational states are populated in bilayers. By contrast, bi-Ub-RHD displays the largest radius of gyration (2.2 nm) (Extended Data Fig. 2g and 2h).
2. Increasing the membrane footprint of the RHD by ubiquitination could enhance the shape of the asymmetric wedge, thus enhancing the local induction of membrane curvature. Using bicelle-to-vesicle simulations (Fig. 2b; Extended Data Fig. 3), we measured the capacity of ubiquitinated RHDs to induce local curvature and to close the bicelles. Two effects determine the rate of bicelle-to-vesicle closure:
 - a. Faster transmembrane hairpin dynamic interactions lead to enhanced wedging, directly bending the membrane and leading to bicelle closure.
 - b. The enhanced asymmetric footprint disallows large membrane fluctuations and bending along one side of the bicelle, providing directionality to the curvature induction process.
3. Ub mediates several non-specific interactions due to proximity and bonded constraints imposed by physically tethering Ub to the RHD molecule close to the bilayer interface:
 - a. Ub linked to K264 and, more specifically, to K160, is geometrically restrained to locations close to the bilayer and therefore mediates more interactions with the bilayer. Hydrophobic residues mediate these enhanced Ub-POPC interactions in our model at the N-terminal of the Ub moiety (Extended Data Fig. 2a,b).
 - b. Ub also contacts the cytosolic loop residues of the RHD primarily due to the proximity imposed by covalent linkage (Extended Data Fig. 2c,d).
 - c. In the case of multiple mono-Ub-RHDs (e.g., (K160+K264)-Ub-RHD), the two Ub moieties display fewer contacts with the bilayer but still enhance intra-molecular (cis) Ub-Ub interactions (Extended Data Fig. 2e,f). Residues mediating these interactions are characterised by their contact maps (Extended Data Fig. 6). We quantified the residues involved in these contacts and mapped them to the Ub sequence and structure. On average, the number of intra-molecular

- contacts across K160-Ub and K264-Ub is high. More specifically, residues from the β -12 hairpin, β -34 hairpin, and the terminal β 5 strand display intra-molecular contacts (Extended Data Fig. 7).
- d. The frequency of pairwise residues involved in contact along the entire length of the trajectory. This contact map is not symmetric due to the linkage of the two Ubs to different lysine residues (Extended Data Fig. 6).
 4. Ub mediates clustering and oligomerisation of RHDs. Non-specific and low-affinity interactions mediated by Ub allow close interactions between RHDs and form nano-sized clusters. A detailed analysis of bound states indicated that almost the entire surface of Ub was involved in these weak and non-specific interactions. By mapping the contacts onto the Ub sequence and structure, we found that the same β -12 hairpin and β -34 hairpin residues showed a slight increase in inter-molecular or trans-Ub-Ub interactions (Extended Data Fig. 6).
 5. Ub-mediated RHD clusters are also formed by altering simulation conditions. We ran the simulations with modified forcefield parameters to demonstrate that the Ub-Ub interactions are weak and non-specific. We scaled the ϵ parameter of the nonbonded interactions, dictating the strength of protein-protein interactions, from $\alpha=1.0$ to $\alpha=0.65$ (Zakaraya et al., 2021) to alter RHD interaction dynamics. By altering the simulation conditions ($\alpha=0.65$), we observed that the RHD clusters still formed. Again, almost the entire surface of Ub mediated trans-interactions, indicating that multiple low-affinity interactions led to cluster formation. We also found slightly more trans-Ub-Ub interactions among the residues forming H2 and H3, indicating that these interactions were not specific to the β -12 hairpin and β -34 hairpin residues found in simulations at $\alpha=1.0$ (Fig. 2d and Extended Data Fig. 6).
 6. The Ub-Ub contact maps capture the character of the intra-molecular and inter-molecular interactions. By changing the simulation conditions (PPI strength from $\alpha=1.0$ to $\alpha=0.65$) and membrane asymmetry ($\Delta N = 0$ to $\Delta N = 300$), we can test what drives the clustering and stabilisation of the RHD clusters. We found that the intra-molecular Ub-Ub contact map does not change under different simulation conditions, indicating that these contacts are primarily driven by constrained geometry and proximity of the two Ub moieties. By contrast, the inter-molecular Ub-Ub contact map changes. At $\alpha=0.65$, we observed a slight change in the residues showing the maximum number of contacts (H2 and H3 vs β -12 hairpin and β -34 hairpin), indicating that Ub-Ub interactions reorganise to compensate for the loss of multiple weak interactions, making them non-specific. Further, in simulations at $\Delta N = 300$, we found that the contacts became more prominent, indicating that the curvature induced at $\Delta N = 300$ increased the lifetime and stabilized the RHD-bound states within the clusters. (Fig. 2d and Extended Data Fig. 6).
 7. Nonspecific Ub-Ub interactions create a crowded membrane environment with multiple RHDs, further increasing the propensity of proteins to sort and aggregate locally in the membrane. These crowded regions appear to be driven by volume-exclusion effects and curvature-mediated protein sorting mechanisms.
 8. Steric hindrance or impediment of ubiquitinated-RHDs is expected to exclude other macromolecules from the neighbourhood, creating a volume-exclusion zone. However, in crowded environments with increasing concentrations of ubiquitinated RHDs, the volume of solution available to the molecules is restricted to the space from which they are not excluded. This decreases the random distribution of particles, thus reducing the entropy of the crowded solutions. The excluded volume around each Ub-RHD dimer is smaller than twice that of each monomeric Ub-RHD. To keep the overall entropy as high as possible, the system tends to minimise the total excluded volume by favouring the formation of dimers (or high-order oligomers and aggregates) in the crowded milieu. Furthermore, non-specific trans-Ub-Ub interactions and the membrane curvature stabilise these clusters in the crowded membranes.
 9. Volume-exclusion effects are seen in our simulations, along with low-affinity non-specific Ub-Ub interactions that nucleate RHD clusters, which then induce membrane bud formation. The high curvature of the membrane bud stabilises the RHD clusters, increases their longevity, favouring the sorting of individual Ub-RHDs to the site of the bud.

Supporting evidence for these statements is provided in new Figure 2, as well as Extended Figures 2, 5, 6 and 7.

(3) There is no direct experimental approach to test the effect of ubiquitination on FAM134B clustering and curvature induction (in marked contrast to a previous paper by the same group, where they could reconstitute the protein into liposomes and analyzed membrane fragmentation by EM).

As suggested by the reviewer, we have now performed the liposome remodelling assay and the results have further strengthened our evidence that ubiquitination enhances FAM134B activity.

We expressed and purified N- and C- terminal fusions of Ub to FAM134B-RHD₉₀₋₂₆₄ (described as Ub-RHD-Ub) and as a control we expressed RHD without ubiquitin (RHD₉₀₋₂₆₄). Using these constructs, we conducted *in vitro*

liposome remodelling assays. Ub-RHD-Ub led to smaller proteoliposomes with a narrow size distribution compared to the RHD without Ub, indicating a significant gain of membrane-remodeling activity for the chimera Ub-RHD-Ub (Fig. 2c).

We also achieved the *in vitro* ubiquitination of full-length GST-FAM134B by AMFR. Using proteomics analysis, we detected ubiquitination at K160, K278 and K299. Compared to liposomes treated with GST, the non-ubiquitinated GST-FAM134B (in the presence of AMFR, no ATP) decreased the liposome diameter, but ubiquitinated GST-FAM134B (in the presence of AMFR+ATP) reduced the diameter even further (Fig. 5e). This significant difference between non-ubiquitinated and ubiquitinated samples suggests that multiple ubiquitination sites on full-size FAM134 are targeted by AMFR, and ubiquitination promotes the formation of smaller liposomes.

Other points:

(4) The western blots shown in panel f of figure 1 is key to understand the initial reasoning of the authors. I would suggest this panel to be scale up, perhaps at the expense of panels b and c, which could be smaller. I have some problems with this experiment. The two left lanes (1,2) show HA-IP, which allows the authors to capture HA-FAM134B (with or without the 17 KR mutations), whereas the two right lanes (5,6) shows mycIP, which allows the authors to capture ubiquitinated proteins. The comparison between 5 and 6 shows a dramatic decrease in the fraction of ubiquitinated protein upon the 17KR mutation. However, this decrease is much more moderated when looking at the oligomer SDS resistant fraction of HA-FAM134B (lanes 2 vs 1). Does it mean that the SDS resistant fraction is poorly correlated with ubiquitination?

The statistics of the experiment seems OK (as quantified in panel f), both for oligomerization and LC3BII binding. However, the overall effect is very modest (1.32 and 1.5-fold decrease, respectively), despite the fact that 17KR mutations were needed. Note that the LC3B low exposure and high exposure immunoblots might have been inverted.

We thank the reviewer for this suggestion. We corrected LC3B immunoblot exposure (Fig. 1f).

On page 1, we provided data to explain how different FAM134B-RHD structural elements affect the formation of SDS-resistant oligomers and binding to LC3B-II.

We also tested the formation of SDS-resistant oligomers by different FAM134B lysine mutants (17KR, 13KR and 8KR). Ubiquitination levels and SDS-resistant oligomers were only strongly reduced in the 17KR mutant (Fig. R2). In Figure 4e (main text) we showed that the clustering of FAM134B-17KR led to reduced interactions with other RHD-containing proteins, ATG8s and the ubiquitination machinery. Western blots confirmed that 17KR reduced the interaction of FAM134B with endogenous ARL6IP1 (Fig. R3 lane 2 compared to lane1). ARL6IP1 is a RHD-like protein that is recruited to ER-phagy receptor complexes to facilitate membrane remodelling. Interestingly, similar effects were observed following the deletion of FAM134B transmembrane segments (Fig. R4, lanes 5 and 6), which are constructs without maximal membrane shaping and remodelling activity (Bhaskara, et. al., 2019). These data suggest that FAM134B-RHD ubiquitination is not only required for self-interaction but also to form ER-phagy receptor supra-complexes. Based on these observations and correlations, we proposed that the differences between WT and 17KR in Figures 1g and 4e are sufficient to have significant and strong effects on ER-phagy and membrane remodelling.

Figure R3:
Co-immunoprecipitation of endogenous ARL6IP1 with WT and 17KR FAM134B

Figure R4:
Co-immunoprecipitation of endogenous ARL6IP1 with FAM134B, upon the deletion of different FAM134B-RHD element

(5) The coarse-grained MD simulations of FAM134B with one Ub at position K160 suggest that « the ubiquitin-moiety pulled on the conserved linker amphipathic helix (AHL) to maximise interactions with the bilayer (Extended Data Fig. 2a-b). ». This sentence is obscure: I don't understand on which lever the Ub chain can act to exert a mechanical effect on the amphipathic helix.

K160-Ub is bonded to the linker segment, proximal to the AHL, and flanks it on its N-terminus. Interactions between Ub and the POPC bilayer (Extended Data Fig. 2) directly affect the organisation of the AHL. We found that when

Ub increased contact with the lipid bilayer, it affected the proximal AH_L and altered its relative depth and orientation to the rest of the RHD.

(6) the number of contacts that the ubiquitinated chain can undergo with the POPC bilayer (in the 100-300 range) is intriguing. Is it known whether Ub can partition to liposomes (even when not connected to a transmembrane protein)? More generally, the MD data shown in the extended Figure 2 suggest many different interaction types of Ub (cis and trans interactions, lipid membrane binding...). It would be important to experimentally validate these interactions. For instance, the fact that the hydrophobic face displayed significant contacts with POPC lipids, should be tested with liposome-based assays (e.g. using NBD labelling or H/D exchange). Perhaps more important are the trans Ub interactions suggested by the simulations of figure 2 e and f. If the model is correct, such interactions should occur whatever the protein holding the Ub moieties as long as the Ub chains have some mobility. Would it be possible to conduct experiments with an artificial construct to mimic this general scenario?

We tested the role of Ub-Ub interactions by reducing the protein-protein interaction strength in our simulations. At $\alpha=0.65$, although there were fewer trans-Ub-Ub contacts in total, they could still induce the formation of RHD clusters. By contrast, at $\alpha=1.0$, the interactions were non-specific and spread throughout the Ub surface. Low-affinity interactions mediated by ubiquitin primarily allowed close RHD interactions, leading to the formation of nano-sized clusters and nucleation of the membrane bud (at $\Delta N=300$). However, by reducing the strength of the interactions even at $\Delta N=300$, we found only transient cluster formation but no bud formation, indicating that Ub-Ub interactions, although non-specific, are needed to stabilise dimers and higher-order oligomers to induce membrane budding and large-scale remodelling in the ER (Fig. 2d, columns 3 & 4, and Extended Data Fig. 6).

Testing and validation of Ub-mediated interactions by direct mutagenesis experiments is challenging, especially when almost the entire surface of Ub is involved in non-specific interactions. Changes in Ub structure can have unintended effects on the entire quality control process. Therefore, we tested the role of Ub by making chimeric RHDs. We added two Ub molecules flanking the RHD₉₀₋₂₆₄ to mimic the ubiquitinated RHDs. We found that they could remodel liposomes into smaller vesicles at the same concentration by inducing stronger curvature (Fig. 2c) compared to native RHD domains alone. This indicated that the addition of Ub close to the interfacial region of the bilayer could mediate strong interactions with the bilayer and with other molecules, favouring the formation of higher-order oligomers essential to remodel liposomes.

(7) In Figure 2, the effect of FAM134B ubiquitination on the rate of curvature induction (using the bicelle to vesicle model) appears quite modest (factor in the range of x 1.4). Can the authors comment on this factor. In a previous paper, (Bakhara et al Nat Comm 2019 ; figure 3), the authors convincingly shown a huge difference between a model transmembrane peptide and FAM134B. Could it be that any transmembrane protein asymmetrically decorated with Ub promotes membrane deformation? Do the authors favor a model similar to the one proposed by Derganc/Copic/Miller/Stachowiak where membrane curvature is induced by protein crowding and increases with the asymmetric mass distribution of the protein (see e.g. PMID: 23999615, PMID: 26969088, PMID: 22300850)?

We used an *in silico* bicelle-to-vesicle transition assay to measure the capacity of ubiquitinated RHDs to induce local curvature and to transit open flat bicelles into closed vesicles. It is important to highlight that two effects are observed in these simulations:

1. The sign of the curvature induced in the bicelle (positive or negative) and the number of vesicles undergoing complete closure within the simulation time.
2. The waiting times to reach a threshold curvature denoting closure of the bicelle into a vesicle. These waiting times are Poisson distributed, and their CDF can be fitted to a single exponential with lag times to extract rates of vesicle formation.

The above effects are determined by the internal dynamics of the protein inclusion in the bicelle and its overall asymmetric footprint on the two sides of the bicelle.

- a. Faster Inter-transmembrane-hairpin dynamics leading to enhanced wedging directly bend the membrane and thus promoting bicelle closure.
- b. Enhanced asymmetric footprint, preventing large membrane fluctuations and bending along one leaflet of the bicelle, providing strong directionality to the curvature induction process.

Previously, we compared the dynamic RHD-induced vesicle formation rates to the rates of vesicle formation from empty bicelles and bicelles with symmetric KALP-peptide (Bhaskara et al., 2019). Here, we compare Ub-RHD variants against the RHD alone. This shows a modest acceleration in vesicle closure rates.

In the current manuscript, we demonstrate that ubiquitination of the already asymmetric RHD further increases its asymmetric membrane footprint (Extended Data Fig. 2g,h). Although this leads to a moderate increase in local curvature induction as shown above in our bicelle-to-vesicle simulations, it has enormous consequences in the clustering and oligomerisation of RHDs, especially in crowded membranes. Adding multiple Ub to RHDs on the cytosolic side creates many non-specific interaction patches on the RHD, favouring dimerisation and forming higher-order oligomeric receptor clusters (Fig. 2d and Extended Data Fig. 4e). The volume-exclusion effects

further drive this process under the crowded membrane environment creating a large membrane perturbation and subsequent budding.

It is already known that reticulons such as FAM134B-RHDs are effective curvature generators because they combine a wedge-shaped hydrophobic structure with oligomerisation, inducing significant local curvature without occupying a large amount of the membrane surface (Bhaskara et al., Nat Commu. 2019; Stachowiak et al., Nat Cell Biol 2013). Ubiquitination of the RHD causes volume exclusion effects to drive this process, favouring the formation and stabilisation of otherwise short-lived clusters. This allows the clusters to grow in a crowded environment into even larger macro-sized clusters compatible with the recruitment of autophagic machinery (e.g., phagophore-embedded LC3C engagement).

(8) In Figure 2c, the control « empty » (red) to calibrate the curvature preference of FAM134B, whether ubiquitinated or not, is obscure. The reader has to refer to a previous figure in Bhaskara et al. (figure 3) to understand the panel.

We thank the reviewer for pointing this out, and we have rewritten the legend to improve clarity.

(9) In figure 2d, the authors noticed: « By contrast, upon ubiquitination, the two RHD molecules diffused slowly towards each other, but formed a tight cluster at the top of the buckle and remained bound for the entirety of the simulation (up to 25 μ s). » Could the authors discuss the basis of this surprising speed reduction? Does Ub create some frictional effect?

We thank the reviewer for this remark. In general, the diffusion of proteins in curved membranes, more specifically in highly viscous and dense bilayers, is much slower than in flat bilayers. To determine if there is any frictional effect, we estimated the lateral self-diffusion coefficients of the FAM134B-RHD and its ubiquitinated variants. We found that K160-Ub-FAM134B diffused slowly compared to the RHD alone, whereas K264-Ub and (K160+K264)-bi-Ub variants diffused slightly faster. We reason that the membrane interactions of Ub in the K160-Ub-FAM134B-RHD variant slowed it considerably, thus producing a slight frictional effect. However, the Ub in the K264-Ub and (K160 + K264)-Ub-FAM134B-RHD variants interact primarily with the cytosolic loops and diffuse faster in comparison to the RHD alone.

Protein	Lateral Diffusion Coefficient ($\times 10^5$ cm ² /s)	Remark (Relative to FAM134B-RHD)
FAM134B-RHD	0.0003161 \pm 0.006916	1.00
K160-Ub-FAM134B-RHD	0.0002192 \pm 0.004679	0.69 (slow)
K264-Ub-FAM134B-RHD	0.0075910 \pm 0.017430	24.01 (fast)
(K160 + K264)-Ub-FAM134B-RHD	0.0091770 \pm 0.000492	29.03 (fast)

Figure 5

(10) Extended Figure 6. I don't understand the sentence in the legend: « The two Ub moieties (blue and purple) provide volume exclusion and break the symmetry of the bilayer, predominantly driving the spontaneous budding and coating the RHD clusters. ». Please clarify what you mean by breaking the symmetry of the bilayer and coating the RHD clusters.

We have rewritten this figure legend to improve clarity (and added a further explanation in main text Fig. 2).

(11) In figure 2f, the authors mention that « Ubiquitination of K160 + 144 K264 promoted sorting and clustering of all the RHDs onto the nascent bud ». From what I understood, this simulation also includes non-ubiquitinated RHD. Does this mean that even these RHD become included in the clusters and in the membrane buds and if so, what is the proposed molecular basis for this effect?

No, the current stimulation does not include non-Ub-RHDs (Fig. 2d and Extended Data Fig. 6). All nine molecules in the simulation box have Ub modelled at positions K160 and K264. We have already shown that the RHDs alone can also induce the formation of oligomers in curved bilayers (Bhaskara et al., 2019). Furthermore, at high concentrations, RHDs alone can induce clustering and nucleate membrane buds from asymmetric bilayers (Siggel et al., 2021). Here we demonstrate that the strength of non-specific Ub-Ub interactions is critical for stabilising clusters, enhancing their longevity, and promoting the nucleation of membrane buds and their subsequent coating.

(12) Please show the inset for Figure 4c.

The inset is shown at the bottom of the image panel (now in main text Fig. 4b).

(13) One strong point of this manuscript is the identification of the E3 ligase AMFR for FAM134B ubiquitination (figure 5). However, the corresponding final scheme (figure 5i) suggests that the two proteins are present in stoichiometric amounts, which is probably not the case, isn't? In addition, I don't understand what the arc membrane decorated with orange proteins represents in the panel.

We modified the model as suggested. In the model, FAM134B and AMFR are present at different ratios forming part of the cluster. Additionally, our BiCAP assay data suggest a 1:1 ratio is the minimal interaction required to form a complex. Further studies can determine the stoichiometry of FAM134B clusters and address how membrane proteins (enzyme or a substrate) co-operate to drive ER-phagy. The current scheme is shown in main text Fig. 5f.

Referee #2 (Remarks to the Author):

Recent studies have revealed that multiple autophagy receptors mediate selective degradation of the ER via autophagy (ER-phagy) in different physiological or pathological situations. However, whereas protein ubiquitination plays central roles in many other selective autophagy pathways, including mitophagy, pexophagy, lysophagy, and xenophagy, its involvement in ER-phagy remains poorly understood. In this study, the group of Dr. Ivan Dikic, who discovered FAM134B as the first ER-phagy receptor, showed that the ubiquitination of this protein promotes the formation of protein clusters containing other FAM134 paralogs and membrane-shaping proteins, leading to an increase in FAM134B binding to the autophagosomal protein LC3. Molecular dynamics simulations suggest that ubiquitination causes a conformational change in FAM134B that enhances its membrane remodeling activity and that inter-ubiquitin interactions facilitate the clustering of ubiquitinated FAM134B. In addition, the authors identified AMFR as an E3 ligase involved in FAM134B ubiquitination and required for ER-phagy. Thus, this study showed that protein ubiquitylation regulates ER-phagy by increasing the membrane remodeling function and clustering of the ER-phagy receptor FAM134B in a way quite different from that in other selective autophagy pathways. However, the authors should address the following issues to more convincingly draw their conclusions.

Specific comments:

1. The authors showed that arginine replacement of all the lysine residues found to be ubiquitinated was not sufficient to abolish overall FAM134B ubiquitination. However, it is not described whether this mutant was defective in ER-phagy. The authors should clarify this point to conclude that the position of ubiquitination is important or not for FAM134B regulation.

We mutated the lysine residues identified by MS (K90, K160, K247 and K264) and their neighbouring lysines (K252, K265, K278 and K291) with arginines (FAM134B-8KR). However, the overall ubiquitination level, binding to LC3B-II, formation of SDS-resistant oligomers, and number of FAM134B/LC3B-decorated ER fragments, were not affected by this mutant (Extended Data Fig. 1i,j).

2. In this study, the authors evaluated ER-phagy flux based on the lysosomal transport of FAM134B itself but should also examine that of other ER proteins such as CLIMP63 to more convincingly draw their conclusions.

We performed new assays to show that ER-phagy is reduced when cells express the 17KR mutant. We generated monoclonal U2OS cell lines co-expressing ssRFP-GFP-KDEL (Chino *et al.*, 2019) with either HA-FAM134WT or HA-FAM134B-17KR under the control of doxycycline-inducible promoters. We observed a general reduction in ER-phagy flux in the presence of FAM134B-17KR (Extended Data Fig. 8g-j). REEP5 degradation triggered by EBSS was also reduced in cells expressing this mutant (Extended Data Fig. 8j). These results confirm that FAM134B-RHD ubiquitination is important for FAM134B-mediated ER-phagy.

3. The 17KR mutation might affect the basic functions of FAM134B, such as its LC3 binding and membrane remodeling activity, rather than their enhancement via ubiquitination. The authors should exclude this possibility by confirming that these functions of FAM134B are not affected by the 17KR mutation using recombinant proteins.

Extended Data Fig. 8a shows the localisation of 17KR within the ER membranes, indicating that this mutant is capable of normal membrane insertion. In addition, we purified full-length GST-tagged FAM134B WT and 17KR from bacteria. Using these constructs, we conducted *in vitro* liposome remodelling assays, and found that the 17KR mutant does not affect the intrinsic ability of FAM134B to induce membrane curvature/shaping in liposomes (Extended Data Fig. 8d,e). Moreover, compared to liposomes treated with GST control, GST-FAM134B-WT and 17KR decreased the liposome diameter in a similar extent (Extended Data Fig. 8f).

It will also be great if the authors could reconstitute the enhancement of liposome fragmentation activity of FAM134B by its AMFR-mediated ubiquitination.

We also achieved the *in vitro* ubiquitination of full-length GST-FAM134B with recombinant full length AMFR. MS analysis revealed ubiquitination at K160, K278 and K299 (Extended Data Fig. 11m,n). FAM134B ubiquitination led to smaller proteoliposomes in comparison to the non-ubiquitinated protein, confirming that ubiquitination enhances the membrane remodelling activity of FAM134B (Fig. 5e).

4. Although how ubiquitination regulates FAM134B functions and behaviors was suggested by MD simulations, experimental validation would make them more reliable. Could the authors specify residues involved in interactions between ubiquitin and FAM134B as well as those between ubiquitin moieties on FAM134B? If possible, mutations into these residues may allow the authors to validate their models suggested by MD simulations.

We tested the role of Ub-Ub interactions by reducing the protein-protein interaction strength in our simulations. At $\alpha=0.65$, although there were fewer trans-Ub-Ub contacts in total, they could still induce the formation of RHD clusters. By contrast, at $\alpha=1.0$, the interactions were non-specific and spread throughout the Ub surface. Low-affinity interactions mediated by ubiquitin primarily allowed close RHD interactions, leading to the formation of nano-sized clusters and nucleation of the membrane bud (at $\Delta N=300$). However, by reducing the strength of the interactions even at $\Delta N=300$, we found only transient cluster formation but no bud formation, indicating that Ub-Ub interactions, although non-specific, are needed to stabilise dimers and higher-order oligomers to induce membrane budding and large-scale remodelling in the ER (Fig. 2d, columns 3 & 4, and Extended Data Fig. 6).

We mapped the critical interactions mediated by Ub onto its sequence and structure (Extended Data Fig. 6, 7) and found that almost the entire surface of Ub mediates non-specific interactions with other Ub-RHD molecules, leading to the formation of small and then larger clusters.

Validation of Ub-mediated interactions by direct mutagenesis experiments is challenging, especially when almost the entire surface of Ub is involved in non-specific interactions. Changes in Ub structure can have unintended effects on the entire quality control process. Therefore, we tested the role of Ub by making chimeric RHDs. We added two Ub molecules flanking the RHD to mimic the ubiquitinated RHDs. We found that they could remodel liposomes into smaller vesicles at the same concentration by inducing stronger curvature (Fig. 2c) compared to native RHD domains alone. This indicated that the addition of Ub close to the interfacial region of the bilayer could mediate strong interactions with the bilayer and with other molecules, favouring the formation of higher-order oligomers essential to remodel liposomes.

5. MD simulations suggested interactions between ubiquitin moieties on FAM134B. Has this type of ubiquitin-ubiquitin interaction been described in previous studies? Is this interaction possible because ubiquitin molecules are located at specific positions in FAM134B?

Ubiquitin bound to different sites on the RHD can be constrained geometrically. This allows varied interactions with membranes and other Ub moieties. In our simulations, we found that K160-Ub shows increased interactions with POPC lipids in the bilayer (Extended Data Fig. 2b), whereas K264-Ub does not (Extended Data Fig. 2c-d). In our simulations, the trans-interactions that mediate clustering were modelled only for Ub bound to K160 and K264. We found that Ub tethered to both sites showed similar interaction properties, and there were no significant differences in the residue-wise contacts. This is manifested as almost symmetric contact maps obtained even for intra-molecular K160-Ub and K264-Ub interactions. These results indicate that non-specific interactions primarily mediate the clustering of RHDs.

6. Fig. 3d: It would be interesting if the authors could estimate how many FAM134B molecules reside in microscale clusters.

The precise cluster size, geometry, organisation, and stoichiometry of oligomers observed under the simulation conditions are variable. It depends on the parameters controlling the protein-protein interaction strength, local membrane asymmetry, and the number of ubiquitinated RHD molecules in the simulation box. Our MD models so far show that, in the absence of Ub, transmembrane hairpins of three RHDs interact to form tight nano-clusters (Bhaskara et al., 2019; Siggel et al, 2021). The geometry and organisation of the cluster varies following the addition of Ub. Trans-Ub-Ub mediated interactions can extend RHD clusters to large sizes, such that all the RHD molecules within the box are part of the same cluster (e.g., Fig. 2c, column 3) and coat the membrane bud.

Based on our microscopy results, we inferred molecule numbers in nanoscale clusters via the kinetic analysis of single-molecule DNA-PAINT data. We found that ubiquitination favoured the oligomeric state of FAM134B in nanoclusters on average five-fold ($n_{17KR} = 2.5$, $n_{WT} = 12.6$), suggesting that ubiquitination promotes the assembly of high-density clusters (Fig. 3h).

7. Fig. 3e: Does the enlargement of FAM134B clusters upon Torin1 treatment depend on autophagosome formation? This will provide another insight into the mechanism of FAM134B clustering.

Using DNA-PAINT, we identified microscale and nanoscale clusters of HA-FAM134B (Fig. 3d). The microscale clusters corresponded to the puncta observed by confocal microscopy (Fig. 1h), which are autophagosomes FAM134B⁺, LC3B⁺ and Ub⁺. These structures enlarge following Torin 1 treatment because this treatment also triggers the ubiquitination of FAM134B. This raises the question if FAM134B ubiquitination is dependent on autophagosomes formation.

To determine whether the ubiquitination of FAM134B is dependent on autophagosome targeting, we used an LC3B-binding-deficient FAM134B mutant (LIR-mut) which is not targeted to autophagosomes (Khaminets, A. et al., 2015). FAM134B ubiquitination increases in the LIR-mut compared to FAM134B-WT (Extended Data Fig. 1n), indicating that autophagosomes are not required for the ubiquitination of FAM134B. Modification of FAM134B occurs before sequestration into autophagosomes. The effect observed in Fig. 3e can be attributed to an increase in the abundance of ubiquitinated FAM134B, which enhances its clustering and leads to the formation of larger complexes.

8. Extended Data Fig. 7g: The results seem to show that FAM134B resides in the tubular ER inconsistent with their original report that it localizes to ER sheets.

The aim of this figure was to demonstrate the integration of FAM134B within the ER rather than its preferred location. However, the reviewer raises a very important point.

The ER is a continuous endomembrane system and dynamic components can diffuse along the sheet and tubules

Our work expands what is known about FAM134B localisation and function. Using BICAP (Fig. 4d), we showed that FAM134B clusters can recruit RTN3, which was described as an ER tubule ER-phagy receptor (Grumati, et. al., 2017), as well; as REEPs (ER tubular proteins). We propose that FAM134B and RTN3 cooperate to degrade different ER compartments. BIFC analysis showed the co-localisation of FAM134B clusters with REEP5 (Fig. R6). These clusters are also LC3B⁺ (Extended Data Fig. 10a). REEP5 cellular turnover induced by Torin 1 is diminished when cells express LIR-deficient FAM134B, indicating that REEP5 is also a substrate for ER-phagy driven by FAM134B (Fig. R7). As highlighted in question 2, REEP5 degradation triggered by EBSS is reduced in cells expressing 17KR compared to those expressing WT FAM134B (Extended Data Fig. 8j).

9. Examining the size and shape of ER fragments within autophagosomes in FAM134B 17KR cells by electron microscopy may add important information how FAM134B ubiquitination affects the process of ER fission.

We thank the reviewer for this excellent suggestion. However, we decided to use other approaches to evaluate how ubiquitination affects ER morphology and membrane remodelling. We have performed the following *in vivo* and *in vitro* experiments:

First, we designed two FAM134B-RHD constructs: the RHD (from K90 to K264) and the chimera Ub-K90-RHD-K264-Ub. Both constructs lack the C-terminal LIR and both proteins localise in the ER. Larger and more abundant puncta were observed for the Ub-RHD-Ub construct (Extended Data 5, lower panels and inset) compared to RHD without Ub (a, upper panels and inset) (b). The Ub-RHD-Ub puncta were REEP5⁺ (Extended Data Fig. 5a,b). These structures resemble the clusters driven by ubiquitination.

Second, we purified GST-tagged FAM134B-RHD (from K90 to K264) and the chimera Ub-RHD-Ub. These recombinant proteins were reconstituted in liposomes for *in vitro* membrane shaping assays. The membrane remodelling was analysed by TEM and the diameter of the liposomes/vesicles was determined. Ub-RHD-Ub produced smaller proteoliposomes with a narrow distribution compared to the RHD without Ub (Fig. 2c), indicating a significant gain of membrane-remodelling activity for the chimera Ub-RHD-Ub. Taken altogether, the *in vivo* and *in vitro* evidence suggests that the ubiquitination of FAM134B enhances ER remodelling and membrane shaping.

10. Fig. 4d: The LC3B peak at distance of ~5 microns is FAM134B-positive and ubiquitin-negative. Does this suggest that ER-phagy can proceed without FAM134B ubiquitination?

In order to address this question, we performed additional co-localisation analysis to determine the fraction of Ub⁺ FAM134B autophagosomes in response to Torin 1 (Fig. 1h,i). We found that ~70% of HA-FAM134B/LC3B-II puncta were Ub⁺, indicating that ER-phagy driven by FAM134B is dependent on ubiquitin. However, Ub-independent pathways involving the participation of other ER-phagy receptors may also exist.

11. The authors identified many proteins including RHDs and RHD-like proteins by mass spectrometry of immunopurified FAM134B clusters. However, it is not convincing to me that all of these proteins are associated via direct interactions. Instead, some of these proteins might be concentrated in the clusters due to other reasons such as their preferring high membrane curvature but co-isolated with FAM134B due to incomplete membrane solubilization. This point should be discussed in the text.

We agree with this comment. The data suggest that all these proteins reside in the clusters either by direct or indirect interactions and we have adjusted the text accordingly.

Figure R6: Confocal fluorescence microscopy analysis of the BiFC signals following the interaction between V1-FAM134B and V2-FAM134B. Fixed cells expressing V1-FAM134B and V2-FAM134B were stained for REEP5 (red).

Figure R7: Cells expressing HA-FAM134BWT or HA-FAM134B-LIR incubated with 250 nM Torin 1 for 0, 2, 4, 6 and 8 h. Detergent-soluble extracts were analysed by western blot using antibodies against

12. Fig. 5: The authors should examine FAM134B ubiquitination in cells simply knocked out or knocked down for AMFR. This is important to evaluate how significantly endogenous AMFR contributes to FAM134B ubiquitination.

We performed this assay as suggested. Using MS, we evaluated FAM134B ubiquitination levels in AMFR knockdown cells. We found that AMFR knockdown reduced the ubiquitination of several lysine residues in the FAM134B-RHD (Fig. 5c). Because we did not completely abolish FAM134B ubiquitination by knocking down AMFR, other E3 ligase must also target FAM134B. Further studies will elucidate which other E3 ligases modify ER-phagy receptor clusters and whether their activity is cell-type specific or stress-inducible.

In addition, to show that AMFR targets FAM134B directly, we performed an *in vitro* ubiquitination assay on FAM134B-RHD₉₀₋₂₆₄ and the chimera Ub-RHD₉₀₋₂₆₄-Ub in the presence of purified AMFR (Extended Data Fig. 11p-q). Proteomics analysis revealed that K160 of RHD and K160/K241 of Ub-RHD-Ub are targeted directly for ubiquitination by AMFR. Moreover, we found that ubiquitination promoted the formation of larger super-complexes, which were detected by blue native polyacrylamide gel electrophoresis (BN-PAGE) (Extended Data Fig. 11r).

13. Extended Data Fig. 1c and d: A decrease in the protein level of FAM134B is not completely blocked by BafA1. This might suggest that ubiquitinated FAM134B is also degraded by the proteasome. Is this true? Given this possibility, it would be necessary to confirm that TAK243 indeed blocks autophagic degradation of FAM134B rather than its proteasomal degradation.

This is an interesting and valid point. Our data indicate that the pool of ubiquitinated FAM134B is degraded by lysosomes (Extended Data Fig. 1a,b). In addition, BafA1 but not MG132 causes the accumulation of total endogenous FAM134B (Extended Data Fig. 1c), indicating that FAM134B is not a substrate of the proteasome.

Moreover, proteasome proteins were poorly and not significantly detected in our proteomic analysis (Fig. 4d; Extended Data Fig. 9c,d) whereas ubiquitinated FAM134B accumulated in the LIR-mut compared to FAM134B-WT (Extended Data Fig. 1n). Ubiquitinated FAM134B therefore undergoes lysosomal degradation via ER-phagy.

14. Extended Data Fig. 9g and 9h and Fig. 5f and 5g: The authors should confirm that these changes in FAM134B levels were due to degradation via ER-phagy rather than proteasomal degradation.

First, in Figure 5f,g (Figure 5a,b in the revised manuscript) we showed that FAM134B levels decreased in response to Torin 1, but are rescued in response to Torin1 + BafA1 (Extended Data Fig. 1a,b). Torin 1 thus induces FAM134B degradation via ER-phagy.

Furthermore, in Figure 5f,g (Figure 5a,b in the revised manuscript) and extended Data Fig. 9g,h (Extended Data Fig. 11a,b in the revised manuscript) we observed that AMFR knockdown causes the accumulation of FAM134B under basal conditions and delayed FAM134B turnover induced by Torin 1. In order to determine whether these changes in FAM134B levels are due ER-phagy rather than proteasome degradation, we made use of cells expressing the ER-phagy flux reporter mCherry-EGFP-tagged FAM134B WT. The siRNA-mediated knockdown of AMFR significantly slowed the flux of FAM134B-mediated ER-phagy compared to control siRNA (siNT) in cells treated with Torin 1 (Fig. 5d). These data indicate that AMFR regulates FAM134B levels via ER-phagy.

In addition, BafA1 (ER-phagy flux inhibitor) but not MG132 (proteasome inhibitor) increases the total accumulation of endogenous FAM134B (Extended Data Fig. 1c).

15. Fig. 5f and Extended Data Fig. 9i: The authors should show that a decrease in AMFR levels caused by Torin 1 treatment depends on ER-phagy.

As requested, we evaluated the degradation of AMFR and its dependence on ER-phagy. First, the BIFC signal produced by interaction between V1-FAM134B and V2-FAM134B (clusters) co-localises with endogenous AMFR and LC3B-II (Extended Data Fig. 10a,b). AMFR is first recruited to clusters to ubiquitinate ER-phagy receptor complexes and is then degraded by ER-phagy. In addition, the cellular turnover of endogenous AMFR induced by Torin 1 was significantly impaired in the presence of the FAM134B LIR-mut (Extended Data Fig. 11i,j). Similar results were obtained in cells expressing 17KR (Extended Data Fig 11f-h). These data confirm that AMFR degradation is dependent on FAM134B-driven ER-phagy.

16. The authors seem to assume that FAM134B is mono-ubiquitinated. Is there any evidence for it?

ER-phagy receptors are mono-ubiquitinated at multiple sites based on our proteomic mapping of several ubiquitinated lysine residues (Fig. 1a,b and Extended Data 9b). In proteomic samples of immunoprecipitated FAM134B, we found peptides corresponding to linkages between Ub chains (diGly with K63, K48 and K27 linkages). However, we detected the same Ub linkages in 17KR immunoprecipitates, suggesting that FAM134B interacts with other proteins that are polyubiquitinated.

17. Lines 209-211 "dimeric FAM134B (observed using BiCAP) was found in ER fragments" and lines 250-251 "FAM134B-mediated ER-fragmentation in response to Torin 1 was also reduced": How could the authors determine

that FAM134B puncta represent ER fragments? These puncta might represent FAM134B clusters in the ER network.

We defined ER fragments as puncta containing FAM134B, LC3B-II and ER proteins that do not bind LC3B directly. These ER proteins can act as ER-phagy regulators but also as substrates. For example, BiCAP analysis revealed the co-localisation of FAM134B (BIFC signal), LC3B-II and AMFR (Extended Data Fig. 10a) or REEP5 in cellular puncta (Fig. R6).

Now, we are able to detect puncta that represent FAM134B clusters in the ER network by characterising in mammalian cells the artificial construct FAM134B-RHD₉₀₋₂₆₄ and the chimera Ub-RHD₉₀₋₂₆₄-Ub. We found a significant increase in the number of RHD/REEP5-containing puncta in cells expressing Ub-RHD-Ub. These puncta may include clusters of RHD-containing proteins, which are enhanced by the presence of ubiquitinated-FAM134B-RHDs (Extended Data Fig. 5a,b). These proteins do not have their LIR. Accordingly, we captured the clustering event involving the Ub-RHD-Ub chimera in the ER before autophagosome targeting.

18. Fig. 4c: Arrows seem not to be correctly positioned.
We have now positioned the arrows correctly.

19. Fig. 5i should be cited in the text. It would also be better if the authors could depict the interaction between FAM and LC3B in this model.
We have cited Fig.5i (now Fig. 5f) in the text.

In addition, to confirm the specificity of the BiCAP assay, we included the ER-phagy receptor CCPG1 (an ER-resident protein that does not form part of the clusters) as a negative control. This was achieved by co-expressing V1-CCPG1 with V2-FAM134B. Immunofluorescence analysis (Fig. R8) showed that Venus was not efficiently reconstituted, indicating the absence of interaction. Importantly, both proteins were expressed and localized at the ER (as indicated by the CCPG1 and FAM134B staining).

Figure **R8**: Specificity of the BiCAP assay: Fixed cells expressing V1-CCPG1 and V2-FAM134B were stained for CCPG1 (red) and the V2 fragment (mAb AntiGFP clone 3H9), in order to detect FAM134B (gray). BiFC (green) signals are not observed in cells expressing CCPG1 and FAM134B, indicating no interaction. .

Reviewer Reports on the First Revision:

Referees' comments:

Referee #1 (Remarks to the Author):

In this extensive revision, the authors have addressed most of my points through additional experiments (eg intermediate K>R mutants, liposome deformation experiments), additional simulations (bicelle deformation at 280 K) and textual clarifications, making this very original but complex study more compelling.

Referee #2 (Remarks to the Author):

The authors have satisfactorily addressed all the issues I raised in the review of the original manuscript.